# Variational Mirror Descent for Robust Learning in Schrödinger Bridge

## Abstract

Schrödinger bridge (SB) has evolved into a universal class of probabilistic generative models. Recent studies regarding the Sinkhorn algorithm through mirror descent (MD) have gained attention, revealing geometric insights into solution acquisition of the SB problems. In this paper, we propose a variational online MD framework for the SB problems, which provides further stability to SB solvers. We formally prove convergence and a regret bound $\mathcal{O}(\sqrt{T})$ of online mirror descent under mild assumptions. As a result of analysis, we propose a simulation-free SB algorithm called Variational Mirrored Schrödinger Bridge (VMSB) by utilizing the Wasserstein-Fisher-Rao geometry of the Gaussian mixture parameterization for Schrödinger potentials. Based on the Wasserstein gradient flow theory, our variational MD framework offers tractable gradient-based learning dynamics that precisely approximate a subsequent update. We demonstrate the performance of the proposed VMSB algorithm in an extensive suite of benchmarks.

## 1 Introduction

Schrödinger bridge (SB; Schrödinger, 1932) has emerged as a universal class of probabilistic generative models. However, learning methods of SB remain somewhat *atypical*, each requiring a sophisticated approach to derive a solution. Recently, learning an SB model with Sinkhorn (Peyré et al., 2019) has been generalized into a collection of convex optimization methods, called mirror descent (MD; Nemirovsky & Yudin, 1983; Léger, 2021; Aubin-Frankowski et al., 2022). For a parameters sequence $\{w_t\}_{t=1}^T$ and a convex function $\Omega$, an update of MD for a cost function $F_t$ is derived as

$$\nabla\Omega(w_{t+1}) = \nabla\Omega(w_t) - \eta_t\nabla F_t(w_t). \tag{1}$$

In the equation, the gradient operation denoted as $\nabla\Omega(\cdot)$ creates a transformation that links a parametric space to a dual space. The collective perspective of considering SB problems (SBPs) as an ordinary instance of optimization problems broadly opens new avenues for algorithmic advancements of probabilistic generative models in a learning theoretical direction, particularly within the context of the learning theory and stability improvements in probabilistic generative modeling.

In general, one can consider constrained distributional optimization problems with generalized gradient dynamics on the space of distributions endowed with the Wasserstein metric. Leveraging the Wasserstein gradient flow discovered by Jordan, Kinderlehrer, and Otto (JKO; Jordan et al., 1998), the desired dynamics of a functional $F : \mathcal{P}_2(\mathcal{X}) \to \mathbb{R}$ can be modeled, where $\mathcal{P}_2(\mathcal{X})$ denotes the set of probability distributions with finite second-order moments. Despite the extensive theoretical findings of the Wasserstein gradient flow regarding OT problems (Ambrosio et al., 2005a; Santambrogio, 2015; Villani, 2021), the computational challenges remain. The established methods are commonly based on numerical methods for partial differential equations (PDEs) (Carlier et al., 2017; Carrillo et al., 2023), whose exhaustive numerical computations make them unsuitable for systems with high-dimensional probability densities.

A favored strategy to mitigate this issue is to narrow down the solution space into a subset of tractable distributions, often referred to as taking a *variational* form (Paisley et al., 2012; Blei et al., 2017). For example, mean-field formulations of SB (Liu et al., 2022; Claisse et al., 2023) are variational approximations. Unfortunately, this does not faithfully yield an analytical submanifold and it is obligated to physically simulate among particles. Recently, a Gaussian mixture parameterization of the Schrödinger potentials has been proposed by Korotin et al. (2024). The simulation-free *LightSB*

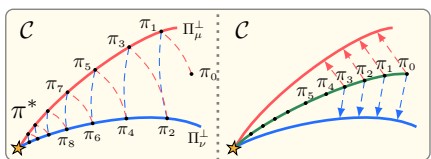

Figure 1: Learning for an SB model $\{\pi_t\}_{t=1}^{\infty}$. We propose to learn in the distributional space $\mathcal{C}$. Left: Sinkhorn (Lemma 1). Right: Steepest Wasserstein descent in $\mathcal{C}$ (Lemma 2).

Table 1: A technical overview. VMSB is a simulation-free algorithm that iteratively produces solutions. Our VMSB additionally provides a strong theoretical guarantee of convergence.

| | Iterative | Simulation-free | Regret bound |
|---|---|---|---|
| DSB (De Bortoli et al.) | ✓ | ✗ | ✗ |
| DSBM (Shi et al.) | ✓ | ✗ | ✗ |
| LightSB (Korotin et al.) | ✗ | ✓ | ✗ |
| LightSB-M (Gushchin et al.) | ✗ | ✓ | ✗ |
| **VMSB (ours)** | ✓ | ✓ | ✓ |

solver is simple yet general, with the guarantee of universal approximation for SB. The expressiveness of the solver coincides with geometric properties of Gaussian variational inference and mixture models (Chen et al., 2018; Daudel et al., 2021; Diao et al., 2023). However, its shortcoming—as well as other simulation-free solvers (Tong et al., 2023; Gushchin et al., 2024a)—is the uncertainty of data-driven learning signals of non-convex objectives. This reveals room for improvement in the rich geometric properties of SB using a variational framework.

In this paper, we explore a new way of stable Schrödinger bridge acquisition through the lens of online mirror descent (OMD; Srebro et al., 2011). As illustrated in Fig. 1, we utilize a constrained space $\mathcal{C}$ equipped with the Wasserstein metric, allowing a new formulation similar to the classical mirror descent algorithm. As an online learning algorithm, we postulate the optimization errors of an SB solver and propose an OMD framework to reduce these errors in terms of regrets. To this end, we propose a new simulation-free SB algorithm called Variational Mirrored Schrödinger Bridge (VMSB). Learning of VMSB is based on an approximation of the MD updates that solve iterative subproblems by Wasserstein gradient dynamics. We introduce a gradient computation method of parameterized SB models based on gradient flows with respect to Wasserstein-Fisher-Rao (WFR) geometry (Liero et al., 2018). Our framework allows us to efficiently perform OMD, which is more tolerant of unreliable objective estimation (Lei & Zhou, 2020). Our experiments show that the proposed VMSB outperforms existing SB solvers in benchmark problems.

**Our contributions.** Our work complements earlier studies on SB, building on the theoretical and technical insights derived from a geometric perspective that views MD solutions as gradient flows across the Wasserstein space. To the best of our knowledge, VMSB is the first SB algorithm based on online mirror descent that verifies its ability to solve high-dimensional real-world SB problems. Table 1 shows that VMSB is a simulation-free SB solver that brings solid convergence results in general situations. We summarize our main contributions below:

- Based on the learning theory, we derive gradient-based OMD update rules that provide robust dynamics for reaching local objectives, which ensures a rigorous regret bound (§ 4).
- We propose a new SB solver based on the Wasserstein-Fisher-Rao geometry, which retains asymptotic stability results in Wasserstein gradient flows (§ 5).
- We demonstrate our algorithm on a variety of SBPs demonstrating the effectiveness of the learning theoretic approach in the Schrödinger bridge problems (§ 6).

## 2 RELATED WORKS

**MD and Sinkhorn.** The Bregman divergence (Bregman, 1967) is a family of statistical divergence that is particularly useful when analyzing constrained convex problems in various settings (Beck & Teboulle, 2003; Boyd & Vandenberghe, 2004; Hiriart-Urruty & Lemaréchal, 2004). Notably, Léger (2021) and Aubin-Frankowski et al. (2022) adopted the Bregman divergence into entropic optimal transport (EOT; Peyré et al., 2019) and SBPs with probability measures, and the studies revealed that Sinkhorn can be considered to be an MD with a constant step size $\eta \equiv 1$. In statistical geometries, the Bregman divergence is a first-order approximation of a Hessian structure (Shima & Yagi, 1997; Butnariu & Resmerita, 2006), which is natural discretization on a gradient flow. Deb et al. (2023) introduced Wasserstein mirror flow, and the results include a geometric interpretation of Sinkhorn for unconstrained OT, *i.e.*, when $\varepsilon \to 0$ in our setup. Karimi et al. (2024) formulated a half-iteration of the Sinkhorn algorithm for SB into a mirror flow, *i.e.*, $\eta_t \to 0$ with a continuous-time formulation.

**Wasserstein Gradient Flows** have drawn significant attention whose geometry is formally described by the Wasserstein-2 metric (Ambrosio et al., 2005a; Villani, 2009; Santambrogio, 2017). Otto (2001) introduced a formal Riemannian structure to interpret various evolutionary equations as gradient flows with the Wasserstein space, which is closely related to our variational approach. The mirror Langevin dynamics is an early work describing the evolution of the Langevin diffusion (Hsieh et al., 2018), and was later incorporated in the geometry of the Bregman Wasserstein divergence (Rankin & Wong, 2023). We relate our methodology with recent approaches of variational inference on the Bures-Wasserstein space (Lambert et al., 2022; Diao et al., 2023). Utilizing Bures-Wasserstein geometry, the Wasserstein-Fisher-Rao geometry (Liero et al., 2016; Chizat et al., 2018; Liero et al., 2018) additionally provides "liftings," which yield an interaction among measures.

**Learning Theory.** Suppose we have time-varying costs $\{F_t\}_{t=1}^{\infty}$. We generally referred to learning through these signals as *online learning* (Fiat & Woeginger, 1998). Our interest lies in temporal costs defined in a probability space, where following the ordinary gradient may not the best choice due to the geometric constraints (Amari, 2016; Amari & Nagaoka, 2000). In this sense, we primarily relate our work to the online form of MD (Srebro et al., 2011; Raskutti & Mukherjee, 2015; Lei & Zhou, 2020). Another relevant design of the online algorithm is the follow-the-regularized-leader (FTRL; McMahan, 2011; Chen & Orabona, 2023), where the distinction between two schemes is the way of handling costs and regularization. OMD focuses on minimizing a current loss, dynamically scheduling proximity of updates through $\{\eta_t\}_{t=1}^{T}$. In contrast, FTRL aims to minimize historical losses $\sum_t F_t(w)$ with a fixed regularization term.

## 3 PRELIMINARIES

Let $\mathcal{P}(\mathcal{S})$ ($\mathcal{P}_2(\mathcal{S})$) denote the set of (absolutely continuous) Borel probability measures on $\mathcal{S} \subseteq \mathbb{R}^d$ (with a finite second moment). For a transport plan $\pi$, a notation $\vec{\pi}^x$ ($\vec{\pi}^y$) denotes a conditional distribution $\vec{\pi}(\cdot|x)$ ($\bar{\pi}(\cdot|y)$; see Fig. 2). We use $\mathrm{KL}(\cdot\|\cdot)$ to denote the KL functional and assume $+\infty$ if an argument is not absolutely continuous. We employ $\mathbb{P}([0,1], \mathcal{S})$ for a set of trajectories from time 0 to 1.

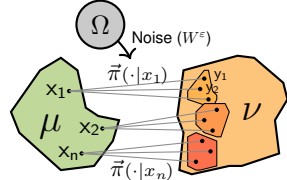

Figure 2: An SB problem.

For marginals $\mu, \nu \in \mathcal{P}_2(\mathcal{S})$ and a regularization coefficient $\varepsilon \in \mathbb{R}^+$, the EOT/SB problem with a quadratic cost function is defined as finding the unique minimizer $\pi^*$ for the following problem:

$$\mathrm{OT}_\varepsilon(\mu, \nu) := \inf_{\pi \in \Pi(\mu, \nu)} \iint_{\mathcal{S} \times \mathcal{S}} \tfrac{1}{2}\|x - y\|^2 \, \mathrm{d}\pi(x, y) + \varepsilon \mathrm{KL}(\pi\|\mu \otimes \nu), \tag{2}$$

where $\Pi(\mu, \nu)$ denotes the set of couplings (Peyré et al., 2019) and $\mu \otimes \nu$ is the product of measures. For an induced dual problem the constrained optimization (2), consider the *log-Schrödinger potentials* (Nutz, 2021) $(\varphi^*, \psi^*) \in L^1(\mu) \times L^1(\nu)$, which represent the EOT solution with $\mathrm{d}\pi^* = e^{\varphi^* \oplus \psi^* - c_\varepsilon}\mathrm{d}(\mu \otimes \nu)$, $(\mu \otimes \nu)$-almost surely, for the quadratic cost $c_\varepsilon(x, y) := \frac{1}{2\varepsilon}\|x - y\|^2$. The Sinkhorn algorithm is given as the following updates (Cuturi, 2013):

$$\psi_{2t+1}(y) = -\log \int_{\mathcal{S}} e^{\varphi_{2t}(x) - c_\varepsilon(x,y)}\mu(\mathrm{d}x), \quad \varphi_{2t+2}(x) = -\log \int_{\mathcal{S}} e^{\psi_{2t+1}(x) - c_\varepsilon(x,y)}\nu(\mathrm{d}y), \tag{3}$$

where each update for a potential is called iterative proportional fitting (IPF; Kullback, 1968). Let $W^\varepsilon \in \mathbb{P}$ be the Wiener process with volatility $\varepsilon$. The fundamental equivalence between EOT and SB (Pavon & Wakolbinger, 1991; Léonard, 2012) allows us to consider the optimality $\pi^*$ when solving the Schrödinger bridge problem, and we can transform $\pi^*$ to $\mathcal{T}^*$ such that:

$$\mathcal{T}^* := \underset{\mathcal{T} \in \mathcal{Q}(\mu, \nu)}{\arg\min} \mathrm{KL}(\mathcal{T}\|W^\varepsilon), \tag{4}$$

where $\mathcal{Q}(\mu, \nu) \subset \mathbb{P}(\mathcal{S}, [0, 1])$ is the set of processes with marginals $\mu$ and $\nu$. The SB process $\mathcal{T}^*$ is uniquely describe by a stochastic differential equation (SDE; Léonard, 2013): $\mathrm{d}X_t = g^*(t, X_t) + \mathrm{d}W_t^\varepsilon$ in $t \in [0, 1]$ with an optimal drift function $g^*$. Under the Girsanov theorem for the stochastic processes (Vargas et al., 2021), the Sinkhorn scheme can be designed as a drift matching algorithm.

Léger (2021) and Aubin-Frankowski et al. (2022) have discovered a major link between Sinkhorn and MD: solving SB with Sinkhorn corresponds to MD with a constant step size $\eta_t \equiv 1$. Since our objective does not ensure Gâteaux differentiablility (see Definition 4), one needs an alternative for a generalized notion of derivatives. Consequently, we provide the definitions of *directional derivatives* (Aliprantis & Border, 2006) and *first variations* (Aubin-Frankowski et al., 2022).

**Definition 1** (Directional derivative). Given a locally convex topological vector space $\mathcal{M}$, The directional derivative of $F$ in the direction $\xi$ is defined as $d^+F(x;\xi) = \lim_{h \to 0^+} \frac{F(x+h\xi)-F(x)}{h}$.

**Definition 2** (First variation). Given a topological vector space $\mathcal{M}$ and a convex constraint $\mathcal{C} \subseteq \mathcal{M}$, for a function $F$ and $x \in \mathcal{C} \cup \mathrm{dom}(F)$, define the first variation of $F$ over $\mathcal{C}$ to be an element $\delta_{\mathcal{C}}F(x) \in \mathcal{M}^*$, where $\mathcal{M}^*$ is the topological dual of $\mathcal{M}$, such that it holds for all $y \in \mathcal{C} \cup \mathrm{dom}(F)$ and $v = y - x \in \mathcal{M}$: $\langle \delta_{\mathcal{C}}F(x), v \rangle = d^+F(x;v)$. $\langle \cdot, \cdot \rangle$ denotes the duality product of $\mathcal{M}$ and $\mathcal{M}^*$.

From the above definitions, we can consider a Bregman divergence defined with a weak notion of the directional derivative, enabling a formal analysis akin to standard convex optimization problems. Following Karimi et al. (2024), we explicitly set the Bregman potential $\Omega = \mathrm{KL}(\cdot \| e^{-c_\varepsilon} \mu \otimes \nu)$ in the SB problems, which enforces the Gibbs parameterization for EOT couplings.

**Definition 3** (Bregman divergence). Let $\Omega : \mathcal{M} \to \mathbb{R} \cup \{+\infty\}$ be a convex functional. Define the Bregman divergence as $D_\Omega(x\|y) := \Omega(x) - \Omega(y) - d^+\Omega(y; x - y)$, for all $x, y \in \mathcal{M}$.

Lastly, our analysis requires a certain form of measure concentration to address the desired properties of OMD. Thus, we primarily works with asymptotically log-concave distributions initially discussed by Otto & Villani (2000). Let us define asymptotically log-concave distributions on $\mathbb{R}^d$:

$$\mathcal{P}_{\mathrm{alc}}(\mathbb{R}^d) := \{\zeta(\mathrm{d}x) = \exp(-U(x))\mathrm{d}x : U \in C_2(\mathbb{R}^d), U \text{ is asymptotically strongly convex}\} \quad (5)$$

Since $\mathcal{P}_{\mathrm{alc}}$ ensures the log Sobolev inequality (LSI; Gross, 1975), providing Fisher information as an upper bound of the KL functional. We defer the additional theoretical details to Appendix A.

## 4 LEARNING SCHRÖDINGER BRIDGE VIA ONLINE MIRROR DESCENT

The goal in this section is to derive an OMD update rule for SB, and analyze its convergence. To accomplish this, we postulate on the existence of temporal estimates and an online learning problem. Our analysis suggests that applying an MD approach can reduce the uncertainty of these estimates.

### 4.1 SINKHORN AND WASSERSTEIN DESCENT

We start with our characterization of Sinkhorn and a static MD variant illustrated in Fig. 1, which will lead to a better understanding of the OMD framework. Using the first variation $\delta_{\mathcal{C}}$ in Definition 2 instead of standard gradient $\nabla$, we write a proximal form of an MD update as (Karimi et al., 2024)

$$\pi_{t+1} = \arg\min_{\pi \in \mathcal{C}} \left\{ \langle \delta_{\mathcal{C}}F_t(\pi_t), \pi - \pi_t \rangle + \frac{1}{\eta_t} D_\Omega(\pi\|\pi_t) \right\}, \quad (6)$$

where $F_t$ denotes a temporal cost function for SB models in $\mathcal{C}$. In Eq. (6), the updates are determined by the first order approximation of $F_t$ and proximity of previous iterate $\pi_t$ with respect to the Bregman divergence (Beck & Teboulle, 2003). We assume that a parameterized SB model $\pi_t = e^{\varphi_t \oplus \psi_t - c_\varepsilon}(\mu \otimes \nu)$ obeys the following constraints for marginals and potentials:

$$\mathcal{C} := \left\{ \pi : (\mu, \nu) \in \mathcal{P}_2(\mathbb{R}^d) \cap \mathcal{P}_{\mathrm{alc}}(\mathbb{R}^d), (\varphi, \psi) \in L^1(\mu) \times L^1(\nu), \text{ and } \varphi, \psi \in C^2(\mathbb{R}^d) \cap \mathrm{Lip}(\mathcal{K}) \right\},$$

where $\mathrm{Lip}(\mathcal{K})$ denotes a set of functions with $\mathcal{K}$-Lipschitz continuity. Using the model space $\mathcal{C}$, IPF projections Eq. (3) writes as following subproblems of alternating Bregman projections:

$$\arg\min_{\pi \in \Pi_\mu^\perp} \left\{ \mathrm{KL}(\pi\|\pi_{2t}) : \pi \in \mathcal{C}, \gamma_2\pi = \nu \right\}, \quad \arg\min_{\pi \in \Pi_\nu^\perp} \left\{ \mathrm{KL}(\pi\|\pi_{2t+1}) : \pi \in \mathcal{C}, \gamma_1\pi = \mu \right\}, \quad (7)$$

where $\gamma_1\pi(x) := \int \pi(x, y)\mathrm{d}y$ and $\gamma_2\pi(y) := \int \pi(x, y)\mathrm{d}x$ and the symbols $(\Pi_\mu^\perp, \Pi_\nu^\perp)$ denote the Sinkhorn projection spaces that preserve the property of marginals. As an optimization problem in $\mathcal{C}$, one can consider a temporal cost $\widetilde{F}_t(\pi) := a_t \mathrm{KL}(\gamma_1\pi\|\mu) + (1 - a_t)\mathrm{KL}(\gamma_2\pi\|\nu)$ with sequence $\{a_t\}_{t=1}^\infty = \{0, 1, 0, 1, \dots\}$. By construction, MD for $\widetilde{F}_t$ with a step size $\eta_t \equiv 1$ matches the Sinkhorn.

**Lemma 1** (Sinkhorn). *For $\Omega = \mathrm{KL}(\pi\|e^{-c_\varepsilon}\mu\otimes\nu)$, iterates from $\pi_{t+1} = \arg\min_{\pi \in \mathcal{C}} \{\langle \delta_{\mathcal{C}}\widetilde{F}_t(\pi_t), \pi - \pi_t\rangle + D_\Omega(\pi\|\pi_t)\}$ is equivalent to estimates from $(\varphi_t, \psi_t)$ of (3), for every update step $t \in \mathbb{N}_0$.*

In contrast, we can alternatively consider a "static" objective, namely $F(\cdot) := \mathrm{KL}(\cdot\|\pi^*)$, where the KL functional is originated from the formal definition of SBP (Vargas et al., 2021; Chen et al., 2022). The following lemma show that the MD updates directly correspond to Wasserstein gradient descent on SB models, which can be considered as the Riemannian steepest descent in the space $\mathcal{C}$.

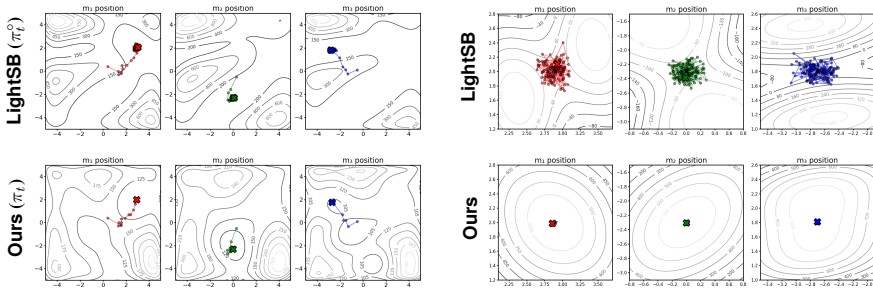

Figure 4: Loss landscapes and gradient dynamics in a 2D problem. Left: In an early stage, parameters of three modalities $\{m_k\}_{k=1}^{3}$ (mean estimations) for both LightSB (top) and VMSB (bottom) methods approach the optimality with different costs. Right: In a late stage, while LightSB is vibrant (magnified 10 times), whereas our method emits strictly convex landscape and stable dynamics.

**Lemma 2** (Wasserstein descent). *Suppose that $F(\pi) \coloneqq \mathrm{KL}(\pi\|\pi^*)$ and $f(\vec{\pi}^x) \coloneqq \mathrm{KL}(\vec{\pi}^x\|(\vec{\pi}^*)^x)$ for $\pi \in \mathcal{S}$. The MD formulation of $F$ corresponds to a discretization of a geodesic flow such that $\lim_{\eta_t \to 0^+} \frac{\pi_{t+1}^x - \pi_t^x}{\eta_t} = -\nabla_{\mathrm{w}} f(\vec{\pi}_t^x)$, where $\nabla_{\mathrm{w}}$ denotes the Wasserstein-2 gradient operator.*

Therefore, updates for $F(\cdot)$ approximately lies the geodesic of $\mathcal{C}$ in terms of Wasserstein-2 metric. Note that optimizing the cost ensures unbiased minimization (green line in Fig. 1) in $\mathcal{C}$. This interpretation allows us to consider $F(\cdot)$ as the ground truth cost in our SB framework.

## 4.2 THEORETICAL ANALYSIS

In contrary to the ideal case of Lemma 2, we postulate on an online learning problem that nonstationary estimates $\{\pi_t^\circ\}_{t=1}^{\infty}$ are offered instead of $\pi^*$ as learning signals, making an optimization process with $F_t(\cdot) \coloneqq \mathrm{KL}(\cdot\|\pi_t^\circ)$. We require some geometric conditions on $\{\pi_t^\circ\}_{t=0}^{\infty}$ to start our analysis. As previously studied (Bernhard & Rapaport, 1995; Karimi et al., 2024), the directional derivative of the Fenchel conjugate $\Omega^*$ of $\Omega + i_\mathcal{C}$, $\Omega$ with an indicator function $i_\mathcal{C}$ (defined as $i_\mathcal{C}(x) = 0$ if $x \in \mathcal{C}$ and $+\infty$ otherwise), exists by the Danskin's theorem, such that

$$\delta_\mathcal{D}\Omega^*(\varphi \oplus \psi) = \arg\max_{\pi \in \mathcal{C}}\{\langle\varphi \oplus \psi, \pi\rangle - \Omega(\pi)\},$$

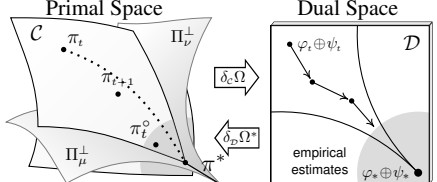

Figure 3: A schematic illustration. The primal and dual spaces $(\mathcal{C}, \mathcal{D})$ retain bidirectional maps $(\delta_\mathcal{C}\Omega, \delta_\mathcal{D}\Omega^*)$. $\Pi_\nu^\perp$ and $\Pi_\mu^\perp$ indicate projection spaces of $\gamma_1\pi = \mu$ and $\gamma_2\pi = \nu$, respectively. The current $\pi_t$ performs an update following a "unreliable" leader $\pi_t^\circ$ in a region shaded in gray.

where every direct sum of potentials $\varphi \oplus \psi = \delta_\mathcal{C}\Omega(\pi) \in \mathcal{D} \coloneqq \delta_\mathcal{C}\Omega(\mathcal{C})$ represent an element of the generalized dual space. In the dual geometry illustrated in Fig. 3, we assume uncertainty of the ground truth in $\mathcal{D}$, characterized with the following assumption.

**Assumption 1** (Dually stationary process). Suppose a process $\{\pi_t^\circ\}_{t=1}^{\infty} \subset \mathcal{C}$ with ergodicity (Cornfeld et al., 2012) of $\{\delta_\mathcal{C}\Omega(\pi_t^\circ)\}_{t=1}^{\infty}$. Consider $\pi_\mathcal{D}^\circ \in \mathcal{C}$, which is a primal representation for an asymptotic mean upon $\mathcal{D} = \delta_\mathcal{C}\Omega(\mathcal{C})$: $\pi_\mathcal{D}^\circ \coloneqq \delta_\mathcal{D}(\lim_{t\to\infty} \frac{1}{t}\sum_t \delta_\mathcal{C}\Omega(\pi_t^\circ)])$.

The assumption manifests statistical properties (such as the mean) that $\{\pi_t^\circ\}_{t=0}^{\infty}$ remain in a stationary region as $T \to \infty$. This is closely related asymptotically mean stationary processes (Gray & Kieffer, 1980) which have been used to analyze stochastic dynamics.[1] Fig. 4 demonstrates our objective that OMD stabilizes learning of $\pi_t$, even when the reference $\pi_t^\circ$ tends to have perturbation.

We state two step size conditions, which will be justified in Theorem 1 and Proposition 1.

**Assumption 2** (Step sizes). Assume two conditions for $\{\eta_t\}_{t=0}^{\infty}$. (a) *Convergent sequence & divergent series:* $\lim_{t\to\infty} \eta_t = 0$ and $\sum_{t=1}^{\infty} \eta_t = \infty$. (b) *Convergent series for squares:* $\sum_{t=1}^{\infty} \eta_t^2 < \infty$.

Using the conditions above, we firstly argue that online mirror descent with respect to Bregman potential $\Omega = \mathrm{KL}(\cdot\|e^{-c_\varepsilon}\mu \otimes \nu)$ requires Assumption (2a) for the sake of convergence.

---

[1]Since iterates are updated through dual parameters in MD, we refer to the process as being dually stationary.

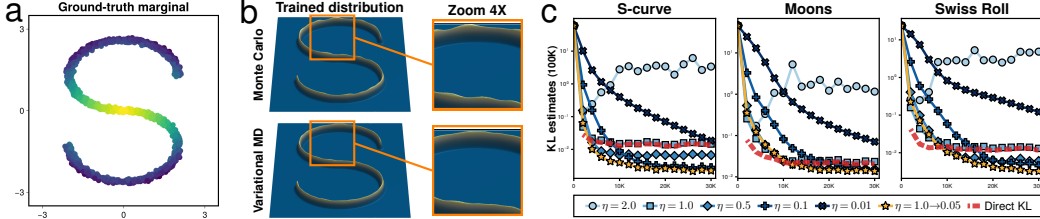

Figure 5: Variational MD with synthetic datasets. (a) A distribution is accessible by finite batch data. (b) 3D surfaces of $(\vec{\pi}_T^\circ, \vec{\pi}_T)$ trained by Monte Carlo method for KL (top) and variational MD (bottom) show that the MD results in more stable outcomes. (c) The plots show the estimated $\text{KL}(\vec{\pi}_t \| \vec{\pi}^*)$ with different step size scheduling (5 runs), with red dashed baselines $\text{KL}(\vec{\pi}_t^\circ \| \vec{\pi}^*)$.

**Theorem 1** (Step size considerations). *Suppose a Bregman potential $\Omega = \text{KL}(\cdot \| e^{c_\varepsilon} \mu \otimes \nu)$ and strongly convex $c_\varepsilon$. Assume the idealized case of $\pi_D^\circ = \pi^*$. Then, for $\{\pi_t\}_{t_1}^T \subset \mathcal{C}$ we get $\lim_{T \to \infty} \mathbb{E}_{1:T}[D_\Omega(\pi_D^\circ \| \pi_T)] = 0$ if and only if Assumption (2a) is satisfied. Furthermore, if the step size is in the form of $\eta_t = \frac{2}{t+1}$, then $\mathbb{E}_{1:T}[D_\Omega(\pi^* \| \pi_t)] = \mathcal{O}(1/T)$.*

Therefore, we can assure for the ideal convergence in the SB learning when the scheduling of $\eta_t$ follows the step size assumptions. Next, we show that almost sure convergence toward $\pi_D^\circ$ is guaranteed under Assumption (2b). Given the convex nature of SB cost functionals, we argue that this convergence toward $\pi_D^\circ$ is beneficial as long as $\pi_t^\circ$ is trained to approximate $\pi^*$ and remain bounded. Therefore, we argue that the convergence of SB is beneficial and address the following statement.

**Proposition 1** (Convergence). *Suppose that $\pi^* \neq \pi_D^\circ$, hence $\inf_{\pi \in \mathcal{C}} \mathbb{E}[F_t(\pi)] > 0$. If the step sizes $\{\eta_t\}_{t=0}^\infty$ satisfies Assumption 2, then $\lim_{t \to \infty} \mathbb{E}_{1:t}[D_\Omega(\pi_D^\circ \| \pi_t)]$ converges to 0 almost surely.*

Lastly, assume that a type of log Sobolev inequality holds (see Assumption 3) with continuity of potentials. We present a regret bound of $\mathcal{O}(\sqrt{T})$; this newly shows that enforcing certain measure properties of SB generalize the classical OMD results (Srebro et al., 2011; Lei & Zhou, 2020).

**Theorem 2** (Regret bound). *Assume $\varphi, \psi \in C^2(\mathbb{R}^d) \cap \text{Lip}(\mathcal{K})$ and Assumption 3 in Appendix A holds with a constant $\omega > 0$. Define $D^2 = \max_{1 \leq t \leq T} D_\Omega(u \| \pi_t)$ for a total step $T$. (a) For a constant step size $\eta \equiv \frac{D\sqrt{\omega}}{\sqrt{2\mathcal{K}T}}$ the regret is bounded to $D\sqrt{2\omega^{-1}\mathcal{K}T}$. (b) For a heuristic scheduling $\eta_t = D\sqrt{\omega}/\sqrt{2\sum_t \|\hat{g}_t\|^2}$ the regret is bounded to $D\sqrt{2\omega^{-1}\sum_t \|\hat{g}_t\|^2}$ where $\hat{g}_t = \delta_c\Omega(\pi_t) - \delta_c\Omega(\pi_t^\circ)$.*

Fig. 5 shows our experiments for Gaussian mixture models (GMMs). Let a reference estimation be fitted using a Monte Carlo method, and our model be trained through an OMD method. We observed that the OMD method provides stability improvement when $\eta < 1$. The performance of OMD was greatly improved by choosing a harmonic step size scheduling in the interval $[1.0, 0.05]$.

### 4.3 ONLINE MIRROR DESCENT USING A WASSERSTEIN GRADIENT FLOW

For the computation, we adopt the Wasserstein gradient flow theory. Learning with Wasserstein gradient flows Eq. (9) is asymptotically stable due to the LaSalle's invariance principle (Carrillo et al., 2023). Suppose we expand a time step interval $[t, t+1)$ for OMD into continuous dynamics of $\rho(\tau) \in \mathcal{C}$ for $\tau \in [0, \infty)$. By Otto's calculus on the Wasserstein space (Otto, 2001), known as the Otto calculus, one can describe the gradient dynamics of minimizing a functional $\mathcal{E}_t(\cdot)$ by a PDE:

$$\partial_\tau \rho_\tau = -\nabla_{\text{w}} \mathcal{E}_t(\rho), \tag{8}$$

where $\nabla_{\text{w}}$ denotes the Wasserstein-2 gradient operator $\nabla_{\text{w}} := \nabla \cdot (\rho \nabla \frac{\delta}{\delta\rho})$. Recall that that the objective $F_t$ satisfies the 1-relative-smoothness and 1-strong-convexity relative to $\Omega$ (Aubin-Frankowski et al., 2022) (see Definition 6). Then, we can convert the MD update problem (10) into another problem with identical smoothness and convexity. We present the following theorem for computation.

**Theorem 3** (Dynamics equivalence in first variation). *Consider the Wasserstein gradient dynamics in PDE (8) governed by the convex problem of OMD updates (6). The gradient dynamics of updates are equivalent to that of a linear combination of KL functionals such that*

$$\eta_t \delta_c \mathcal{E}_t(\rho_\tau) = \delta_c \{\eta_t \text{KL}(\rho_\tau \| \vec{\pi}_t^\circ) + (1 - \eta_t)\text{KL}(\rho_\tau \| \pi_t)\} \quad \forall \rho_\tau \in \mathcal{C}, \tag{9}$$

*and the PDE (8) converges a unique equilibrium of subsequent OMD iterate of Eq. (6) as $\tau \to \infty$.*

*Sketch of Proof.* We identify $\delta \mathcal{E}_t$ as a dynamics that reaches an equilibrium solution for

$$\underset{\pi \in \mathcal{C}}{\text{minimize}} \left\langle \delta_c F_t(\pi_t), \pi - \pi_t \right\rangle + \frac{1}{\eta_t} D_\Omega(\pi \| \pi_t)$$

$$\iff \quad \underset{\pi \in \mathcal{C}}{\text{minimize}} \; \eta_t \underbrace{D_\Omega(\pi \| \pi_t^\circ)}_{\text{empirical estimates}} + (1 - \eta_t) \underbrace{D_\Omega(\pi \| \pi_t)}_{\text{proximity}}, \tag{10}$$

and then the equivalence of first variation for recursively defined Bregman divergences is applied (Lemma 4). At a glance, Eq. (10) appears analogous to the interpolation search between two points, where the influence of $\pi_t^\circ$ is controlled by $\eta_t$. We leave the entire proof in Appendix A.5. $\qquad \square$

Theorem 3 holds practical importance since following the argument allows us to perform MD without directly computing Bregman divergence. Therefore, we propose to perform updates with a linear combination of two KL functionals, where such gradient flows has been extensively studied both theoretically and computationally (Jordan et al., 1998; Lambert et al., 2022).

## 5 ALGORITHM: VARIATIONAL MIRRORED SCHRÖDINGER BRIDGE

In this section, we propose a simulation-free method that offers iterative MD updates for parameterized SB models with mixture models, using the Wasserstein-Fisher-Rao geometry.

### 5.1 GAUSSIAN MIXTURE PARAMETERIZATION FOR THE SCHRÖDINGER BRIDGE PROBLEM

Recently, Korotin et al. (2024) proposed the GMM parameterization, which provides theoretically and computationally desirable models for our variational OMD approach. The parameterization considers the *adjusted* Schrödinger potential $u^*(x) \coloneqq \exp(\varphi^*(x) - \|x\|^2/2\varepsilon)$ and $v^*(y) \coloneqq \exp(\psi^*(y) - \|y\|^2/2\varepsilon)$. With a finite set of parameters $\theta \triangleq \{\alpha_k, m_k, \Sigma_k\}_{k=1}^K$ for $\alpha_k > 0, m_k \in \mathbb{R}^d$ and $\Sigma_k \in \mathbf{S}_{++}^d$. The adjusted Schrödinger potential $v_\theta$ and conditional probability density $\vec{\pi}_\theta$ write

$$v_\theta(y) \coloneqq \sum_{k=1}^K \alpha_k \mathcal{N}(y | m_k, \varepsilon \Sigma_k), \qquad \vec{\pi}_\theta^x(y) \coloneqq \frac{1}{z_\theta^x} \sum_{k=1}^K \alpha_k^x \mathcal{N}(y | m_k^x, \varepsilon \Sigma_k), \tag{11}$$

where each parameter for $\vec{\pi}^x$ conditioned by an input $x$: $m_k^x \coloneqq m_k + \Sigma_k x$, $\alpha_k^x \coloneqq \alpha_k \exp\left(\frac{x^\top \Sigma_k x + \langle m_k, x \rangle}{2\varepsilon}\right)$, $z_\theta^x \coloneqq \sum_{k=1}^K \alpha_k^x$ (see Proposition 3.2 of Korotin et al.). For this parameterization, the closed-from expression of SB process $\mathcal{T}_\theta$ is given as the following SDE:

$$\mathcal{T}_\theta : \mathrm{d}X_t = g_\theta(t, X_t)\,\mathrm{d}t + \sqrt{\varepsilon}\,\mathrm{d}W_t, \qquad t \in [0, 1)$$

$$g_\theta(t, x) \coloneqq \varepsilon \nabla \log \mathcal{N}(x | 0, \varepsilon(1-t) I_d) \sum_{k=1}^K \alpha_k \mathcal{N}(m_k | 0, \varepsilon \Sigma_k) \mathcal{N}\big(m_k(t, x) \big| 0, A_k(t)\big), \tag{12}$$

where $m_k(t, x) \triangleq \frac{x}{\varepsilon(1-t)} + \frac{1}{\varepsilon} \Sigma_k^{-1} m_k$ and $A_k(t) \triangleq \frac{t}{\varepsilon(1-t)} I_d + \frac{1}{\varepsilon} \Sigma_k^{-1}$. Korotin et al. (2024) also presented theoretical properties for probabilistic inference and diffusion models, including universal approximation of $\vec{\pi}_\theta$ and $\mathcal{T}_\theta$. Furthermore, the GMM parameterization makes the computation of the Wasserstein gradient flow with respect to the KL divergence tractable, which is elaborated in § 5.2.

### 5.2 COMPUTATION OF VARIATIONAL MD IN THE WASSERSTEIN-FISHER-RAO GEOMETRY

**Wasserstein-Fisher-Rao.** The space of Gaussian parameters $\mathbb{R}^d \times \mathbf{S}_{++}^d$ equipped with $W_2$ is formally known as the Bures-Wasserstein (BW) geometry (Bures, 1969; Bhatia et al., 2019; Lambert et al., 2022) $\mathtt{BW}(\mathbb{R}^d) \subseteq \mathcal{P}_2(\mathbb{R}^d)$. On top of the BW space, the Wasserstein-Fisher-Rao geometry of GMMs, namely $\mathcal{P}_2(\mathtt{BW}(\mathbb{R}^d))$ provides *liftings* of Gaussian particles (Liero et al., 2018; Chizat et al., 2018; Lu et al., 2019; Lambert et al., 2022) satisfying the distributional property. We present the following proposition, which describes the WFR dynamics $\theta_\tau$ for the LightSB parameterization $\vec{\pi}_\theta^x$.

**Proposition 2** (WFR gradient dynamics). *Suppose a GMM $\rho_{\theta_\tau}$ with $\theta_\tau = \{\alpha_{k,\tau}, m_{k,\tau}, \Sigma_{k,\tau}\}_{k=1}^K$. Let $y_{k,\tau} \sim \mathcal{N}(m_{k,\tau}, \Sigma_{k,\tau})$ denote a sample from the k-th Gaussian particle of $\rho_{\theta_\tau}$. Then, the WFR*

*dynamics* $\nabla_{\text{WFR}}\text{KL}(\rho_{\theta_\tau}\|\rho^*)$ *wrt* $\dot{\theta}_\tau = \{\dot{\alpha}_{k,\tau}, \dot{m}_{k,\tau}, \dot{\Sigma}_{k,\tau}\}_{k=1}^K$ *are given as*

$$\dot{\alpha}_{k,\tau} = -\left(\mathbb{E}\left[\log\frac{\rho_{\theta_\tau}}{\rho^*}(y_{k,\tau})\right] - \frac{1}{z_\tau}\sum_{\ell=1}^K \alpha_\ell \mathbb{E}\left[\log\frac{\rho_{\theta_\tau}}{\rho^*}(y_{\ell,\tau})\right]\right)\alpha_{k,\tau}, \tag{13}$$

$$\dot{m}_{k,\tau} = -\mathbb{E}\left[\nabla\log\frac{\rho_{\theta_\tau}}{\rho^*}(y_{k,\tau})\right], \quad \dot{\Sigma}_{k,\tau} = -\mathbb{E}\left[\nabla^2\log\frac{\rho_{\theta_\tau}}{\rho^*}(y_{k,\tau})\right]\Sigma_{k,\tau} - \Sigma_{k,\tau}\mathbb{E}\left[\nabla^2\log\frac{\rho_{\theta_\tau}}{\rho^*}(y_{k,\tau})\right],$$

*for* $\tau \in [0,\infty)$, *where* $z_\tau := \sum_{k=1}^K \alpha_k$; $\nabla$ *and* $\nabla^2$ *denote gradient and Hessian with respect to* $y_{k,\tau}$.

Appendices A.6 and B contain the complete theory. Proposition 2 implies that the one parameter family $\theta_\tau$ predicts a gradient-based algorithm of $\nabla_{\text{WFR}}\text{KL}(\rho_{\theta_\tau}\|\rho^*)$, thus Eq. (13) can be directly used for training GMM models. Recall that GMMs have a closed-form expression of log-likelihoods, which means each likelihood difference can be driven without errors. Given that the target has the identical number of Gaussian particles, both Eq. (13) and its approximation using finite samples will strictly have zero gradients after the flow reaches a certain equilibrium. Hence, abiding WFR gradient dynamics will result in more stable outcomes than standard gradient-based learning.

**Algorithmic considerations.** We introduce SB parameters $\theta$ and $\phi$, which represents $\vec{\pi}_t$ and $\vec{\pi}_t^\circ$ from the theoretical framework in § 4.2, and $\vec{\pi}_\phi$ is independently fitted using an arbitrary data-driven SB solver, such as LightSB and its variants. Also, we introduce the following gradient operation

$$\texttt{WFRgrad}(\theta; \phi, x, n_y) \approx \nabla_{\text{WFR}}\text{KL}(\vec{\pi}_\theta\|\vec{\pi}_\phi) = \{\dot{\alpha}_k^x, \dot{m}_k^x, \dot{\Sigma}_k\}_{k=1}^K \text{ in Proposition 2.} \tag{14}$$

For the operator $\texttt{WFRgrad}$, the WFR gradient (13) is estimated using finite $n_y$ samples from each Gaussian particle of $\vec{\pi}_\theta$, expressed as $\{\boldsymbol{Y}_k^x\}_{k=1}^K \in \mathbb{R}^{k \times n_y}$. At each iteration $t$, we propose to update the SB model $\vec{\pi}_\theta$ with $\eta_t \texttt{WFRgrad}(\theta; \phi) + (1 - \eta_t)\texttt{WFRgrad}(\theta; \phi)$, as stated in Theorem 3.

---

**Algorithm 1** Variational Mirrored SB (VMSB).

---

**Input:** SB models $(\vec{\pi}_\theta, \vec{\pi}_\phi)$ parameterized by Gaussian mixtures, step sizes $(\eta_1, \eta_T)$.
1: **for** $t \leftarrow 1$ **to** $T$ **do**
2:    $\eta_t \leftarrow 1/\left(\eta_1^{-1} + (\eta_T^{-1} - \eta_1^{-1})(t^{-1}/T^{-1})\right)$
3:    **for** $n \leftarrow 1$ **to** $N$ **do**
4:       Update $\vec{\pi}_\phi$ with a data-driven SB solver.
5:       $\{x_i\}_{i=1}^B \leftarrow$ sample batch data from $\mu$.
6:       $\frac{\partial\mathcal{L}}{\partial\theta} \leftarrow \frac{1}{B}\sum_{i=1}^B \eta_t \texttt{WFRgrad}(\theta; \phi, x_i) + (1 - \eta_t)\texttt{WFRgrad}(\theta; \theta_{t-1}, x_i)$
7:       Update $\theta$ with the gradient $\frac{\partial\mathcal{L}}{\partial\theta}$.
8:    **end for**
9: **end for**
**Output:** Trained SB model $\vec{\pi}_\theta$.

---

We propose to gradually minimize the step size by a harmonic series for $1 \geq \eta_1 \geq \eta_T > 0$. According to Proposition 1, one can schedule of the step size $\eta_t$ with a harmonic progression. We set $\eta_1 = 1$ and $\eta_T \in \{0.05, 0.1\}$ which varies depending the total length of training. We can also put a few "warm up" steps for complex problems and start from $\theta = \phi$ after certain updates enforcing $\eta_t \equiv 1$ for the early training stage. For the distribution $\mu$, we set $x_i = 0$ and $B = 1$ only when $\mu$ is a zero-centered Gaussian distribution. This is equivalent to directly training the potential $v_\theta \propto \pi_\theta(\cdot|x = 0)$, and this tricks makes the algorithm run efficiently for certain generation problems. Algorithm 1 outlines the overall procedure.

# 6 EXPERIMENTAL RESULTS

**Experiment goals.** We delineate our objectives as follows: ① We aimed to affirm our online learning hypothesis by demonstrating consistent improvements. ② We sought to corroborate our

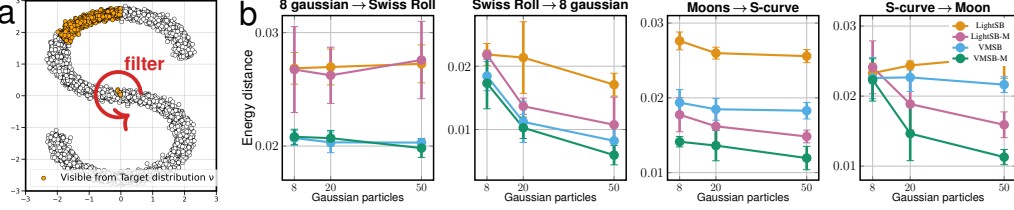

Figure 6: Online SBPs for synthetic dataset streams. (a) An online learning problem with a rotating filter. (b) The plots show that our VMSB and VMSB-M show consistent improvements from their references regarding the ED metric with 95% confidence intervals for 5 runs with different seeds.

Table 2: A summary of EOT benchmark scores with $cB\mathbb{W}_2^2$-UVP $\downarrow$ (%) between the optimal plan $\pi^*$ and the learned plan $\pi_\theta$ across five different seeds. We highlighted the VMSB results in bold when they exceed their reference algorithm. See Appendix E for more comprehensive statistics.

| Type | Solver | $\varepsilon = 0.1$ | | | | $\varepsilon = 1$ | | | | $\varepsilon = 10$ | | | |
|------|--------|-------|-------|-------|-------|-------|-------|-------|-------|-------|-------|-------|-------|
| | | $d=2$ | $d=16$ | $d=64$ | $d=128$ | $d=2$ | $d=16$ | $d=64$ | $d=128$ | $d=2$ | $d=16$ | $d=64$ | $d=128$ |
| Classical solvers (best; Korotin et al.)[†] | | 1.94 | 13.67 | 11.74 | 11.4 | 1.04 | 9.08 | 18.05 | 15.23 | 1.40 | 1.27 | 2.36 | 1.31 |
| rev. KL | LightSB (Korotin et al.) | 0.007 | 0.040 | 0.100 | 0.140 | 0.014 | 0.026 | 0.060 | 0.140 | 0.019 | 0.027 | 0.052 | 0.092 |
| Bridge-M | LightSB-M (Gushchin et al.) | 0.017 | 0.088 | 0.204 | 0.346 | 0.020 | 0.069 | 0.134 | 0.294 | 0.014 | 0.029 | 0.207 | 0.747 |
| Var-MD | VMSB (ours) | **0.004** | **0.012** | **0.038** | **0.101** | **0.010** | **0.018** | **0.044** | **0.114** | **0.013** | **0.019** | **0.021** | **0.040** |
| Var-MD | VMSB-M (ours) | **0.015** | **0.067** | **0.108** | **0.253** | **0.010** | **0.019** | **0.094** | **0.222** | **0.013** | **0.029** | **0.193** | 0.748 |

theoretical results, aiming for stable performance that consistently exceeds that of benchmarks. ③ We aimed to verify that our algorithm effectively induces OMD by the Wasserstein gradient flow.

**Baselines and VMSB variants.** Korotin et al. (2024) introduced a streamlined, simulation-free solver called LightSB that optimizes $\phi$ through Monte Carlo approximation of $\mathrm{KL}(\vec{\pi}^*\|\vec{\pi}_\phi)$. As an alternative, LightSB-M (Gushchin et al., 2024a) reformulated the reciprocal projection from DSBM (Shi et al., 2023) to a projection method termed *optimal projection*, establishing approximated bridge matching for the trajectory distribution $\mathcal{T}_\phi$. For the implementation of Algorithm 1, we derived two distinct methods called VMSB and VMSB-M ($\vec{\pi}_\theta$), trained upon LightSB and LightSB-M ($\vec{\pi}_\phi$), respectively. Since the theoretical arguments imply that the algorithm is agnostic to targets, the performance benefits of VMSB variants from their references support the generality of our claims.

## 6.1 STABILITY OF SB IN SYNTHETIC DATA STREAMS

To validate our online learning hypothesis, we considered 2D SBPs for data streams depicted in Fig. 6 (a). We applied an angle-based rotating filter, making the marginal as a data stream where only 12.5% (or 45-degree angle) of the total data is accessible for each step $t$. We trained conditional models $\vec{\pi}_\theta$ for ordinary SB for the 2D coordinates. Fig. 6 (b) shows the plots of squared energy distance (ED), which is a special instance of squared maximum mean discrepancy (MMD), approximating the $\mathrm{L}^2$ distance between distributions: $\mathrm{ED}(P,Q) \approx \int (P(x) - Q(x))^2 \mathrm{d}x$ (Rizzo & Székely, 2016). In our ED evaluation, the MD algorithm achieved a strictly lower divergence than the LightSB and LightSB-M solvers for various numbers of Gaussian particles $K$. Therefore, we concluded that these results aligned with our hypothesis and theory of online mirror descent.

## 6.2 QUANTITATIVE EVALUATION ON THE EOT BENCHMARK

Next, we considered the EOT benchmark proposed by Gushchin et al. (2024b), which contains 12 entropic OT problems with different volatility and dimensionality settings. Table 2 shows that among 24 different settings, our MD approach exceeded the reference model in 23 settings in terms of the $cB\mathbb{W}_2^2$-UVP metric (Gushchin et al., 2024b). From our replication of LightSB/LightSB-M, which achieved better performance than originally reported results. As a result, our method reached the state-of-the-art performance in this benchmark with stability, which represents strong evidence of Proposition 1. Among all cases, the only exception was LightSB-M, which had the highest dimension and volatility. We suspected that the drift form Eq. (12), which is proportional to $\varepsilon$, might have violated our assumptions Assumption 1 and the boundedness assumption during the training. Thus, we conclude that our variational MD training is effective in various setups.

## 6.3 SB ON BIOLOGICAL DATA

We also evaluated VMSB on unpaired single-cell data problems in the high-dimensional single-cell experiment (Tong et al., 2023). The MSCI dataset provided single-cell data from four donors on days 2, 3, 4, and 7, describing the gene expression levels of distinct cells. Given samples collected on two different dates, the task involves performing inference on temporal evolution, such as interpolation and extrapolation of

Table 3: Energy distance on the MSCI dataset (95% confidence interval, ten trials with different instances). Results marked with ‡ are from (Gushchin et al., 2024a).

| Type | Solver | $d=50$ | $d=100$ | $d=1000$ |
|------|--------|--------|---------|----------|
| Sinkhorn | Vargas et al. (2021)[†] | 2.34 | 2.24 | 1.864 |
| Bridge-M | DSBM (Shi et al.)[‡] | $2.46 \pm 0.1$ | $2.35 \pm 0.1$ | $1.36 \pm 0.04$ |
| Bridge-M | SF$^2$M-Sink (Tong et al.)[‡] | $2.66 \pm 0.18$ | $2.52 \pm 0.17$ | $1.38 \pm 0.05$ |
| rev. KL | LightSB | $2.31 \pm 0.08$ | $2.15 \pm 0.09$ | $1.264 \pm 0.06$ |
| Bridge-M | LightSB-M | $2.30 \pm 0.08$ | $2.15 \pm 0.08$ | $1.267 \pm 0.06$ |
| Var-MD | VMSB (ours) | $\mathbf{2.28 \pm 0.09}$ | $\mathbf{2.13 \pm 0.09}$ | $\mathbf{1.260 \pm 0.06}$ |
| Var-MD | VMSB-M (ours) | $\mathbf{2.26 \pm 0.10}$ | $\mathbf{2.12 \pm 0.09}$ | $1.265 \pm 0.05$ |

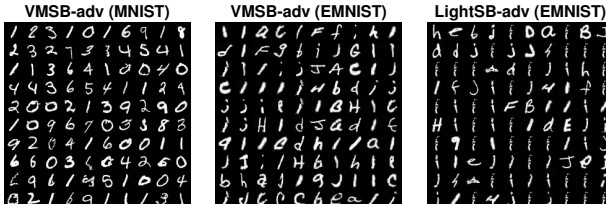

Figure 7: Generated MINST/EMNIST translation samples.

Table 4: FID and MSD similarity scores in EMNIST-to-MNIST.

| | Method | FID | MSD |
|---|---|---|---|
| U-net | SF²M-Sink | 23.215 | 0.456 |
| | DSBM-IPF | 15.211 | 0.352 |
| | DSBM-IMF | 11.429 | 0.373 |
| GMM | LightSB-adv | 20.017 | 0.362 |
| | VMSB-adv | 15.471 | 0.356 |

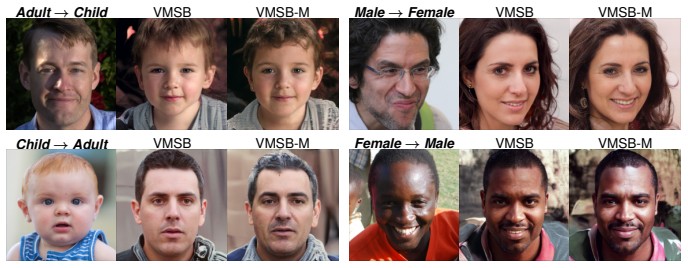
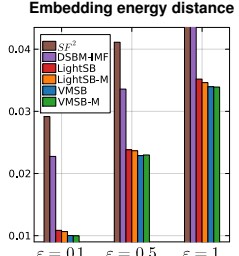

Figure 8: Image-to-Image translation on a latent space. Left: Generation results for the FFHQ dataset ($1024 \times 1024$) using our two SB variants. Right: Quantitative results using MMD metrics.

PCA projections with $\{50, 100, 1000\}$ dimensions. Table 3 shows that our VMSB method achieved the best results, verifying that VMSB is well-suited for the real-world EOT problems.

### 6.4 INTERACTING WITH NETWORKS: UNPAIRED IMAGE-TO-IMAGE TRANSFER TASKS

**Adversarial learning.** We applied VMSB to unpaired image translation tasks. LightSB methods struggled to generate raw pixels for the MNIST and EMNIST datasets. As our analysis did not specify a training algorithm for the target $\{\pi_t^\circ\}_{t=1}^\infty$, we opted to find a viable alternative, and we discovered that extending the capabilities of GMM parameterization by incorporating learning dynamics with an adversarial learning technique (Goodfellow et al., 2014; see Appendix C.5) was effective in providing rich learning signals. Therefore, we named the adversarial method and the VMSB adaptation LightSB-adv and VMSB-adv. Fig. 7 shows that VMSB-adv outperformed LightSB-adv (with identical architecture) in the quality of samples, efficiently mitigating mode-collapsing (Salimans et al., 2016). In Table 4, VMSB also achieved competitive FID and input/output MSD similarity scores for $K = 4096$, comparable to deep SB models with a smaller number of parameters.

**Latent diffusion bridge.** Following the latent diffusion bridge practice of (Korotin et al., 2024), we assessed our method by utilizing the ALAE model (Pidhorskyi et al., 2020) for generating $1024 \times 1024$ images of the FFHQ dataset (Karras et al., 2019). With the predefined 512-dimensional embedding space, we trained our SB models on the latent space to solve four distinct tasks: *Adult → Child*, *Child → Adult*, *Female → Male*, and *Male → Female*. Fig. 8 illustrates that our method delivered high-quality translation results. We also conducted a quantitative analysis using the ED on the ALAE embedding as a metric for evaluation. The result also verifies that our VMSB algorithm consistently achieved lower ED scores, demonstrating its applicability for pretrained latent spaces. Consequently, adversarial learning and latent diffusion applications showed that the proposed algorithm is highly capable of interacting with neural networks of complex architectures.

## 7 CONCLUSION

In this paper, we have presented an OMD framework developed to solve SBPs with robustness. Our geometric interpretation of the dual space allowed us to construct a robust OMD algorithm with theoretical guarantees for convergence and regrets. We substantially reduced the computational challenge in the MD framework using the WFR geometry. The proposed method demonstrated stable benchmark performance, exhibiting enhanced stability. We argue that the VMSB algorithm offers a promising approach for solving probabilistic generative modeling in the context of learning theory. The limitations and potential directions for future research are thoroughly discussed in Appendix D.

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

---

Appendices for

## Variational Mirror Descent for Robust Learning in Schrödinger Bridge

---

## ABBREVIATION AND NOTATION

| Abbreviation | Expansion | | Notation | Usage |
|---|---|---|---|---|
| SB | Schrödinger Bridge | | $\mu, \nu$ | marginal distributions |
| SBP | Schrödinger Bridge Problem | | $\varepsilon$ | volatility of reference measure |
| EOT | Entropy-regularized Optimal Transport | | $c_\varepsilon$ | cost $c_\varepsilon(x, y) \coloneqq \frac{1}{2\varepsilon}\|x - y\|^2$ |
| MD | Mirror Descent | | $\pi$ | a coupling of $\mu$ and $\nu$ |
| OMD | Online Mirror Descent | | $\vec{\pi}, \overleftarrow{\pi}$ | conditional distributions |
| KL | Kullback-Leibler | | $\gamma_n$ | $n$-th marginal |
| IPF | Iterative Proportional Fitting | | $\varphi, \psi$ | log-Schrödinger potential |
| BW | Bures-Wasserstein | | $u, v$ | adjusted Schrödinger potential |
| WFR | Wasserstein-Fisher-Rao | | $\Omega, D_\Omega$ | Bregman potential/divergence |
| SDE | Stochastic Differential Equation | | $d^+$ | directional derivative |
| PDE | Partial Differential Equation | | $\delta_c, \delta_{\mathcal{D}}$ | First variations |
| FP | Fokker–Planck | | $\nabla_{\mathrm{w}}$ | Wasserstein-2 gradient operator |
| GMM | Gaussian mixture model | | $\mathcal{T}$ | dynamic stochastic process in SB |
| | | | $g$ | drift function |
| | | | $i_{\mathcal{C}}$ | indicator function |

## A  THEORETICAL DETAILS AND PROOFS

In this appendix, we first introduce an comprehensive theoretical background supporting our arguments. Then, we provide the formal proofs in the main paper.

**Background on first variation operators.** We utilize the notations $\delta_c$ and $\delta_{\mathcal{D}}$ to denote the first variation operators in generalized primal and dual spaces, respectively. This is because SB is classified as an infinite-dimensional optimization problem (Aliprantis & Border, 2006). The theoretical necessity of these operators follows the discussion provided by Aubin-Frankowski et al. (2022).

**Definition 4** (Gâteaux and Fréchet differentiablility). Let $\mathcal{M}$ be a topological vector space of measures on the space $\mathcal{X}$. Define the Gâteaux differentiablity of a functional $F$, if there exists a gradient operator $\nabla_{\mathrm{Gât}}$ such that for any direction $v \in \mathcal{M}$, defined as the limit

$$\nabla_{\mathrm{Gât}} F(x)[v] = \lim_{h \to 0} \frac{F(x + hv) - F(x)}{h}, \quad x \in \mathcal{M}$$

If the limit exists in the unit ball in $\mathcal{M}$, the function $F$ is called Fréchet differentiable with $\nabla_{\mathrm{Fré}} F(x)$.

The problem of the Gâteaux and Fréchet differentiability in the context of SB is that the limit must be given in *all* directions, implying that every neighboring point must be within the domain $\mathcal{M}$. For the case of functionals such as the KL divergence functional $F(\cdot) = \mathrm{KL}(\cdot|\pi^*)$, the domain of $F$ and has an empty interior (Aubin-Frankowski et al., 2022). To resolve this issue, we can use the notion of *directional derivative* and *first variation*, defined in Definitions 1 and 2.

**First variations of KL.** Suppose that for distribution $\rho, \rho' \in \mathcal{P}_2(\mathcal{X}), \mathcal{X} \subseteq \mathbb{R}^d$, and define a function $\ell'(x) \coloneqq \log \rho'(x)$, and suppose $\kappa$ in a tangent space $T_\rho \mathcal{P}(\mathcal{X})$. We can achieve the followings:

$$\mathrm{KL}(\rho \| \rho') = \int_{\mathcal{X}} \log \rho(x) \, \mathrm{d}\rho(x) - \int_{\mathcal{X}} \ell'(x) \, \mathrm{d}\rho(x) \tag{15}$$

$$\int \ell'(x)[\rho(x) + h\kappa(x)] \, \mathrm{d}x = \int \ell'(x)\rho(x) \, \mathrm{d}x + h \int \ell'(x)\kappa(x) \, \mathrm{d}x \tag{16}$$

Given that $\log(z + \varepsilon)(z + \varepsilon) = \log(z)z + [\log(z) + 1]\varepsilon + o(\varepsilon)$, and $\int_{\mathcal{X}} \kappa(x)\,\mathrm{d}x = 0$, we achieve

$$
\begin{aligned}
\int_{\mathcal{X}} &\log\big(\rho(x) + h\kappa(x)\big)\big(\rho(x) + h\kappa(x)\big)\,\mathrm{d}x \\
&= \int_{\mathcal{X}} \log\rho(x)\rho(x) + [\log\rho(x) + 1]h\kappa(x) + o(h)\,\mathrm{d}x \\
&= \int_{\mathcal{X}} \log\rho(x)\rho(x)\mathrm{d}x + h\int_{\mathcal{S}}\log\rho(x)\kappa(x)\mathrm{d}x + h\int\kappa(x)\mathrm{d}x + o(h)
\end{aligned}
\tag{17}
$$

Recall that a first variation of a functional $\delta F : \mathcal{P}(\mathcal{X}) \to T^*\mathcal{P}(\mathcal{X})$ satisfies:

$$
F(\rho + h\kappa) = F(\rho) + h\left\langle \log\left(\frac{\rho}{\rho'}\right),\ \kappa \right\rangle + o(h).
$$

We leave the following remark for the first variation operator works in KL functionals.

**Remark 1.** Combining Eqs. (15 -17), the first variation of the functional $\delta\mathrm{KL}(\rho\|\rho') = \log\frac{\rho}{\rho'}$.

For some distributions, log-likelihoods are often given in a closed-form expression, incentivizing our development of computational continuous EOT/SB algorithms. Generally, identical arguments generally apply to all KL functionals with respect to distributions ($\pi$, $\vec{\pi}$, and marginals) in our setup.

**Asymptotically log-concave distributions.** For convergence analysis, we assume each marginal distribution is in log-concave distribution, particularly satisfying the log Sobolev inequality (Otto & Villani, 2000; Conforti, 2024). This assumption works a wider range of costs and marginals beyond popular choices bounded costs and compact marginals (Nutz & Wiesel, 2023; Conforti et al., 2023). Suppose that marginals admit densities of the form

$$
\mu(\mathrm{d}x) = \exp(-U_\mu(x))\mathrm{d}x \qquad \text{and} \qquad \nu(\mathrm{d}y) = \exp(-U_\nu(y))\mathrm{d}y.
\tag{18}
$$

We exploit the following definition from (Conforti et al., 2023) in order to describe asymptotically log-concaveness.

**Definition 5** (Asymptotically strongly log-concavity). We assume that marginals $\mu$ and $\nu$ admit a positive density against the Lebesgue measure, which can be written in the form (18). $U_\mu, U_\nu$ are of class $C^2(\mathbb{R}^d)$. Define a set $\mathcal{G} := \{g \in C^2((0, +\infty), \mathbb{R}_+)|r \mapsto r^{1/2}g(r^{1/2})$is non-increasing and concave, $\lim_{r\to 0} rg(r) = 0\}$.

$$
\tilde{\mathcal{G}} := \{g \in \mathcal{G} \text{ bounded and s.t. } \lim_{r\to 0^+} g(r) = 0,\ g' \geq 0 \quad \text{and} \quad 2g'' + gg' \leq 0\} \subset \mathcal{G}.
$$

Define *convexity profile* $\kappa_U : \mathbb{R}_+ \to \mathbb{R}$ of a differentiable function $U$ as the following

$$
\kappa_U(r) := \left\{ \frac{\langle \nabla U(x) - \nabla U(y), x - y \rangle}{|x - y|^2}\ :\ |x - y| = r \right\}.
$$

We say a potential is asymptotically strongly convex if there exists $\alpha_U \in \mathbb{R}_+$ and $\tilde{g}_U \in \tilde{\mathcal{G}}$ such that

$$
\kappa_U(r) \geq \alpha_U - r^{-1}\tilde{g}_U(r)
$$

holds for all $r > 0$. We consider the set of asymptotically strongly log-concave probability measures

$$
\mathcal{P}_{\mathrm{alc}}(\mathbb{R}^d) := \{\zeta(\mathrm{d}x) = \exp(-U(x))\mathrm{d}x : U \in C_2(\mathbb{R}^d),\ U \text{ is asymptotically strongly convex}\}.
$$

From the work of (Otto & Villani, 2000; Conforti et al., 2023), asymptotically log-concave functions satisfy a certain form of log Sobolev inequality (Gross, 1975). The simplest case of LSI for the Gaussian measure is represented as follows.

**Remark 2** (log-Sobolev inequality for the standard Gaussian). Suppose that $f$ is a nonnegative function, integrable with respect to a measure $\gamma$, and that the entropy is defined as $\mathrm{Ent}_\gamma(f) = \int_{\mathbb{R}^d} f\log f\mathrm{d}\gamma - \left(\int_{\mathbb{R}^d} f\mathrm{d}\gamma\right)\log\left(\int_{\mathbb{R}^d} f\mathrm{d}\gamma\right)$. the logarithmic Sobolev inequality when $\gamma$ is the standard Gaussian measure reads $\mathrm{Ent}_\gamma(f) \leq \frac{1}{2}\int_{\mathbb{R}^d}\frac{|f|^2}{f}\mathrm{d}\gamma$.

The important extension of asymptotically strong log-concave distributions for Schrödinger bridge $\mathrm{d}\pi = e^{\varphi\oplus\psi-c_\varepsilon}\mathrm{d}(\mu\otimes\nu)$, $(\mu\otimes\nu)$-a.s. is that induced SB model also satisfies asymptotically strongly log-concaveness and the log Sobolev inequality (Conforti, 2024). Therefore, the Gaussian mixture parameterization in Eq. (11) is a representative model that our theoretical analysis is dealing with.

**Remark 3** (Conforti, 2024). Let $\mu, \nu \in \mathcal{P}_{\text{alc}}(\mathbb{R}^d)$ with finite entropy on a Lebesgue measure and $\pi \in \mathcal{C}$ be a coupling in a static Schrödinger bridge problem. Then, for a quadratic cost function, the coupling distribution is also asymptotically log-concave and satisfies a form of logarithmic Sobolev inequality.

Using the disintegration theorem for probability measures (Léonard, 2014), we assume the boundedness of Bregman divergence between two transport plans using derivatives of first variations with some positive constraint $\omega > 0$ by the following assumption.

**Assumption 3** (LSI for EOT couplings). Let us suppose $\Omega = \text{KL}(\pi \| \mathcal{R})$ for a reference measure $\mathcal{R}$. We assume arbitrary $\pi, \bar{\pi} \in \mathcal{C}$ satisfy a type of logarithmic Sobolev inequality for relative entropy (KL divergence) is upper bounded by (relative) Fisher information (Gross, 1975), namely $\text{LSI}(\omega)$ for some $\bar{\omega} \in \mathbb{R}_+$ as follows.

$$D_\Omega(\pi \| \mathcal{R}) = \text{KL}(\pi \| \mathcal{R}) \leq \frac{1}{2\bar{\omega}} \iint_{\mathbb{R}^d \times \mathbb{R}^d} \left| \nabla \log \frac{\pi(x,y)}{\mathcal{R}(x,y)} \right|^2 \pi(\mathrm{d}x, \mathrm{d}y)$$

where $\Omega = \text{KL}(\cdot \| \mathcal{R})$. By the first variation of KL (Remark 1), equivalence in the first variation of Bregman divergences (explained later in Lemma 4) and an application of the Hölder's inequality, assume that we can find a constant $\omega > 0$ such that that

$$D_\Omega(\pi \| \bar{\pi}) \leq \frac{1}{2\omega} \left\| \nabla (\delta_c \Omega(\pi) - \delta_c \Omega(\bar{\pi})) \right\|^2_{L^2(\pi)} \tag{19}$$

for the Bregman potential $\Omega = \text{KL}(\cdot \| e^{-c_\varepsilon} \mu \otimes \nu)$.

In general, the log-Sobolev inequality has often been used to analyze the convergence of partial differential equations (Malrieu, 2001). In the same vein, to make an analysis on improvement (Lemma 12) and a solid regret bound of OMD (Lemma 14), we found that Assumption 3 is necessary to ensure a certain asymptotical concentration of measure.

**General assumptions and justifications.** We need the following assumptions for our OMD framework. ① (Existence) The sequence of MD from Eq. (6) exists $\{\pi_t\}_{t \in \mathbb{N}} \subset \mathcal{C}$, and are unique, ② (Relative smoothness/convexity) For some $l, L \geq 0$, the functional $F_t$ is $L$-smooth and $l$-strongly-convex relative to $\Omega$. ③ (Existence of first variations) For each $t \geq 0$, the first variation $\delta_c \Omega(\pi_t)$ exists. ④ (Boundedness of estimations) The asymptotic dual mean $\pi_{\mathcal{D}}^\circ$ is almost surely bounded $\Pr(D_\Omega(\pi_t \| \pi_{\mathcal{D}}^\circ) \leq R) = 1$ for some $R > 0$. ⑤ (Ergodicity) The estimation process of $\{\pi_t^\circ\}_{t=1}^\infty$ is governed by a measure-preserving transformation on a measure space $(\mathcal{Y}, \Sigma, \varsigma)$ with $\varsigma(\mathcal{Y}) = 1$; for every event $E \in \Sigma$, $\varsigma(T^{-1}(E) \Delta E) = 0$ (that is, $E$ is invariant), either $\varsigma(E) = 0$ or $\varsigma(E) = 1$.[2] For ①, the temporal cost $F_t(\cdot) = \text{KL}(\cdot | \pi_t^\circ)$ is well defined since KL is a strong Bregman divergence with lower semicontinuity, where the existence of a primal solution in guaranteed as discussed in Aubin-Frankowski et al. (2022). For ②-③, we can identify $l = L = 1$ and close-form expression of the first variation that is shown in Definition 6 and Proposition 2. For the assumptions ④-⑤, we postulate the existence of estimates produced from a Monte-Carlo method, using a fixed amount of updates on topological vector space. Hence, it is natural to consider that these estimates will be bounded in a probabilistic sense and yield Markovian transitions, which are aperiodic and irreducible.

## A.1 Proofs of Lemmas 1 and 2

The EOT in Eq. (2) can be reformulated as a divergence minimization problem with respective to a reference parameterization. If a Gibbs parameterization is enforced for the quadratic cost functional $c_\varepsilon(x,y) = \frac{1}{2\varepsilon} \|x - y\|^2$ for $\varepsilon > 0$, the problem has the equivalence (Nutz, 2021)

$$\text{OT}_\varepsilon(\mu, \nu) := \min_{\pi \in \Pi(\mu,\nu)} \text{KL}(\pi \| e^{-c_\varepsilon} \mu \otimes \nu), \tag{20}$$

which corresponds $\text{KL}(\mathcal{T} \| W^\varepsilon)$ in Eq. (4) by the disintegration theorem of Schrödinger bridge (Appendix A of Vargas et al. (2021)). While the Bregman projection formulation of Sinkhorn Eq. (7) are described by the spaces $(\Pi_\mu^\perp, \Pi_\nu^\perp)$, it is (equally) natural to think that considering the problem as convex problem with the distributional constraint $\mathcal{C}$ (see the primal space in illustrated in

---

[2]Here, $\Delta$ denotes the symmetric difference, equivalent to the exclusive-or with respect to set membership.

Fig. 3). As a problem in $\mathcal{C}$, one can consider a temporal cost functional $\widetilde{F}_t(\pi) := a_t \mathrm{KL}(\gamma_1 \pi \| \mu) + (1 - a_t)\mathrm{KL}(\gamma_2 \pi \| \nu)$ with sequences $\{a_t\}_{t=1}^{\infty} = \{0, 1, 0, 1, \dots\}$ for $\gamma_1 \pi(x) := \int \pi(x, y)\mathrm{d}y$ and $\gamma_2 \pi(y) := \int \pi(x, y)\mathrm{d}x$. By construction, we have the following MD update:

$$\underset{\pi \in \mathcal{C}}{\text{minimize}} \langle \widetilde{F}_t, \pi - \pi_t \rangle + D_\Omega(\pi \| \pi_t). \tag{21}$$

The optimization problem (21) is equivalent to having the property for subsequent $\pi_{t+1}$:

$$d^+ \widetilde{F}_t(\pi_t; \pi - \pi_t) + D_\Omega(\pi \| \pi_t) \geq d^+ \widetilde{F}_t(\pi_t; \pi_{t+1} - \pi_t) + D_\Omega(\pi_{t+1} | \pi_t)$$
$$\iff \langle \delta_c \widetilde{F}_t(\pi_t) - \delta_c \Omega(\pi_t), \pi - \pi_{t+1} \rangle + (\Omega(\pi) - \Omega(\pi_{t+1})) \geq 0, \quad \forall \pi \in \mathcal{C}. \tag{22}$$

Setting the free parameter $\pi = \pi_{t+1} + h(\pi - \pi_{t+1})$ and taking the limit $h \to 0^+$ yields described the time evolution of the log-Schrödinger potentials for $\pi_t = e^{\varphi_t \oplus \psi_t - c_\varepsilon} \mathrm{d}(\mu \otimes \nu)$:

$$\dot{\varphi}_t = -\log \frac{\mathrm{d}(\gamma_1 \pi_t)}{\mathrm{d}\nu_*} = -\alpha \left( \varphi_t - \varphi^* + \log \int_{\mathbb{R}^d} e^{\psi_t - \psi^*} \nu(\mathrm{d}y) \right), \tag{23a}$$

$$\dot{\psi}_t = -\log \frac{\mathrm{d}(\gamma_2 \pi_t)}{\mathrm{d}\mu_*} = -\beta \left( \psi_t - \psi^* + \log \int_{\mathbb{R}^d} e^{\varphi_t - \varphi^*} \mu(\mathrm{d}x) \right), \tag{23b}$$

for $\alpha = a_t$ and $\beta = 1 - a_t$.[3] Setting a discrete approximation of dynamics Eq. (23): $\varphi_{t+1} = \varphi_t + \dot{\varphi}_t$ and $\psi_{t+1} = \psi_t + \dot{\psi}_t$ yields the following alternating updates:

$$\psi_{2t+1}(y) = -\log \int_{\mathbb{R}^d} e^{\varphi_{2t}(x) - c_\varepsilon(x,y)} \mu(\mathrm{d}x), \quad \varphi_{2t+2}(x) = -\log \int_{\mathbb{R}^d} e^{\psi_{2t+1}(x) - c_\varepsilon(x,y)} \nu(\mathrm{d}y).$$

Therefore, the proof of Lemma 1 is complete.

From the dual iteration of KL stated in Eq. (34), for the static cost $\mathrm{KL}(\cdot \| \pi^*)$, we get the closed-form expression:

$$\delta_c \Omega(\pi_t) - \delta_c \Omega(\pi_{t+1}) = \eta_t (\delta_c \Omega(\pi_t) - \delta_c \Omega(\pi^*)),$$

where the equation implies that setting $\eta_t \equiv 1$ for MD yields one-step optimality $\pi^*$ in this idealized condition. Utilizing the equivalence of first variation stated in Lemma 4 and the disintegration theorem for the Radon-Nikodym derivatives, we get the first variation of $F$ with respect to $\pi$ for all $x$ as

$$\delta F(\pi) = \log \frac{\mathrm{d}\pi^*}{\mathrm{d}\pi},$$

and by the disintegration theorem (Léonard, 2014), we achieve the first variation of $f$ with respect to $\vec{\pi}$ for all $x$ as

$$\delta f(\vec{\pi}^x) = \log \frac{\mathrm{d}(\vec{\pi}^*)^x}{\mathrm{d}\vec{\pi}^x}. \tag{24}$$

Using Otto's formalization of Riemannian calculus (Otto, 2001) discussed in Appendix B, the probability space equipped with the Wasserstein-2 metric $(\mathcal{P}_2(\mathbb{R}^d), W_2)$, is represented as Riemannian gradient flow:

$$\partial_t \vec{\pi}_t^x = -\nabla_{\mathbb{w}} f(\vec{\pi}_t^x), \forall x \in \mathbb{R}^d \tag{25}$$

where $\nabla_{\mathbb{w}}$ denotes the Wasserstein-2 gradient operator $\nabla_{\mathbb{w}} := \nabla \cdot \left( \rho \nabla \frac{\delta}{\delta \rho} \right)$.

$$\partial_t \vec{\pi}_t^x = -\nabla \cdot (\vec{\pi}^x \nabla \log(\vec{\pi}^*)^x) + \Delta \vec{\pi}_t^x,$$

where the results on Wasserstein gradients are initially founded by Jordan et al. (1998). Since the above equation represent the Fokker–Planck equation, following the Wasserstein gradients always operate within $\mathcal{C}$. □

## A.2 PROOF OF THEOREM 1

We start with the following idempotence property that taking a Bregman divergence associated with a Bregman divergence $D_\Omega(\cdot | y)$ remains as the identical divergence. We use $\mathcal{M}(\mathcal{X})$ to denote a topological vector space (Aliprantis & Border, 2006) for $\mathcal{X} \subseteq \mathbb{R}^d$.

---

[3]More precisely, one needs to apply Lemma 4 for KL, and the disintegration theorem to get Eq. (23).

**Lemma 3** (Idempotence). *Suppose a convex functional* $\Omega : \mathcal{M}(\mathcal{X}) \to \mathbb{R} \cup \{+\infty\}$, *where* $\mathcal{M}(\mathcal{X})$. *Assume that for all* $z \in \mathrm{dom}(\Omega)$, $\delta_c\Omega(z)$ *exists, then, for all* $x, y \in \mathcal{C} \cap \mathrm{dom}(\Omega)$, $D_{D_{\Omega}(\cdot|y)}(x|y) = D_\Omega(x|y)$.

*Proof of Lemma 3.* By definition, we have $D_{D_\Omega(\cdot|z)}(x|y) = D_\Omega(x\|z) - D_\Omega(y\|z) - \langle \delta_c\Omega(y) - \delta_c\Omega(z), x - y \rangle$ for arbitrary $z$, and setting $z = y$ completes the proof. Note that instead of the (global or universal) idempotence initially stated by Aubin-Frankowski et al. (2022), we only work with localized version of idempotence at the minima $y$. Another (informal) point of view is considering the Bregman divergence as a first-order approximation of a Hessian structure, and $D_{D_\Omega(\cdot|z)}$ converges to $D_\Omega(\cdot|z)$ by taking a limit, knowing that $D_\Omega(y|y) = 0$. □

We then proceed to an equivalence property of the family of recursive Bregman divergences.

**Lemma 4** (Equivalence of first variation). *Suppose* $\Omega : \mathcal{M}(\mathcal{X}) \to \mathbb{R} \cup \{+\infty\}$ *Assume that for all* $z \in \mathrm{dom}(\Omega)$, *the first variation* $\delta_c\Omega(z)$ *exists, then, for all* $x, y, y_1, y_2 \in \mathrm{dom}(\Omega)$, *the first variation taken for the first argument* $x$ *of the following Bregman divergences are equivalent:* $\delta_c D_\Omega(x|y) = \delta_c D_{D_\Omega(\cdot|y_1)}(x|y) = \delta_c D_{D_\Omega(\cdot|y_2)}(x|y)$.

*Proof of Lemma 4.* First, it can be analytically driven $\delta_c D_\Omega(x|y) = \delta_c\Omega(x) - \delta_c\Omega(y)$. Next, by definition, taking the first variation of $D_{D_\Omega(\cdot|z)}(x|y)$ with respect to $x$ for arbitrary $z \in \mathrm{dom}(\Omega)$ yields $\delta_c D_\Omega(x\|z) - \delta_c\langle\Omega(y) - \Omega(z), x - y\rangle$. Knowing that the second term $\delta_c\langle\Omega(y) - \Omega(z), x - y\rangle$ is linear, we achieve $\delta D_{D_\Omega(\cdot|z)}(x|y) = \delta_c\Omega(x) - \delta_c\Omega(z) - (\delta_c\Omega(y) - \delta_c\Omega(z)) = \delta_c\Omega(x) - \delta_c\Omega(y)$, which completes the proof. □

By an inductive reasoning, we arrive at the basic characterization of family of Bregman divergence in Definition 3, that all divergence recursively defined by $\Omega$, has the (local) idempotence and the (global) equivalence of first variation.

We introduce the notions of relative smoothness and convexity wrt a Bregman potential $\Omega$.

**Definition 6** (Relative smoothness and convexity). Let $G : \mathcal{M}(\mathcal{X}) \to \mathbb{R} \cup \{+\infty\}$ be a proper convex functional. Given scalar $l, L \geq 0$, we define that $G$ is $L$-smooth and $l$-strongly-convex relative to $\Omega$ over $\mathcal{C}$ if for every $x, y \in \mathrm{dom}(G) \cap \mathrm{dom}(\Omega) \cap \mathcal{C}$, we have

$$D_G(x\|y) \leq L D_\Omega(x\|y), \qquad D_G(x\|y) \geq l D_\Omega(x\|y),$$

respectively, where $D_G$ and $D_G$ are Bregman divergences associated with $G$ defined in Definition 3.

Due to the idempotence lemma, we immediately recognize that the Bregman divergence $D_\Omega$ is relatively 1-smooth and 1-strongly-convex for $\Omega$.

To start our analysis we reintroduce the well-known three-point identity for a Bregman divergence.

**Lemma 5** (Three-point identity). *For all* $\pi_a, \pi_b, \pi_c \in \mathcal{C} \cap \mathrm{dom}(\Omega)$, *we have the following identity*

$$\langle \delta_c\Omega(\pi_a) - \delta_c\Omega(\pi_b), \pi_c - \pi_b \rangle = D_\Omega(\pi_c\|\pi_b) - D_\Omega(\pi_c\|\pi_a) + D_\Omega(\pi_b\|\pi_a)$$

*when* $D_\Omega$ *is the Bregman divergence defined in Definition 3.*

*Proof of Lemma 5.* By the definition of Bregman divergence, we have

$$\begin{aligned}
D_\Omega(\pi_c\|\pi_b) - D_\Omega(\pi_c\|\pi_a) + D_\Omega(\pi_b\|\pi_a) &= \Omega(\pi_c) - \Omega(\pi_b) - \langle \delta\Omega(\pi_b), \pi_c - \pi_b \rangle \\
&\quad - \Omega(\pi_c) + \Omega(\pi_a) + \langle \delta_c\Omega(\pi_a), \pi_c - \pi_a \rangle \\
&\quad + \Omega(\pi_b) - \Omega(\pi_a) - \langle \delta_c\Omega(\pi_a), \pi_b - \pi_a \rangle \\
&= \langle \delta_c\Omega(\pi_a) - \delta_c\Omega(\pi_b), \pi_c - \pi_b \rangle.
\end{aligned}$$

Therefore, the proof is complete. □

Utilizing Lemma 5, we present the following useful lemmas for dealing inequalities regarding improvements (Han et al., 2022), which we call "Bregman differences."

**Lemma 6** (Left Bregman difference). *For all $\pi_a, \pi_b, \pi_c \in \mathcal{C} \cap \mathrm{dom}(\Omega)$, the following identity holds.*

$$D_\Omega(\pi_b \| \pi_a) - D_\Omega(\pi_c \| \pi_a) = -\langle \delta_c \Omega(\pi_c) - \delta_c \Omega(\pi_a), \pi_c - \pi_b \rangle + D_\Omega(\pi_b \| \pi_c). \qquad (26)$$

*Proof of Lemma 6.* Using Lemma 5, we have

$$D_\Omega(\pi_b \| \pi_a) - D_\Omega(\pi_c \| \pi_a) = -D_\Omega(\pi_c \| \pi_b) + \langle \delta_c \Omega(\pi_a) - \delta_c \Omega(\pi_b), \pi_c - \pi_b \rangle.$$

Utilizing an identity of two Bregman divergences for arbitrary $(\rho, \bar{\rho})$:

$$D_\Omega(\rho \| \bar{\rho}) + D_\Omega(\bar{\rho} \| \rho) = \langle \delta_c \Omega(\rho) - \delta_c \Omega(\bar{\rho}), \rho - \bar{\rho} \rangle. \qquad (27)$$

We separate $\delta_c \Omega(\pi_a) - \delta_c \Omega(\pi_b)$ into $\delta_c \Omega(\pi_a) - \delta_c \Omega(\pi_c)$ and $\delta_c \Omega(\pi_c) - \delta_c \Omega(\pi_b)$ and write the rest of the derivation as follows.

$$D_\Omega(\pi_b \| \pi_a) - D_\Omega(\pi_c \| \pi_a)$$
$$= \underbrace{-D_\Omega(\pi_c \| \pi_b) + \langle \delta_c \Omega(\pi_c) - \delta_c \Omega(\pi_b), \pi_c - \pi_b \rangle}_{\text{Eq. (27)}} + \langle \delta_c \Omega(\pi_a) - \delta_c \Omega(\pi_c), \pi_c - \pi_b \rangle$$
$$= D_\Omega(\pi_b \| \pi_c) + \langle \delta_c \Omega(\pi_a) - \delta_c \Omega(\pi_c), \pi_c - \pi_b \rangle$$

Therefore, we achieve the desired identity. $\qquad \square$

**Lemma 7** (Right Bregman difference). *For all $\pi_a, \pi_b, \pi_c$, the following identity holds.*

$$D_\Omega(\pi_c \| \pi_b) - D_\Omega(\pi_c \| \pi_a) = D_\Omega(\pi_a \| \pi_b) + \langle \delta_c \Omega(\pi_a) - \delta_c \Omega(\pi_b), \pi_c - \pi_a \rangle \qquad (28)$$

*Proof of Lemma 7.* By Lemma 5, we have

$$D_\Omega(\pi_c \| \pi_b) - D_\Omega(\pi_c \| \pi_a) = -D_\Omega(\pi_b \| \pi_a) + \langle \delta_c \Omega(\pi_a) - \delta_c \Omega(\pi_b), \pi_c - \pi_b \rangle.$$

We separate $\pi_c - \pi_b$ into $\pi_c - \pi_a$ and $\pi_a - \pi_b$ and write the rest of the derivation as follows.

$$D_\Omega(\pi_c \| \pi_b) - D_\Omega(\pi_c \| \pi_a)$$
$$= \underbrace{-D_\Omega(\pi_b \| \pi_a) + \langle \delta_c \Omega(\pi_a) - \delta_c \Omega(\pi_b), \pi_a - \pi_b \rangle}_{\text{Eq. (27)}} + \langle \delta_c \Omega(\pi_a) - \delta_c \Omega(\pi_b), \pi_c - \pi_a \rangle$$
$$= D_\Omega(\pi_a \| \pi_b) + \langle \delta_c \Omega(\pi_a) - \delta_c \Omega(\pi_b), \pi_c - \pi_a \rangle$$

Therefore, we achieve the desired identity. $\qquad \square$

Additionally, we introduce the three-point inequality (Chen & Teboulle, 1993), which has been a key statement for proving MD convergence for a static cost functional (Aubin-Frankowski et al., 2022), and OMD improvement for temporal costs. Note that this three-point inequality lemma and corresponding proof mostly follows Aubin-Frankowski et al. (2022) with a slight change of notation.

**Lemma 8** (Three-point inequality). *Given $\pi \in \mathcal{M}(\mathcal{X})$ and some proper convex functional $\Psi : \mathcal{M}(\mathcal{X}) \to \mathbb{R} \cup \{+\infty\}$, if $\delta_c \Omega$ exists, as well as $\bar{\rho} = \arg\min_{\rho \in \mathcal{C}} \{\Psi(\rho) + D_\Omega(\rho \| \pi)\}$, then for all $\rho \in \mathcal{C} \cap \mathrm{dom}(\Omega) \cap \mathrm{dom}(\Psi)$: $\Psi(\rho) + D_\Omega(\rho \| \pi) \geq \Psi(\bar{\rho}) + D_\Omega(\bar{\rho} \| \pi) + D_\Omega(\rho \| \bar{\rho})$.*

*Proof of Lemma 8.* The existence of $\delta_c \Omega$ implies $\mathcal{C} \cap \mathrm{dom}(D_\Omega(\cdot | y)) = \mathcal{C} \cap \mathrm{dom}(\Omega) \cap \mathrm{dom}(\Psi)$. Set $G(\cdot) = \Psi(\cdot) + D_\Omega(\cdot \| y)$. By linearity and idempotence, we have for any $\rho \in \mathcal{C} \cap \mathrm{dom}(\Omega) \cap \mathrm{dom}(\Psi)$

$$D_G(\rho \| \bar{\rho}) = D_\Psi(\rho \| \bar{\rho}) + D_\Omega(\rho \| \bar{\rho}) \geq D_\Omega(\rho \| \bar{\rho}). \qquad (29)$$

By $\bar{\rho}$ being the optimality for $G$, for all $x \in \mathcal{C}$,

$$d^+ G(\bar{\rho}; \rho - \bar{\rho}) = \lim_{h \to 0^+} \frac{G((1-h)\bar{\rho} + h\rho) - G(\bar{\rho})}{h} \geq 0,$$

which suggests $G(\rho) \geq G(\bar{\rho}) + D_G(\rho \| \bar{\rho})$. Applying (29) to this inequality complete the proof. $\quad \square$

The following argument is from the convergence rate of mirror descent for relatively smooth and convex pairs of functionals, and extend to infinite dimensional convergence results of Lu et al. (2018) and Aubin-Frankowski et al. (2022). We aim to reformulate the statements in online learning.

**Lemma 9** (OMD improvement). *Suppose a cost $F_t : \mathcal{M}(\mathcal{X}) \to \mathbb{R}$ which is L-smooth and l-strongly-convex relative to $\Omega$ and $\eta_t \leq \frac{1}{L}$. Then, MD improves for current cost $F_t(\pi_{t+1}) \leq F_t(\pi_t)$.*

*Proof of Lemma 9.* Since $F$ is $L$ relatively smooth, we initially have

$$F_t(\pi_{t+1}) \leq F_t(\pi_t) + d^+F(\pi_t; \pi_{t+1} - \pi_t) + LD_\Omega(\pi_{t+1}|\pi_t) \tag{30}$$

Applying the three-point inequality of Lemma 8 to Eq. (30), setting a linear functional $\Psi(\rho) = \eta_t d^+F_t(\pi_t; \pi - \pi_t)$, $\rho = \pi_t$ and $\bar{\rho} = \pi_{t+1}$ yields

$$d^+F_t(\pi_t; \pi_{t+1} - \pi_t) + \frac{1}{\eta_t}D_\Omega(\pi_{t+1}|\pi_t) \leq d^+F_t(\pi_t; \rho - \pi_t) + \frac{1}{\eta_t}D_\Omega(\rho|\pi_t) - \frac{1}{\eta_t}D_\Omega(\rho\|\pi_{t+1}).$$

Since $F_t$ is $l$-strongly convex relative to $\Omega$, we also have

$$d^+F(\pi_t; \rho - \pi_t) \leq F_t(\rho) - F_t(\pi_t) - lD_\Omega(\rho|\pi_t), \tag{31}$$

Then, using (31), Eq. (30) becomes

$$F_t(\pi_{t+1}) \leq F_t(\rho) + (\frac{1}{\eta_t} - l)D_\Omega(\rho|\pi_t) - \frac{1}{\eta_t}D_\Omega(\rho|\pi_{t+1}) + (L - \frac{1}{\eta_t})D_\Omega(\pi_{t+1}\|\pi_t). \tag{32}$$

By substituting $\rho = \pi_t$, since $D_\Omega(\rho|\pi_{t+1}) \geq 0$ and $L - \frac{1}{\eta_t} \leq 0$, this shows $F_t(\pi_{t+1}) \leq F_t(\pi_t)$, *i.e.*, $F_t$ is decreasing at each iteration. This completes the proof. $\square$

A fundamental property with the dual space $\mathcal{D}$ induced by the first variation $\delta_c$ holds in our online mirror descent setting. The existence of such sequence–particularly in Sinkhorn–is well discussed by Nutz (2021) and Aubin-Frankowski et al. (2022). Focusing on mirror descent, we explicitly call this relationship with arbitrary step size $\eta_t$ as "dual iteration."

**Lemma 10** (Dual iteration). *Suppose that first variations $\delta_c F_t(\pi_t)$ and $\delta_c\Omega(\pi_t)$ exists for $t \geq 0$. Then, online mirror descent updates Eq. (6) is equivalent to $\delta_c\Omega(\pi_{t+1}) - \delta_c\Omega(\pi_t) = -\eta_t\delta_c F_t(\pi_t)$, for all $\pi_t \in \mathcal{C}, t \in \mathbb{N}$.*

*Proof of Lemma 10.* The optimization (6) is equivalent to having the property for subsequent $\pi_{t+1}$:

$$d^+F_t(\pi_t; \pi - \pi_t) + \frac{1}{\eta_t}D_\Omega(\pi\|\pi_t) \geq d^+F_t(\pi_t; \pi_{t+1} - \pi_t) + \frac{1}{\eta_t}D_\Omega(\pi_{t+1}|\pi_t)$$
$$\iff \langle \delta_c F_t(\pi_t) - \frac{1}{\eta_t}\delta_c\Omega(\pi_t), \pi - \pi_{t+1}\rangle + \frac{1}{\eta_t}\big(\Omega(\pi) - \Omega(\pi_{t+1})\big) \geq 0, \quad \forall \pi \in \mathcal{C}. \tag{33}$$

Setting the free parameter $\pi = \pi_{t+1} + h(\pi - \pi_{t+1})$ and taking the limit $h \to 0^+$ yields the result. $\square$

**Remark 4.** With applications of Lemma 10 and Lemma 4, we can achieve a concise form of iteration in the dual using our temporal cost as:

$$\delta_c\Omega(\pi_t) - \delta_c\Omega(\pi_{t+1}) = \eta_t\big(\delta_c(-H)(\pi_t) - \delta_c(-H)(\pi_t^\circ)\big)$$
$$= \eta_t\big(\delta_c\Omega(\pi_t) - \delta_c\Omega(\pi_t^\circ)\big), \tag{34}$$

where $H$ denotes the entropy, *i.e.*, the minus KL divergence with the Lebesgue measure.

Finally, we are ready to describe a suitable step size scheduling by the following arguments.

**Lemma 11** (Step size I). *Suppose that $F_t = \text{KL}(\pi\|\pi_t^\circ)$ and $\Omega = \text{KL}(\pi\|e^{-c_\varepsilon}\mu \otimes \nu)$. If ① $\lim_{t\to\infty} \eta_t = 0^+$ and ② $\sum_{t=1}^\infty \eta_t = +\infty$ ③ $\eta \leq \frac{1}{L}$, the OMD algorithm converges to a certain $\pi_\mathcal{D}^\circ$*

*Proof of Lemma 11.* From Lemma 9, we have

$$\eta_t(F_t(\pi_{t+1}) - F_t(\pi_t)) \leq -D_\Omega(\pi_t\|\pi_{t+1}) + (\eta_t L - 1)D_\Omega(\pi_{t+1}\|\pi_t). \tag{35}$$

Taking $\lim_{t\to\infty} \eta_t = 0$ ensures improvements; this means for any $\varepsilon > 0$ there exists some $0 < \delta \leq 1$ such that $D_\Omega(\pi_t\|\pi_{t+1}) + D_\Omega(\pi_{t+1}\|\pi_t) < \varepsilon$ whenever $\eta_t < \delta$. Since convexity and the lower semicontinuity of the Bregman divergence $D_\Omega$ induced by KL, we conclude that OMD to a certain point upon the assumed step size scheduling. $\square$

**Lemma 12** (Step size II). *Assume that $\min_{\pi\in\mathcal{C}} \mathbb{E}_t[D_\Omega(\pi_t, \pi_t^\circ)] > 0$ for all $t \in [1, \infty)$. Suppose that $\eta_t \to 0$ and $\lim_{T\to\infty} \mathbb{E}[\frac{1}{T}\sum_{t=1}^T D_\Omega(\pi_t\|\pi_t^\circ)] = 0$ if and only if $\sum_{t=1}^\infty \eta_t = +\infty$.*

*Proof of Lemma 12.* We note that due to dual iteration equation Eq. (34), improvements on KL in Lemma 9 are also improvements in the Bregman divergence, *i.e.* $D_\Omega(\pi_{t+1}\|\pi_t^\circ) \leq D_\Omega(\pi_t\|\pi_t^\circ)$, and if $\eta_t \to 0$, then the process $\{\pi_t\}_{t=1}^\infty$ is convergent. By the dominated convergence theorem, assuming ergodicity of nonstationary $\{\pi_t^\circ\}_{t=1}^\infty$, there is a constant $\varepsilon$ that satisfies $\mathbb{E}_{1:t+1}[D_\Omega(\pi_{t+1}\|\pi_{t+1}^\circ)] \geq \mathbb{E}_{1:t+1}[D_\Omega(\pi_{t+1}\|\pi_t^\circ)] + \varepsilon$ for $t > n$ for some $n$ as $\eta_t \to 0$, where an expectation subscripted by "$1 : t$" indicates the time average from 1 to $t$. Consequently, we achieve the following inequality

$$
\begin{aligned}
\mathbb{E}&_{1:t+1}[D_\Omega(\pi_{t+1}\|\pi_{t+1}^\circ)] \\
&\geq \mathbb{E}_{1:t+1}[D_\Omega(\pi_{t+1}\|\pi_t^\circ)] + \varepsilon \\
&\geq \mathbb{E}_{1:t}[D_\Omega(\pi_t\|\pi_t^\circ) - \langle \delta_c\Omega(\pi_{t+1}) - \delta_c\Omega(\pi_t), \pi_t^\circ - \pi_t \rangle] + \mathbb{E}_{1:t+1}[D_\Omega(\pi_{t+1}\|\pi_t)] + \varepsilon \quad \text{Lem. 6} \\
&= \mathbb{E}_{1:t}[D_\Omega(\pi_t\|\pi_t^\circ) - \eta_t D_\Omega(\pi_t\|\pi_t^\circ) + \eta_t D_\Omega(\pi_t^\circ\|\pi_t)] + \mathbb{E}_{1:t+1}[D_\Omega(\pi_{t+1}\|\pi_t)] + \varepsilon \quad \text{Eq. (34)} \\
&= (1-\eta_t)\mathbb{E}_{1:t}[D_\Omega(\pi_t\|\pi_t^\circ)] + \mathbb{E}_{1:t+1}[D_\Omega(\pi_{t+1}\|\pi_t) + \eta_t D_\Omega(\pi_t^\circ\|\pi_t)] + \varepsilon \\
&\geq (1-\eta_t)\mathbb{E}_{1:t}[D_\Omega(\pi_t\|\pi_t^\circ)] + \varepsilon'
\end{aligned}
\tag{36}
$$

for some $t$ and $0 < \varepsilon < \varepsilon'$, where Lemma 6 and Eq. (34) are used.

*Necessity.* First, we rewrite the inequality in Eq. (36) as

$$
\mathbb{E}_{1:t+1}[D_\Omega(\pi_{t+1}\|\pi_{t+1}^\circ)] \geq (1-\eta_t)\mathbb{E}_{1:t}[D_\Omega(\pi_t\|\pi_t^\circ)], \quad \forall t \geq 0.
\tag{37}
$$

Since we have assumed that $\eta_t$ converges to 0, consider a step size sequence $0 < \eta_t \leq \frac{2}{2+k}$ for $k > 0$ and $t \geq n$, where $\forall n \in \mathbb{N}$. denote a constant $a = \frac{2+k}{2} \log \frac{2+k}{k}$ and apply the elementary inequality

$$
1 - x \geq \exp(-ax), \quad \text{such that} \quad 0 < x \leq \frac{2}{2+k}.
$$

From Eq. (37), it can be seen

$$
\mathbb{E}_{1:t+1}[D_\Omega(\pi_{t+1}\|\pi_{t+1}^\circ)] \geq \exp(-a\eta_t)\mathbb{E}_{1:t}[D_\Omega(\pi_t\|\pi_t^\circ)].
$$

Applying the inequality iterative for $t = n, \ldots, T-1$ gives

$$
\begin{aligned}
\mathbb{E}_{1:T}[D_\Omega(\pi_T\|\pi_T^\circ)] &\geq \mathbb{E}_{1:n}[D_\Omega(\pi_n\|\pi_n^\circ)] \prod_{t=n}^{T-1} \exp(-a\eta_t) \\
&= \exp\left\{-a\sum_{t=n}^{T-1} \eta_t\right\} \mathbb{E}_{1:n}[D_\Omega(\pi_n\|\pi_n^\circ)].
\end{aligned}
\tag{38}
$$

From the assumption $\pi^* \neq \pi_n$, we get $D_\Omega(\pi_n\|\pi_n^\circ) > 0$. Therefore, by Eq. (38), the convergence $\lim_{t\to\infty} \mathbb{E}_{1:t}[D_\Omega(\pi_t\|\pi_t^\circ)] = 0$ implies the series $\sum_{t=1}^\infty \eta_t$ diverges to $+\infty$.

*Sufficiency.* Consider a static Schrödinger bridge problem with a constraint set

$$
\mathcal{C} = \left\{ \pi | (\mu, \nu) \in \mathcal{P}_2(\mathbb{R}^d) \cap \mathcal{P}_{\mathrm{alc}}(\mathbb{R}^d), (\varphi, \psi) \in L^1(\mu) \times L^1(\nu), \text{ and } \varphi, \psi \in C^2(\mathbb{R}^d) \cap \mathrm{Lip}(\mathcal{K}) \right\}.
$$

For $\rho, \bar{\rho} \in \mathcal{P}(\mathbb{R}^2)$ we can see

$$
D_\Omega(\bar{\rho}\|\rho) = \Omega(\bar{\rho}) - \Omega(\rho) - \langle \delta_c\Omega(\rho), \bar{\rho} - \rho \rangle \geq 0 \iff -\langle \delta_c\Omega(\rho), \bar{\rho} - \rho \rangle \geq \Omega(\rho) - \Omega(\bar{\rho}).
$$

By adding $\langle \delta_c\Omega(\bar{\rho}), \bar{\rho} - \rho \rangle$, we achieve a property:

$$
\langle \delta_c\Omega(\rho) - \delta_c\Omega(\bar{\rho}), \rho - \bar{\rho} \rangle \geq D_\Omega(\rho\|\bar{\rho}).
\tag{39}
$$

Then, Suppose that we have the asymptotic dual mean $\pi_\mathcal{D}^\circ$. Using Lemma 7, the one-step progress from the perspective of dual mean writes as

$$
\begin{aligned}
D_\Omega(\pi_\mathcal{D}^\circ\|\pi_{t+1}) - D_\Omega(\pi_\mathcal{D}^\circ\|\pi_t) &= \langle \delta_c\Omega(\pi_t) - \delta_c\Omega(\pi_{t+1}), \pi_\mathcal{D}^\circ - \pi_t \rangle + D_\Omega(\pi_t\|\pi_{t+1}). \\
&= \eta_t \langle \delta_c\Omega(\pi_t) - \delta_c\Omega(\pi_t^\circ), \pi_\mathcal{D}^\circ - \pi_t \rangle + D_\Omega(\pi_t\|\pi_{t+1}) \\
&= \eta_t \langle \delta_c\Omega(\pi_t) - \delta_c\Omega(\pi_\mathcal{D}^\circ), \pi_\mathcal{D}^\circ - \pi_t \rangle + \eta_t \langle \delta_c\Omega(\pi_\mathcal{D}^\circ) - \delta_c\Omega(\pi_t^\circ), \pi_t^\circ - \pi_t \rangle + D_\Omega(\pi_t\|\pi_{t+1}) \\
&\leq -\eta_t D(\pi_\mathcal{D}^\circ\|\pi_t) + \eta_t \langle \delta_c\Omega(\pi_\mathcal{D}^\circ) - \delta_c\Omega(\pi_t^\circ), \pi_t^\circ - \pi_t \rangle + D_\Omega(\pi_t\|\pi_{t+1})
\end{aligned}
\tag{40}
$$

for some $\lambda > 0$, where we used bound $\delta_c$ where the inequality is from Eq. (39). By using the definition followed by Hölder's inequality and Young's inequality, we can bound the expectation as

$$\mathbb{E}_{1:t+1}[D_\Omega(\pi_\mathcal{D}^\circ\|\pi_{t+1})] \leq \mathbb{E}_{1:t}[(1-\eta_t)D_\Omega(\pi_\mathcal{D}^\circ\|\pi_t)] + D_\Omega(\pi_t\|\pi_{t+1})]$$

$$\leq \mathbb{E}_{1:t}[(1-\eta_t)D_\Omega(\pi_\mathcal{D}^\circ\|\pi_t)] + \frac{\omega\eta_t^2}{2}\mathbb{E}_{1:t}[\|\nabla(\delta_c\Omega(\pi_t) - \delta_c\Omega(\pi_t^\circ))\|_{L^2(\pi_t)}]$$

$$\leq \mathbb{E}_{1:t}[(1-\eta_t)D_\Omega(\pi_\mathcal{D}^\circ\|\pi_t)] + 2\eta_t^2\omega^{-1}\mathcal{K} \tag{41}$$

where $\mathcal{K}$ is the Lipschitz constant for each log-Schrödinger potential. For the second inequality, we use the assumptions on Bregman stationary process Assumption 1 on the logarithmic Sobolev inequality LSI($\omega$) from Assumption 3. Let $\{A_t\}_{t=1}^\infty$, denote a sequence of $A_t = \mathbb{E}_{1:t}[D_\Omega(\pi_\mathcal{D}^\circ\|\pi_t)]$. Then, we have

$$A_{t+1} \leq (1-\eta_t)A_t + z\eta_t^2, \quad \forall t > n, \tag{42}$$

where $z := 2\omega^{-1}\mathcal{K}$. For a constant $h > 0$, we argue that $A_{t_1} < h$ for some $t_1 > n'$. Suppose that this statement is *not* true; we find some $t \geq t_1$ such that $A_t > h, \forall t \geq t_2$. Since $\lim_{t\to\infty}\eta_t = 0$, there are some $t > t_3 > t_2$ that $\eta_t \leq \frac{h}{4}$. However, Eq. (42) tells us that for $t \geq t_3$, for $t \geq t_3$,

$$A_{t+1} \leq (1-\eta_t)A_t + z\eta_t^2 \leq A_{t_3} - \frac{h}{4}\sum_{k=t_3}^{T}\eta_k \to -\infty \quad (\text{as } t \to \infty).$$

This results to a contradiction, which verifies $A_t < h$ for $t > n'$. Since $\lim_{t\to\infty}\eta_t = 0$, we can find some $\eta_t$ which makes $A_t$ monotonically decreasing. Therefore, we conclude the nonnegative sequence $\{A_t\}_{t=1}^\infty$ finds convergence by iteratively applying the upper bound in Eq. (42).

We now prove the theorem under consideration of the particular case of $\eta_t = \frac{2}{t+1}$. Then, Eq. (42) becomes

$$A_{t+1} \leq \left(1 - \frac{2}{t+1}\right)A_t + \frac{4z}{(t+1)^2}, \quad \forall t \geq n.$$

It follows that recursive relation writes as

$$t(t+1)A_{t+1} \leq (t-1)tA_t + 4z, \quad \forall t \geq n.$$

Iterative applying the relation, we achieve the following inequality:

$$(T-1)TA_T \leq (n-1)nA_n + 4z(T-n), \quad \forall T \geq n.$$

Therefore, we finally achieve inequality as follows:

$$\mathbb{E}_{1:T}[D_\Omega(\pi_\mathcal{D}^\circ\|\pi_T)] \leq \frac{(n-1)n\mathbb{E}_{1:n}[D_\Omega(\pi_\mathcal{D}^\circ\|\pi_n)]}{(T-1)T} + \frac{4z}{T}, \quad \forall T \geq n. \tag{43}$$

Since we assumed $\pi^* = \pi_\mathcal{D}^\circ$, $\mathbb{E}_{1:T}[D_\Omega(\pi^*\|\pi_T)] = \mathcal{O}(1/T)$, the proof of Theorem 1 is complete.

$\square$

### A.3   PROOF OF PROPOSITION 1

The proof is based on the Doob's forward convergence theorem.

**Theorem 4** (Doob's forward convergence theorem). *Let $\{X_t\}_{t\in\mathbb{N}}$ be a sequence of nonnegative random variables and let $\{\mathcal{F}_t\}_t$ be a random variable and let $\{\mathcal{F}_t\}_{t\in\mathbb{N}}$ be a filtration with $\mathcal{F}_t \subset \mathcal{F}_{t+1}$ for every $t \in \mathbb{N}$. Assume that $\mathbb{E}[\mathcal{X}_{t+1}|\mathcal{F}_t] \leq X_t$ almost surely for every $t \in \mathbb{N}$. Then, the sequence $\{X_t\}$ converges to a nonnegative random variable $X_\infty$ almost surely.*

We follow the derivation of Eq. (41): there exists $n \in \mathbb{N}$ which satisfies

$$\mathbb{E}_t[D_\Omega(\pi_\mathcal{D}^\circ\|\pi_{t+1})] \leq D_\Omega(\pi_\mathcal{D}^\circ\|\pi_t) + 2\eta_t^2\omega^{-1}\mathcal{K}, \quad \forall t \geq n$$

and since the step size is scheduled as $\lim_{t\to\infty}\eta_t = 0$, the condition $\sum_{t=1}^\infty \eta_t^2 < \infty$ enables us to define a stochastic process $\{X_t\}_{t\in\mathbb{N}}$:

$$X_t = D_\Omega(\pi_\mathcal{D}^\circ\|\pi_t) + 2\omega^{-1}\mathcal{K}\sum_{i=t}^\infty \eta_i^2. \tag{44}$$

It is straightforward that the defined random variable satisfies $\mathbb{E}_t[X_{t+1}] \leq X_t$ for $t \geq n$. Since $X_t \geq 0$, the process is a sub martingale. By Theorem 4, the sequence $\{X_t\}_{t\in\mathbb{N}}$ converges to a nonnegative random variable $X_\infty$ almost surely. Therefore $D_\Omega(\pi_\mathcal{D}^\circ\|\pi_t)$ converges to 0 almost surely. $\square$

### A.4 PROOF OF THEOREM 2

To achieve a meaningful regret bound for our problem setup, we first demonstrate the following.

**Lemma 13.** *For all* $w = \arg\min_y \{\langle \hat{g}, y \rangle + \frac{1}{\eta} D_\Omega(y\|z)\}$ *with* $\eta > 0$, *the following equation.*

$$\forall u. \langle \eta \hat{g}, w - u \rangle \leq D_\Omega(u\|z) - D_\Omega(u\|w) - D_\Omega(w\|z) \tag{45}$$

*Proof of Lemma 13.* By the first order optimality of $\{\langle g, y \rangle + D_\Omega(y\|z)\}$ as a function of $w$, we have

$$\langle \hat{g} + \frac{1}{\eta}\delta_c D_\Omega(w\|z), u - w \rangle \geq 0$$

$$\implies \langle \hat{g}, w - u \rangle \leq \frac{1}{\eta}\langle -\delta_c D_\Omega(w\|z), w - u \rangle = \frac{1}{\eta}(D_\Omega(u\|z) - D_\Omega(u\|w) - D_\Omega(w\|z)).$$

where used Lemma 6 in the derivation. This completes the proof. □

Next, we derive the one-step relationship for OMD. The result entails that the regret at each step is related to a quadratic expression of $\eta_t$, which is a key aspect of sublinear total regret. From a technical standpoint, we can see that the assumption for log Sobolev inequality generally works as a premise for Lipschitz continuity of gradient, *i.e.*, $\nabla\Omega$ in classical MD analyses.

**Lemma 14** (Single step regret). *Suppose a static Schrödinger bridge problem with the aforementioned constraint* $\mathcal{C}$. *Let* $D_\Omega$ *be the Bregman divergence wrt* $\Omega : \mathcal{P}(\mathcal{X}) \to \mathbb{R} + \{+\infty\}$. *Then,*

$$\eta_t(F_t(\pi_t) - F_t(u)) \leq D_\Omega(u\|\pi_t) - D_\Omega(u\|\pi_{t+1}) + \frac{\eta_t^2}{2\omega}\|\hat{g}_t\|_{L^2(\pi_t)}^2, \quad \forall u \in \mathcal{C} \tag{46}$$

*holds, where* $\hat{g}_t := \delta_c F_t(\pi_t) = \frac{1}{\eta_t}(\delta_c\Omega(\pi_t) - \delta_c\Omega(\pi_{t+1}))$ *in an MD iteration for the dual space for a step size* $\eta_t$, *and* $\omega > 0$ *is drawn from a type of log Sobolev inequality in Assumption 3.*

*Proof of Lemma 14.* Consider single step regrets by the adversary plays of a linearization for $\hat{g}_t$:

$$F_t(\pi_t) - F_t(u) \leq \langle \hat{g}_t, \pi_t - u \rangle.$$

Therefore, we derive a inequality for $\langle \hat{g}_t, \pi_t - u \rangle$ as follows.

$$\begin{aligned}
\langle \eta_t \hat{g}_t, \pi_t - u \rangle &= \langle \eta_t \hat{g}_t, \pi_{t+1} - u \rangle + \langle \eta_t \hat{g}_t, \pi_t - \pi_{t+1} \rangle \\
&\leq D_\Omega(u\|\pi_t) - D_\Omega(u\|\pi_{t+1}) - D_\Omega(\pi_{t+1}\|\pi_t) + \langle \eta_t \hat{g}_t, \pi_t - \pi_{t+1} \rangle \\
&= D_\Omega(u\|\pi_t) - D_\Omega(u\|\pi_{t+1}) - D_\Omega(\pi_{t+1}\|\pi_t) + \langle \delta_c\Omega(\pi_{t+1}) - \delta_c\Omega(\pi), \pi_t - \pi_{t+1} \rangle \\
&= D_\Omega(u\|\pi_t) - D_\Omega(u\|\pi_{t+1}) + D_\Omega(\pi_t\|\pi_{t+1}).
\end{aligned}$$

Since we assumed that $\hat{g}_t = \frac{1}{\eta_t}(\delta_c\Omega(\pi_t) - \delta_c\Omega(\pi_{t+1}))$ by the dual iteration and that Assumption 3 holds, we can achieve the upperbound $D_\Omega(\pi_t\|\pi_{t+1}) \leq \frac{\eta_t^2}{2\omega}\|\hat{g}_t\|_{L^2(\pi_t)}^2$ by direct calculation. □

We now show our upper bound of total regret by utilizing Lemma 14.

**Lemma 15.** *Assume* $\eta_{t+1} \leq \eta_t$. *Then,* $u \in \mathcal{C}$, *the following regret bounds for fixed* $u \in \mathcal{C}$ *hold*

$$\sum_{t=1}^T F_t(\pi_t) - F_t(u) \leq \max_{1 \leq t \leq T} \frac{D_\Omega(u\|\pi_t)}{\eta_T} + \frac{1}{2\omega}\sum_{t=1}^T \eta_t \|\tilde{g}_t\|_{L^2(\pi_t)}^2 \tag{47}$$

*where* $\hat{g}_t = \frac{1}{\eta_t}(\delta_c\Omega(\pi_t) - \delta_c\Omega(\pi_{t+1}))$.

*Proof of Lemma 15.* Define $D^2 = \max_{1 \leq t \leq T} D_\Omega(u\|\pi_t)$. We get

$$\text{Regret}(u) = \sum_{t=1}^T (F_t(\pi_t) - F_t(u)) \leq \sum_{t=1}^T \left( \frac{1}{\eta_t} D_\Omega(u\|\pi_t) - \frac{1}{\eta_t} D_\Omega(u\|\pi_{t+1}) \right) + \sum_{t=1}^T \frac{\eta_t}{2\omega}\|\hat{g}_t\|_{L^2(\pi_t)}^2$$

$$= \frac{1}{\eta_1} D_\Omega(u\|\pi_1) - \frac{1}{\eta_T} D_\Omega(u\|\pi_{T+1}) + \sum_{t=1}^{T-1} \left( \frac{1}{\eta_{t+1}} - \frac{1}{\eta_t} \right) D_\Omega(u\|\pi_{t+1}) + \sum_{t=1}^T \frac{\eta_t}{2\omega}\|\hat{g}_t\|_{L^2(\pi_t)}^2$$

$$\leq \frac{1}{\eta_1} D^2 + D^2 \sum_{t=1}^{T-1} \left( \frac{1}{\eta_{t+1}} - \frac{1}{\eta_t} \right) + \sum_{t=1}^T \frac{\eta_t}{2\omega}\|\hat{g}_t\|_{L^2(\pi_t)}^2 = \frac{D^2}{\eta_T} + \sum_{t=1}^T \frac{\eta_t}{2\omega}\|\hat{g}_t\|_{L^2(\pi_t)}^2.$$

Therefore, the proof is complete. □

Following Lemma 15 and Assumption 3, we can have the inequality

$$\sum_{t=1}^{T} F_t(\pi_t) - F_t(u) \leq \frac{D^2}{\eta_T} + \sum_{t=1}^{T} \frac{\eta_t}{2\omega} \|\hat{g}_t\|_{L^2(\pi_t)}^2 \leq \frac{D^2}{\eta_T} + 2\eta_t \omega^{-1} \mathcal{K} T.$$

where $D^2 = \max_{1 \leq t \leq T} D_\Omega(u\|\pi_t)$. Setting a constant step size $\eta_t \equiv \frac{D\sqrt{\omega}}{\sqrt{2\mathcal{K}T}}$ yields an upper bound of $D\sqrt{2\omega^{-1}\mathcal{K}T}$ which is $\Omega(\sqrt{T})$. Also, setting a heuristic scheduling $\eta_t = \frac{D\sqrt{\omega}}{\sqrt{2\sum_{t=1}^{T}\|\hat{g}_t\|^2}}$ yields $D\sqrt{2\omega^{-1}\sum_{t=1}^{T}\|\hat{g}_t\|^2}$ which has a possibility to be lower than $\mathcal{O}(\sqrt{T})$ depending on $\{\pi_t^\circ\}_{t=1}^{T}$. Therefore, we have formally expanded the convergence results of OMD (Lei & Zhou, 2020; Srebro et al., 2011) to SBPs. $\qquad\square$

## A.5 PROOF OF THEOREM 3

We first write the following equivalent convex problems.

$$\begin{aligned}
\langle \delta_c F_t(\pi_t), \pi - \pi_t \rangle + \tfrac{1}{\eta_t} D_\Omega(\pi\|\pi_t) &= \langle \delta_c D_\Omega(\pi_t\|\pi_t^\circ), \pi - \pi_t \rangle + \tfrac{1}{\eta_t} D_\Omega(\pi\|\pi_t) \\
&= \langle \delta_c \Omega(\pi_t) - \delta_c \Omega(\pi_t^\circ), \pi - \pi_t \rangle + \tfrac{1}{\eta_t} D_\Omega(\pi\|\pi_t) \\
&= D_\Omega(\pi\|\pi_t^\circ) - D_\Omega(\pi\|\pi_t) + \tfrac{1}{\eta_t} D_\Omega(\pi\|\pi_t) \\
&= \left(\frac{1}{\eta_t}\right) D_\Omega(\pi\|\pi_t^\circ) + \left(\frac{1-\eta_t}{\eta_t}\right) D_\Omega(\pi\|\pi_t)
\end{aligned}$$

Since $D_\Omega(\cdot\|\cdot) := D_{\mathrm{KL}(\cdot\|\mathcal{R})}(\cdot\|\cdot)$ for a reference measure $\mathcal{R} \in \mathcal{C}$, we can apply Lemma 4 and achieve Eq. (9). We refer to Appendix B for the stability of Wasserstein gradient flows according to the LaSalle's invariance principle. $\qquad\square$

## A.6 PROOF OF PROPOSITION 2

The proof is closely related to the work of Lambert et al. (2022) where the difference lies in we correct the Wasserstein gradient term $\dot{\alpha}_{k,\tau}$ for suitable for generally unbalanced weight. Suppose take parameterization $\theta \in (\mathcal{P}_2(\mathrm{BW}(\mathbb{R}^d)), \mathrm{WFR})$, the space of Gaussian mixtures equipped with the Wasserstein-Fisher-Rao metric, over the measure space of Gaussian particles. Following the arguments from Appendix B.2 and the studies for this particular GMM problem (Lu et al., 2019; Lambert et al., 2022) of the Wasserstein-Fisher-Rao of the KL functional is derived as

$$\nabla_{\mathrm{WFR}} \mathrm{KL}(\rho_\theta\|\rho^*) = \left(\nabla_{\mathrm{BW}} \delta\mathrm{KL}(\rho\|\rho^*), \frac{1}{2}\left(\delta\mathrm{KL}(\rho_\theta\|\rho^*) - \int \delta\mathrm{KL}(\rho\|\rho^*)\mathrm{d}\rho\right)\right), \qquad (48)$$

where we can consider the WFR gradient is taken with respect to $\theta$ of its first argument. By Eq. (48), we separately consider Wasserstein gradient in the Bures-Wasserstein space and the space of lighting that controls the amount of each Gaussian particle.

Given a functional $F : \mathcal{P}_2(\mathcal{X}) \to \mathbb{R} \cup \{+\infty\}$, the Wasserstein gradient $\nabla_{\mathbb{W}} F \cap T_\rho \mathcal{P}_2(\mathcal{X})$ such that all $\{\rho_t\}_{t\in\mathbb{R}^+}$ satisfy the continuity eqatuion starting from $\rho_0$ (Jordan et al., 1998; Villani, 2021). If the functional is the KL divergence $\mathrm{KL}(\rho\|\pi)$ we can compute the Bures-Wasserstein gradient for the Gaussian distribution with respect to $(m, \Sigma)$ using Eq. (65)

$$\begin{aligned}
\nabla_{\mathrm{BW}} F(m, \Sigma) &= (\nabla_m F(m, \Sigma), 2\nabla_\Sigma F(m, \Sigma)) \\
&= \left(\int \nabla_m \rho_{m,\Sigma} \log \frac{\rho_{m,\Sigma}}{\pi}, 2 \int \nabla_\Sigma \rho_{m,\Sigma} \log \frac{\rho_{m,\Sigma}}{\pi}\right),
\end{aligned}$$

with some abuse of notation for $\rho$. Using the following closed-form identities for the Gaussian distributions

$$\forall x. \quad \nabla_m \rho_{m,\Sigma}(x) = -\nabla_x \rho_{m,\Sigma}(x) \quad \text{and} \quad \nabla_\Sigma \rho_{m,\Sigma}(x) = \frac{1}{2}\nabla_x^2 \rho_{m,\Sigma}(x).$$

and the equivalence between the Hessian and Fisher information, we achieve the following form:

$$\nabla_{\mathrm{BW}} F(m, \Sigma) = \left(\mathbb{E}_\rho\left[\nabla \frac{\rho}{\pi}\right], \mathbb{E}_\rho\left[\nabla^2 \log \frac{\rho}{\pi}\right]\right).$$

Define $r_{k,\tau} = \sqrt{\alpha_{k,\tau}}$. Since $r_t$ follows the Fisher–Rao metric in Definition 7, by the Proposition A.1 from Lu et al. (2019) and specialization of Lambert et al. (2022), we can think of dynamics of $K$ Gaussian particles $\{\alpha_{k,\tau}, m_{k,\tau}, \Sigma_{k,\tau}\}_{k=1}^{K}$ such that

$$\dot{r}_{k,\tau} = -\frac{1}{2}\left(\mathbb{E}\left[\log\frac{\rho_{\theta_\tau}}{\rho^*}(y_{k,\tau})\right] - \frac{1}{z_\tau}\sum_{\ell=1}^{K}\alpha_\ell\mathbb{E}\left[\log\frac{\rho_{\theta_\tau}}{\rho^*}(y_{\ell,\tau})\right]\right)r_{k,\tau},$$

$$\dot{m}_{k,\tau} = -\mathbb{E}\left[\nabla\log\frac{\rho_{\theta_\tau}}{\rho^*}(y_{k,\tau})\right], \ \dot{\Sigma}_{k,\tau} = -\mathbb{E}\left[\nabla^2\log\frac{\rho_{\theta_\tau}}{\rho^*}(y_{k,\tau})\right]\Sigma_{k,\tau} - \Sigma_{k,\tau}\mathbb{E}\left[\nabla^2\log\frac{\rho_{\theta_\tau}}{\rho^*}(y_{k,\tau})\right],$$

Since $\alpha_{k,\tau} = \sqrt{r_{k,\tau}}$ by previous definition, it is straightforward that

$$\dot{\alpha}_{k,\tau} = -\left(\mathbb{E}\left[\log\frac{\rho_{\theta_\tau}}{\rho^*}(y_{k,\tau})\right] - \frac{1}{z_\tau}\sum_{\ell=1}^{K}\alpha_\ell\mathbb{E}\left[\log\frac{\rho_{\theta_\tau}}{\rho^*}(y_{\ell,\tau})\right]\right)\alpha_{k,\tau}.$$

For $\alpha_k > 0$. This completes the proof. $\qquad\square$

# B   A RIEMANNIAN PERSPECTIVE FOR VARIOUS WASSERSTEIN GEOMETRIES

## B.1   AN INTRODUCTION TO OTTO CALCULUS AND THE LASALLE INVARIANCE PRINCIPLE

We introduce a basic notion of Wasserstein gradient flows in the space of continuous probability measures by describing a historical example of the KL cost, initially introduced by Otto (2001). We refer the reader to (Ambrosio et al., 2005b; Carrillo et al., 2023) for more details and mathematical rigor. For $\mathcal{X} \subset \mathbb{R}^d$, and functions $U : \mathbb{R}_{\geq 0} \to \mathbb{R}$; $V, W : \mathcal{X} \to \mathbb{R}$. We first consider an energy function $\mathcal{E} : \mathcal{P}_2(\mathcal{X}) \to \mathbb{R}$:

$$\mathcal{E}(\rho) = \underbrace{\int_{\mathcal{X}} U(\rho(x))\,\mathrm{d}x}_{\text{internal potential } \mathcal{U}} + \underbrace{\int_{\mathcal{X}} V(x)\,\mathrm{d}\rho(x)}_{\text{external potential } \mathcal{E}_V} + \underbrace{\frac{1}{2}\int_{\mathcal{X}}(W*\rho)(x)\,\mathrm{d}\rho(x)}_{\text{interaction energy } \mathcal{W}}, \ \rho \in \mathcal{P}_2(\mathcal{X}). \quad (49)$$

For this function, we refer to the solution of the following PDE:

$$\partial_t\rho_t = \nabla\cdot\left[\rho\,\nabla(U' + V + W*\rho)\right], \qquad t \geq 0 \quad (50)$$

as the Wasserstein gradient flow of $\mathcal{E}$. Following Otto's formalization of Riemannian calculus on the continuous probability space equipped with the Wasserstein metric $(\mathcal{P}_2(\mathcal{X}), W_2)$, the PDE (50) can be interpreted close to an ODE of Riemannian gradient flow:

$$\partial_t\rho_t = -\nabla_{\mathrm{w}}\mathcal{E}(\rho), \quad (51)$$

where $\nabla_{\mathrm{w}}$ denotes the Wasserstein-2 gradient operator $\nabla_{\mathrm{w}} := \nabla\cdot(\rho\,\nabla\frac{\delta}{\delta\rho})$. Considering the Otto's Wasserstein-2 Riemannian metric $\mathfrak{g}$ (Otto, 2001; Lott, 2008), under the absolute continuity, we see that

$$\frac{\partial}{\partial t}\mathcal{E}(\rho_t) = -\mathfrak{g}_\rho\left(\frac{\partial\rho}{\partial t}, \frac{\partial\rho}{\partial t}\right) = -\int_{\mathcal{X}}\left|\nabla(U' + V + W*\rho)\right|^2\mathrm{d}\rho(x) \leq 0, \quad (52)$$

which is closely related to the strict Lyapunov condition. As a result, dynamical systems following the PDE are guaranteed to reach an equilibrium solution, under the LaSalle invariance principle for probability measures (Carrillo et al., 2023).

For a representative example, we identify Eq. (49) for the relative entropy (the KL functional) for a target density $\rho^* \in \mathcal{P}_2(\mathcal{X})$ writes

$$\mathcal{E}(\rho) = \mathrm{KL}(\rho\|\rho^*) = \underbrace{\int_{\mathcal{X}} U(\rho(x))\,\mathrm{d}x}_{\mathcal{U}} + \underbrace{\int_{\mathcal{X}} V(x)\,\mathrm{d}\rho(x)}_{\mathcal{E}_V} - C,$$

where $U(s) = s\log s$, $V(x) = -\log\rho^*(x)$, and $C = \mathcal{U}(\rho^*) + \mathcal{E}_V(\rho^*)$. Recall that $\delta\mathcal{E}(\rho) = \log\frac{\rho(x)}{\rho^*}$, then we have

$$\nabla_{\mathrm{w}}\mathcal{E}(\rho) = \mathfrak{G}_\rho^{-1}\delta E(\rho) = -\nabla\cdot[\rho\nabla\delta E(\rho)] = \nabla\cdot\left[\rho\nabla\log\frac{\rho}{\rho_*}\right] \quad (53)$$

where $\mathfrak{G}$ denotes the metric tensor in matrix form. We can derive the the Fokker–Planck equation

$$\partial_t \rho_t = -\nabla \cdot (\rho \nabla \log \rho^*) + \Delta \rho_t,$$

describing the time evolution of the probability density. Combining the convexity of KL and the LaSalle invariance principle Wasserstein gradient flows, the PDE reaches a unique stationary solution of $\frac{e^{-V(x)}}{\int_{\mathcal{X}} e^{-V(y)} \mathrm{d}y}$.

## B.2 BACKGROUND ON WASSERSTEIN-FISHER-RAO AND OTHER RELATED GEOMETRIES

The Wasserstein-Fisher-Rao geometry is also known as *Hellinger–Kantorovich* in some of papers (Liero et al., 2016; 2018). In this section, we provide an overview of the geometry tailored to meet our technical needs. Along the way, we also briefly describe relevant metrics and geometries.

**The Wasserstein space.** Let $\mu, \nu \in \mathcal{P}_2(\mathbb{R}^d)$ be a probability densities with respect to the Lebesgue measure. we define the squared Wasserstein distance as

$$W_2^2(\mu, \nu) := \min_{\pi \in \Pi(\mu,\nu)} \int_{\mathbb{R}^2 \times \mathbb{R}^2} \frac{1}{2} \|x - y\|^2 \mathrm{d}\pi(x, y) \tag{54}$$

Then, the Brenier theorem (Villani, 2021) states that there exists the optimal Brenier map that pushes forward $\mu$ to $\nu$, *i.e.* $\nu = \nabla\zeta_{\#}\mu$, where $\zeta : \mathbb{R}^d \to \mathbb{R}^d \cup \{+\infty\}$ is a convex and lower semicontinuous function. In the fluid dynamical version, the Brenier map yields a constant-speed of geodesic $\{\mu_t\}_{t \in [0,1]}$ formally described by

$$\rho_t = (\nabla\zeta_t)_{\#}\mu, \qquad \nabla\zeta_t := (1-t)\mathrm{id} + t\nabla\zeta. \tag{55}$$

Assuming the existence of such geodesic, we can understand finding optimality of Eq. (55) the Benamou-Brenier formulation (Benamou & Brenier, 2000), which finds a velocity $v_t$ by minimizing the functional

$$W_2^2(\mu, \nu) = \min_{\rho, v} \left\{ \int_0^1 \int_{\mathbb{R}^d} \frac{1}{2} \|v_t(x)\|^2 \mathrm{d}\rho_t(x) \mathrm{d}t \;\middle|\; \rho_0 = \mu, \; \rho_1 = \nu, \; \partial_t \rho_t = -\nabla \cdot (v_t \rho_t) \right\}. \tag{56}$$

The equation dictates *how* the mass should be transported (which shall be a constant speed) while satisfying the continuity equation of path measure. In the Otto calculus (Otto, 2001), we can understand the Benamou-Brenier formula (56) as a Riemannian formulation for $W_2$. In this interpretation, the tangent space at $\rho \in \mathcal{P}_2(\mathcal{X})$ are measures of the form $\delta\rho = -\nabla \cdot (v\rho)$ with a velocity field $v \in L^2(\rho, \mathbb{R}^d)$ and the metric is given by

$$\|\rho\|_\rho^2 = \inf_{v \in L^2(\rho, \mathbb{R}^d)} \left\{ \int \|v\|^2 \mathrm{d}\rho \;\middle|\; \delta\rho = -\nabla \cdot (v\rho) \right\}. \tag{57}$$

This exhibits dynamics in the Wasserstein space of probability densities metric generally governed by the continuity equation, implying the mass of probability is preserved.

**Fisher-Rao metric.** The Fisher–Rao metric is a metric on the space of positive measures $\mathcal{P}_+$ with possibly different total masses. We use the following definition throughout the paper.

**Definition 7** (Fisher–Rao metric). The Fisher–Rao distance between measures $\rho_0, \rho_1 \in \mathcal{M}_+$ is given by

$$d_{\mathrm{FR}}^2(\rho_0, \rho_1) := \min_{\rho, v \in \mathcal{A}[\rho_0, \rho_1]} \int_0^1 \int_{\mathbb{R}^d} \frac{1}{2} \omega_t^2(x) \mathrm{d}\rho_t(x) \mathrm{d}t = 2 \int_{\mathbb{R}^d} \left| \sqrt{\frac{\mathrm{d}\rho_0}{\mathrm{d}\lambda}} - \sqrt{\frac{\mathrm{d}\rho_1}{\mathrm{d}\lambda}} \right|^2 \mathrm{d}\lambda$$

where $\mathcal{A}$ is an admissible set for a scalar field on positive measures; $\lambda$ is any reference measure such that $\rho$ and $\rho'$ are both absolutely continuous with respect to $\lambda$, with Radon-Nikodym derivatives $\frac{\mathrm{d}\rho_i}{\mathrm{d}\lambda}$.

The equivalence between the square Fisher–Rao distance and squared Hellinger distance quantifies the similarity between two probability distributions ranging from 0 to 1. The total variation bounds the squared form and is well-studied in the information geometry (Amari, 2016). PDEs of the form $\partial_t \rho_t = \alpha_t \rho_t$ are called reaction equations of $\alpha_t$, which describes dynamics regarding concentration.

**Wasserstein-Fisher-Rao.** The WFR geometry, or spherical Hellinger-Kantorovich distance, considers liftings of positive, complete, and separable measures while preserving the total mass. This

can be expresses as combining the Fisher–Rao and Wasserstein geometries characterized by PDE such as (Liero et al., 2016):

$$\partial_t \rho_t + \nabla \cdot (v_t \rho_t) = \frac{\omega_t}{2} \rho_t. \tag{58}$$

One problem, is that the PDE (58) In order to stay the dynamics on the space of probability measures, which is our interest, we adopt the definition from (Lu et al., 2019; Lambert et al., 2022) the equation becomes

$$\partial_t \rho_t + \nabla \cdot (\rho_t v_t) = \frac{1}{2} \left( \beta_t - \int \beta_t \mathrm{d}\rho_t \right) \rho_t, \tag{59}$$

which satisfies mass conservation. For the geometry, the norm on tangent space is given by

$$\|(\beta_t, \rho)\|_\rho^2 := \int \left\{ \left( \omega - \int \beta_t \, \mathrm{d}\rho \right)^2 + \|v\|^2 \right\} \mathrm{d}\rho. \tag{60}$$

and we define the WFR distance as

$$d_{\mathrm{WFR}}^2(\rho_0, \rho_1) := \inf_{\rho, \beta_t, v} \left\{ \int_0^1 \|(\beta_t, v_t)\|_{\rho_t}^2 \mathrm{d}t \,\Big|\, \{\rho_t, \beta_t, v_t\}_{t \in [0,1]} \text{ satisfies } (59) \right\}. \tag{61}$$

Since WFR gradient dynamics over the Bures-Wasserstein space can be analytically derived, we were able to design a computational method for OMD iterates in the WFR geometry. Using Proposition 2, this geometry allowed the VMSB algorithm to perform tractable gradient computation within Wasserstein space.

### B.3 THE BURES-WASSERSTEIN SPACE AND A MIXTURE OF GAUSSIANS

The space of Gaussian distribution in the Wasserstein space is known as Bures-Wasserstein space, denoted as $\mathtt{BW}(\mathbb{R}^d)$. Given $\theta_0, \theta_1 \in \mathtt{BW}(\mathbb{R}^d)$, we can identify the space with the manifold $\mathbb{R}^d \times \mathbf{S}_{++}^d$, where $\mathbf{S}_{++}^d$ denotes the space of symmetric positive definite matrices. For $\theta_0 = (m_0, \Sigma_0)$ and $\theta_1 = (m_1, \Sigma_1)$ an affine map from $p_{\theta_0}$ to $p_{\theta_1}$ is given as a closed-form expression:

$$\nabla \zeta(x) = m_1 + \Sigma_0^{-1/2} \left( \Sigma_0^{1/2} \Sigma_1 \Sigma_0^{1/2} \right)^{1/2} \Sigma^{-1/2} (x - m_0).$$

Note that the constant-speed geodesic also lies in $\mathtt{BW}(\mathbb{R}^d)$, as pushforward of a Gaussian with an affine map is also a Gaussian. Therefore, it can be said that $\mathtt{BW}(\mathbb{R}^d)$ is a geodesically convex subset of $\mathcal{P}_2(\mathbb{R}^d)$. For the Brenier map, a constant-speed geodesic in $\mathtt{BW}(\mathbb{R}^d)$, for the tangent vector to the geodesic $(r, S)$

$$p_{\theta_t} = \exp_{p_{\theta_0}}\big(t \cdot (r, S)\big) = \mathcal{N}\big(m_0 + tr, (tS + I_d)\Sigma_0(tS + I_d)\big), \tag{62}$$

and the dynamics at its current position at time $t = 0$ is represented as

$$\dot{m}_0 = r, \tag{63}$$

$$\dot{\Sigma}_0 = S\Sigma_0 + \Sigma_0 S. \tag{64}$$

Generalizing this geodesic dynamics, the Bures-Wasserstein gradient $\nabla_{\mathtt{BW}} f$ of a function $f : \mathbb{R}^d \times \mathbf{S}_{++}^d \to \mathbb{R}$ for a tangent vector $(r, S)$ at time 0 Altschuler et al. (2021)

$$\big\langle \nabla_{\mathtt{BW}} f(m_0, \Sigma_0), (r, S) \big\rangle_{\mathtt{BW}} = \partial_t f(m_t, \Sigma_t)\Big|_{t=0}$$

Identifying each component, we achieve the following result of Wasserstein gradient flow in Bures-Wasserstein space as

$$\nabla_{\mathtt{BW}} f = (\nabla_m f, 2\nabla_\Sigma f), \tag{65}$$

where $\nabla_m$ and $\nabla_\Sigma$ denote Euclidean gradient. Please see the work of Altschuler et al. (2021) (Appendix A) and Lambert et al. (2022) (Appendix B) for further geometric properties and discussion for this parameter space.

## C  DETAILS ON THE EXPERIMENTS

### C.1  RATIONALES OF THE GMM PARAMETERIZATION FOR VMSB

Our parameterization choice follows LightSB (Korotin et al., 2024) because of the following two key reasons. First, GMMs ensure that the model space satisfies certain measure concentration, which is suitable for analyzing theoretical properties of SB models (Conforti et al., 2023). For instance, we analyzed the regret under the log Sobolev inequality in Theorem 2. Enforcing the LightSB parameterization will automatically satisfy Assumption 3. Secondly, VMSB requires tractable gradient computation of Wasserstein gradient flow in § 4.3. As shown in Proposition 2, we can perform VMSB using the variational inference in the WFR geometry of the GMM parameterization.

### C.2  STEP SIZE SCHEDULING AND WARM-UPS

For step size scheduling, we followed the theoretical result in Theorem 1 and Proposition 1, and chose $\eta_1 = 1$ and $\eta_T \in \{0.05, 0.1\}$ with harmonic sequences, as illustrated in Fig. 9. For high dimensional tasks in MSCI (1000d), MNIST-EMNIST (784d), and latent FFHQ Image-to-Image transfer tasks (512d), the initial *warmup* steps helped starting a training sequence from a reasonable starting point as this set $\eta_t = 1$ as verified in Fig. 5 (c).

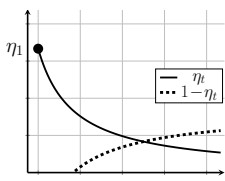

Figure 9: A sequence example of $\eta_t$ and $1-\eta_t$

### C.3  2D SYNTHETIC DATASETS

Fig. 10 demonstrates that our method achieved the SB model for the various volatility $\varepsilon$. For various configurations, most of baseline SB algorithms are capable of learning in the 2D space (10). In order to align our theoretical arguments, we selectively offered only 12.5% of the samples to the SB solvers based on the angles

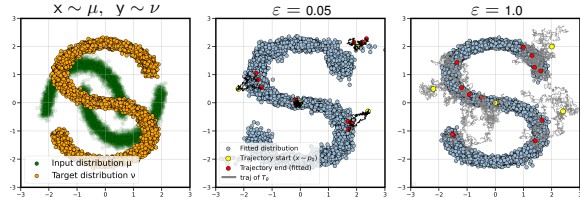

Figure 10: SB processes $\mathcal{T}_\theta$ with different volatility $\varepsilon$.

measured from the origin. For instance, we provided data for angle of $[0, \pi/4]$ for first $t \in [0, 25]$ steps, and so on. Since this requires 200 batches for the full rotation of the filter, the problem became substantially more challenging, and LightSB and LightSB-M algorithms oftentimes failed on this online learning setting.

### C.4  ENTROPIC OPTIMAL TRANSPORT BENCHMARK

Our hyperparameter for the EOT benchmarks choices mostly follow the official repositories of the LightSB[4] and LightSB-M[5]. Since it is known that initial distribution $\mu$ is the standard Gaussian distribution (Gushchin et al., 2024b), we only trained $v_\theta$ using the variational MD algorithm. Due to the huge number of configurations, some hyperparameter settings were not clearly reported. Thus, we conducted our own examination on these cases; we replicated better performance than the reported numbers by carefully dealing each benchmark configuration.

### C.5  MNIST-TO-EMNIST IMAGE TRANSFER

Suppose a discriminator network, denoted as $D$, is equipped with useful architectural properties for discriminating images. In adversarial learning, we only used a simple architecture shown in Table 5 for simplicity, and this can be replaced with more complex architecture for more sophisticated images. The discriminator outputs a binary classification regarding authenticity through sigmoidal outputs, *i.e.*, $D(x) \in [0, 1] \ \forall x \in \mathbb{R}^{28 \times 28 \times 1}$. For

Table 5: A simple discriminator.

| Layer Type | Shape |
|---|---|
| Input Layer | (-1, 28, 28, 1) |
| Conv Layer 1 | (-1, 26, 26, 32) |
| Average Pool | (-1, 13, 13, 32) |
| Conv Layer 2 | (-1, 11, 11, 64) |
| Average Pool | (-1, 5, 5, 64) |
| Flatten | (-1, 1600) |
| Dense | (-1, 512) |
| Dense | (-1, 256) |
| Dense | (-1, 1) |

---

[4] https://github.com/ngushchin/LightSB
[5] https://github.com/SKholkin/LightSB-Matching

image samples $\mathbf{x} = \{x^1, \ldots, x^M\} \sim \mu$, we trained the discriminator $D$ with the logistic regression:

$$\underset{D}{\text{maximize}} \ \frac{1}{N} \sum_{n=1}^{N} \log D(y^n) + \frac{1}{M} \sum_{m=1}^{M} \log(1 - D(\hat{y}_\phi^m)), \tag{66}$$

where $\hat{y}_\phi^m$ in the right-hand side denotes a sample from an SB model parameterized by $\phi$, generated using an input $x^m$. Let us formally define the distribution $\rho_\phi$, which represents the probability of the aforementioned adversarial samples at the law of SB process at time $t = 1$. For a completely separable metric space, the discriminator converges at $D(x) = \frac{\nu(x)}{\nu(x)+\rho_\phi(x)}$ (Goodfellow et al., 2014).

In the adversarial learning technique, retaining a fully differentiable computation path from the input pixels to the discriminator outputs is essential. Therefore, we implemented a differentiable inference function using the categorical reparameterization trick with Gumbel-softmax (Jang et al., 2016), as well as the Gaussian reparameterization trick. These tricks enabled learning with samples generated through LightSB-Adv-$K$, directly by maximizing

$$\tilde{\mathcal{J}}(\phi) = \frac{1}{M} \sum_{m=1}^{M} \log D(y_\phi^m) - \log(1 - D(y_\phi^m)),$$

where the term essentially represents the *logit* function $\mathrm{logit}(D(y)) = \log \frac{D(y)}{1-D(y)}$. When $D$ approaches the equilibrium, the logit can be approximated as $\mathrm{logit}(D(y)) \approx \log \frac{\nu(y)}{\rho_\phi(y)}$, which leads to $\tilde{\mathcal{J}}(\phi) \approx \int \log \frac{\nu(y)}{\rho_\phi(y)} \rho_\phi(y) \mathrm{d}y = \mathrm{KL}(\rho_\phi \| \nu)$. Note that the training directly corresponds to the divergence minimization of the SB/EOT problem as expressed in Eqs. (4) and (20), under the disintegration theorem of Schrödinger bridge (Léonard, 2014). Hence, we considered adversarial learning as the baseline for training the SB model in this experiment. Among our attempts, only the LightSB-Adv method successfully generated learning signals to train GMM-based models, while the losses proposed by LightSB and LightSB-M failed to generate relevant images with high fidelity. We fixed the covariance after warm-ups, and we used $\varepsilon = 10^{-3}$ based on our hyperparameter search.

# D LIMITATIONS AND DIRECTIONS FOR FUTURE RESEARCH

**Computation.** We have presented performance regarding efficiency and scalability up to 1,000 dimensions in the experiments. The computational of VMSB requires quadratic time for computing the Wasserstein gradient flow (asymptotically $\mathcal{O}(K^2 n_y)$) and memory footprints of $\{Y_k^x\}_{k=1}^K$ for estimating with internal Gaussian particles (asymptotically $\mathcal{O}(K n_y)$). For fast computation, we utilized the JAX automatic differentiation library (Bradbury et al., 2018) for computing gradients and Hessians in Proposition 2. For a small number of dimensions less than or equal to 20, this overhead is negligible; VMSB can run on a 4-core CPU, and the training can be reasonably trained within 10 minutes. For a large number of dimensions, such as 512, the wall clock time for finishing the FFHQ dataset in the image-to-image transfer experiment was less than 30 minutes using parallel computing of a single NVIDIA TITAN RTX GPU. While the Wasserstein gradient flow theory in the subspace of $\mathcal{P}_2(\mathbb{R}^d)$ enables us to estimate the mirror descent update more accurately, its computational efficiency is not yet comparable to well-established automatic differentiation libraries. If numerical computation for high order derivatives are readily available with low computational cost in future, we will be able to train more stable and reliable probabilistic models.

**Limitations.** GMM-based SB models, due to the lack of deep structural processing, tend to focus on *instance-level* associations in images in coupling rather than the *subinstance-* or *feature-level* associations that are intrinsic to deep generative models. As a result, while VMSB produces statistically valid representations of optimal transportation within the given architectural constraints, these outcomes may be perceived as somewhat "synthetic." Nevertheless, GMM-based models still hold an irreplaceable role in numerous problems such as latent diffusion and variational methods, due to their simplicity and distinctive properties (Korotin et al., 2024). As we successfully demonstrated in two distinct ways of interacting with neural networks for solving unpaired image transfer, we hope our theoretical and empirical findings help novel neural architecture studies.

**Directions for future research.** One of the primary objectives was to provide a rigorous mathematical analysis of robust SB acquisition through the lens of OMD. We hope that the proposed

Table 6: EOT Benchmark scores of $B\mathbb{W}_2^2$-UVP ↓ (%). Results of classical EOT solvers marked with † are taken from (Korotin et al., 2024). Additionally, LightSB-EMA indicates the exponential moving average (EMA; Morales-Brotons et al., 2024) of parameters in LightSB ($decay = 0.99$).

| Type | Solver | $\varepsilon = 0.1$ | | | | $\varepsilon = 1$ | | | | $\varepsilon = 10$ | | | |
|---|---|---|---|---|---|---|---|---|---|---|---|---|---|
| | | $d=2$ | $d=16$ | $d=64$ | $d=128$ | $d=2$ | $d=16$ | $d=64$ | $d=128$ | $d=2$ | $d=16$ | $d=64$ | $d=128$ |
| | Classical solvers (best) (Korotin et al.)† | 0.016 | 0.05 | 0.25 | 0.22 | 0.005 | 0.09 | 0.56 | 0.12 | 0.01 | 0.02 | 0.15 | 0.23 |
| Bridge-M | DSBM (Shi et al.)‡ | 0.03 | 0.18 | 0.7 | 2.26 | 0.04 | 0.09 | 1.9 | 7.3 | 0.26 | 102 | 3563 | 15000 |
| Bridge-M | SF²M-Sink (Tong et al.)‡ | 0.04 | 0.18 | 0.39 | 1.1 | 0.07 | 0.3 | 4.5 | 17.7 | 0.17 | 4.7 | 316 | 812 |
| rev. KL | LightSB (Korotin et al.) | 0.004±0.004 | 0.009±0.004 | 0.023±0.003 | 0.036±0.003 | 0.004±0.005 | 0.009±0.003 | 0.016±0.002 | 0.035±0.003 | 0.009±0.004 | 0.013±0.007 | 0.034±0.004 | 0.066±0.008 |
| Bridge-M | LightSB-M (Gushchin et al.) | 0.005±0.003 | 0.012±0.004 | 0.034±0.003 | 0.063±0.002 | 0.005±0.001 | 0.027±0.007 | 0.057±0.010 | 0.108±0.004 | 0.004±0.002 | 0.017±0.007 | 0.133±0.010 | 0.409±0.042 |
| EMA | LightSB-EMA | 0.004±0.002 | 0.014±0.003 | 0.021±0.003 | 0.044±0.001 | 0.004±0.003 | 0.009±0.004 | 0.013±0.001 | 0.032±0.004 | 0.004±0.001 | 0.008±0.003 | 0.023±0.013 | 0.010±0.002 |
| Var-MD | VMSB (ours) | 0.003±0.001 | 0.007±0.003 | 0.018±0.002 | 0.039±0.001 | 0.002±0.002 | 0.004±0.001 | 0.009±0.001 | 0.023±0.003 | 0.005±0.007 | 0.006±0.004 | 0.011±0.010 | 0.011±0.004 |
| Var-MD | VMSB-M (ours) | 0.002±0.001 | 0.010±0.067 | 0.031±0.004 | 0.056±0.005 | 0.003±0.004 | 0.005±0.002 | 0.032±0.006 | 0.077±0.018 | 0.003±0.003 | 0.011±0.004 | 0.117±0.012 | 0.429±0.748 |

Table 7: EOT scores of $cB\mathbb{W}_2^2$-UVP, which corresponds to the fully extended version of Table 2.

| Type | Solver | $\varepsilon = 0.1$ | | | | $\varepsilon = 1$ | | | | $\varepsilon = 10$ | | | |
|---|---|---|---|---|---|---|---|---|---|---|---|---|---|
| | | $d=2$ | $d=16$ | $d=64$ | $d=128$ | $d=2$ | $d=16$ | $d=64$ | $d=128$ | $d=2$ | $d=16$ | $d=64$ | $d=128$ |
| | Classical solvers (Korotin et al.)† | 1.94 | 13.67 | 11.74 | 11.4 | 1.04 | 9.08 | 18.05 | 15.23 | 1.40 | 1.27 | 2.36 | 1.31 |
| Bridge-M | DSBM (Shi et al.)‡ | 5.2 | 10.8 | 37.3 | 35 | 0.3 | 1.1 | 9.7 | 31 | 3.7 | 105 | 3557 | 15000 |
| Bridge-M | SF²M-Sink (Tong et al.)‡ | 0.54 | 3.7 | 9.5 | 10.9 | 0.2 | 1.1 | 9 | 23 | 0.31 | 4.9 | 319 | 819 |
| rev. KL | LightSB (Korotin et al.) | 0.007±0.005 | 0.040±0.023 | 0.100±0.013 | 0.140±0.003 | 0.014±0.003 | 0.026±0.002 | 0.060±0.004 | 0.140±0.003 | 0.019±0.005 | 0.027±0.005 | 0.052±0.002 | 0.092±0.001 |
| Bridge-M | LightSB-M (Gushchin et al.) | 0.017±0.004 | 0.088±0.014 | 0.204±0.036 | 0.346±0.036 | 0.020±0.007 | 0.069±0.016 | 0.134±0.014 | 0.294±0.017 | 0.014±0.001 | 0.029±0.004 | 0.207±0.005 | 0.747±0.028 |
| EMA | LightSB-EMA | 0.005±0.002 | 0.040±0.014 | 0.078±0.007 | 0.149±0.006 | 0.012±0.002 | 0.022±0.003 | 0.051±0.001 | 0.127±0.002 | 0.017±0.003 | 0.021±0.003 | 0.025±0.002 | 0.042±0.002 |
| Var-MD | VMSB (ours) | 0.004±0.001 | 0.012±0.002 | 0.038±0.002 | 0.101±0.002 | 0.010±0.001 | 0.018±0.001 | 0.044±0.001 | 0.114±0.001 | 0.013±0.001 | 0.019±0.001 | 0.021±0.008 | 0.040±0.001 |
| Var-MD | VMSB-M (ours) | 0.015±0.016 | 0.067±0.036 | 0.108±0.020 | 0.253±0.107 | 0.010±0.001 | 0.019±0.001 | 0.094±0.010 | 0.222±0.033 | 0.013±0.001 | 0.029±0.003 | 0.193±0.015 | 0.748±0.036 |

OMD theory will find multiple applications across various domains. One line of future studies is a general understanding of learning in diffusion models with various regularizations. This includes diffusion models in various problem-specific constraints, and geometric constraints from manifolds. Another direction is the extension of the theoretical results into network architecture design. From Section 4.2, a pair of Schrödinger potentials represent a dual representation of SB in a statistical manifold. In (Gigli & Tamanini, 2020), such potentials satisfy the Hamilton-Jacobi-Bellman (HJB) equations and, this can be trained with forward-backward SDE (SB-FBSDE) as presented in (Liu et al., 2022). However, this requires many simulation samples from SDEs, and the requirements for applying VMSB contain a tractable way of estimating gradient flows, and a guarantee of measure concentration. Therefore, we expect there will be a new studies of energy-based neural architecture for efficiently representing SB, which will advance various subfields of machine learning.

**Reproducibility statement.** Comprehensive justification and theoretical background are presented in Appendices A and B. Since the primary contributions of this paper pertain to the learning methodology, we ensured that all architectures and hyperparameters remained consistent across the LightSB variants. All datasets utilized in this study are available for download alongside the training scripts. Please refer to Appendix C for more information on the experimental setups.

# E ADDITIONAL EXPERIMENTAL RESULTS

## E.1 ADDITIONAL RESULTS ON THE EOT BENCHMARK

We present the full results of EOT benchmark experiments. Tables 6 and 7 show comprehensive statistics on the EOT benchmark with more SB solvers. As mentioned in § 6.2, the VMSB and VMSB-M solvers consistently brought better performance with low standard deviations of scores for $cB\mathbb{W}_2^2$-UVP and $B\mathbb{W}_2^2$-UVP measures. We note that the experiment was conducted in a highly controlled setting with identical model configurations; with all other aspects controlled and outcomes differing only by learning methods, the consistent performance gains of our work were a well-anticipated result from our theoretical analysis.

## E.2 ADDITIONAL IMAGE GENERATION RESULTS

In the unpaired EMNIST-to-MNIST translation task, we measured FID scores for various $K$ for the SB parameterization. We considered $K \in \{64, 256, 1024, 4096\}$ with $\varepsilon = 10^{-3}$ for our VMSB algorithm. Our observations, both qualitative and quantitative, indicate that higher modalities yield higher-quality samples. In every case of $K$, VMSB-adv outperformed its counterpart. For instance, Fig. 11 demonstrates that VMSB generates more diverse samples with high fidelity. Notably, we achieved an FID score of 15.4 using a naïve

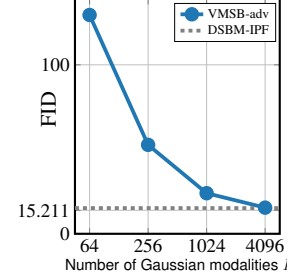

Figure 11: FID vs modality

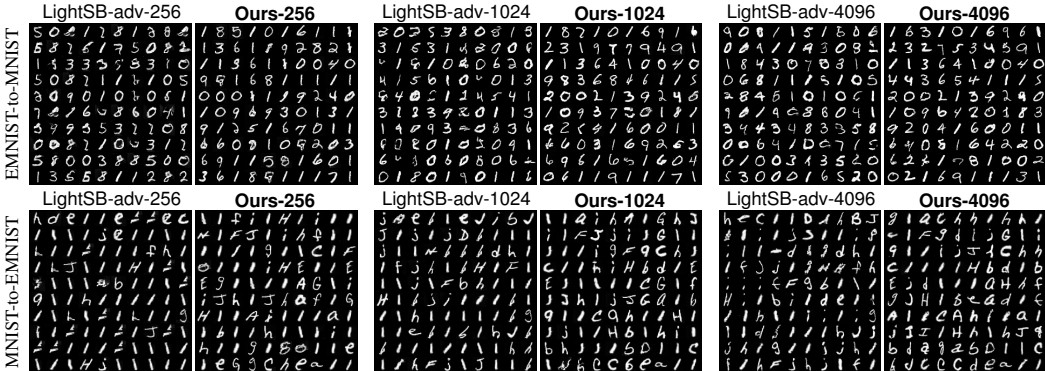

Figure 12: Generation results for unpaired image-to-image translation. We considered image data from MNIST and EMNIST (containing the first ten letters), sized as 28×28 pixels. For comparison, we trained GMM-based models with adversarial learning using a simple logistic discriminator (Table B2). This was used as both a benchmark and a tractable target SB model (LightSB-adv-$K$). Our method in the raw pixel domain, denoted as Ours-$K$, demonstrated qualitative improvements in terms of diversity and clarity of image samples by effectively handling the mode collapsing issue.

Table 8: MNIST transfer statistics.

|  | FID | Time | Parameters |
|---|---|---|---|
| LightSB-256 | 61.257 | 30m | 0.4M |
| LightSB-1024 | 26.487 | 53m | 1.6M |
| LightSB-4096 | 20.017 | 135m | 6.4M |
| VMSB-256 | 52.634 | 76m | 0.4M |
| VMSB-1024 | 24.022 | 203m | 1.6M |
| VMSB-4096 | 15.471 | 44h | 6.4M |
| DSBM-IMF | 11.429 | 42h | 6.6M |

Table 9: FID scores and differences for generated MNIST.

|  | FID (Train) | FID (Test) | Diff. (test − train). |
|---|---|---|---|
| LightSB-adv-256 | 60.746 | 61.604 | 0.858 |
| LightSB-adv-1024 | 25.934 | 26.569 | 0.635 |
| LightSB-adv-4096 | 19.960 | 20.196 | 0.237 |
| VMSB-adv-256 | 51.684 | 52.283 | 0.599 |
| VMSB-adv-1024 | 23.853 | 24.053 | 0.200 |
| VMSB-adv-4096 | 15.508 | 15.496 | −0.012 |

convolutional neural network discriminator with low MSD similarity scores, which represent competitive results for this task (Shi et al., 2023).

Fig. 12 demonstrates that VMSB generated more diverse samples with high fidelity. Note that the proposed method suffers less from mode collapse than LightSB method (especially on the transfer MNIST-to-EMNIST), with the same Gaussian mixture setting. This result is especially a good point where the difference only lies in the learning methodology, which aligns with our theory. Tables 8 and 9 effectively shows the statistics and FID scores on the both train and the test datasets. The quantitative results highlight that the VMSB solver is more preformant with less overfitting than its counterpart. Consequently, our claim regarding the stability of SB solution acquisition is verified by additional experiments involving pixel spaces.

We present Embedding-ED scores (Jayasumana et al., 2023) and some qualitative generation results in Table 10, which is visualized in Fig. 8. SF$^2$M-Sink For quantitative results, we calculated statistics from ED scores on embeddings of the ALAE model (Pidhorskyi et al., 2020), for the four different tasks: *Adult → Child*, *Child → Adult*, *Female → Male*, and *Male → Female*. The results show that VMSB is capable of translating an arbitrary representation, which is closer to target domain than baselines. To qualitatively verify these results, we generated images using LightSB and VMSB in Figures 13 and 14. Since these improvements are purely based on information geometry and learning theory, we anticipate that many following works on the variational principle application across various fields such as image processing, natural language processing, and control systems (Caron et al., 2020; Liu et al., 2023; Alvarez-Melis & Jaakkola, 2018; Chen et al., 2022).

Table 10: ALAE Embedding-ED scores. To evaluate the performance, we computed averages and standard deviations of the ED scores across four different transfer tasks.

|  | $\varepsilon = 0.1$ | $\varepsilon = 0.5$ | $\varepsilon = 1.0$ | $\varepsilon = 10.0$ |
|---|---|---|---|---|
| SF$^2$M-Sink | $0.02916 \pm 0.00145$ | $0.04112 \pm 0.00191$ | $0.05670 \pm 0.00249$ | $0.06641 \pm 0.00441$ |
| DSBM-IMF | $0.02275 \pm 0.00101$ | $0.03358 \pm 0.00142$ | $0.04866 \pm 0.00168$ | $0.06474 \pm 0.00381$ |
| LightSB | $0.01086 \pm 0.00045$ | $0.02382 \pm 0.00093$ | $0.03462 \pm 0.00148$ | $0.05376 \pm 0.00273$ |
| LightSB-M | $0.01066 \pm 0.00055$ | $0.02366 \pm 0.00107$ | $0.03519 \pm 0.00153$ | $0.05975 \pm 0.00298$ |
| VMSB | $\mathbf{0.01002 \pm 0.00055}$ | $\mathbf{0.02288 \pm 0.00101}$ | $\mathbf{0.03396 \pm 0.00174}$ | $\mathbf{0.05315 \pm 0.00307}$ |
| VMSB-M | $\mathbf{0.00997 \pm 0.00054}$ | $\mathbf{0.02298 \pm 0.00106}$ | $\mathbf{0.03391 \pm 0.00140}$ | $\mathbf{0.05351 \pm 0.00241}$ |

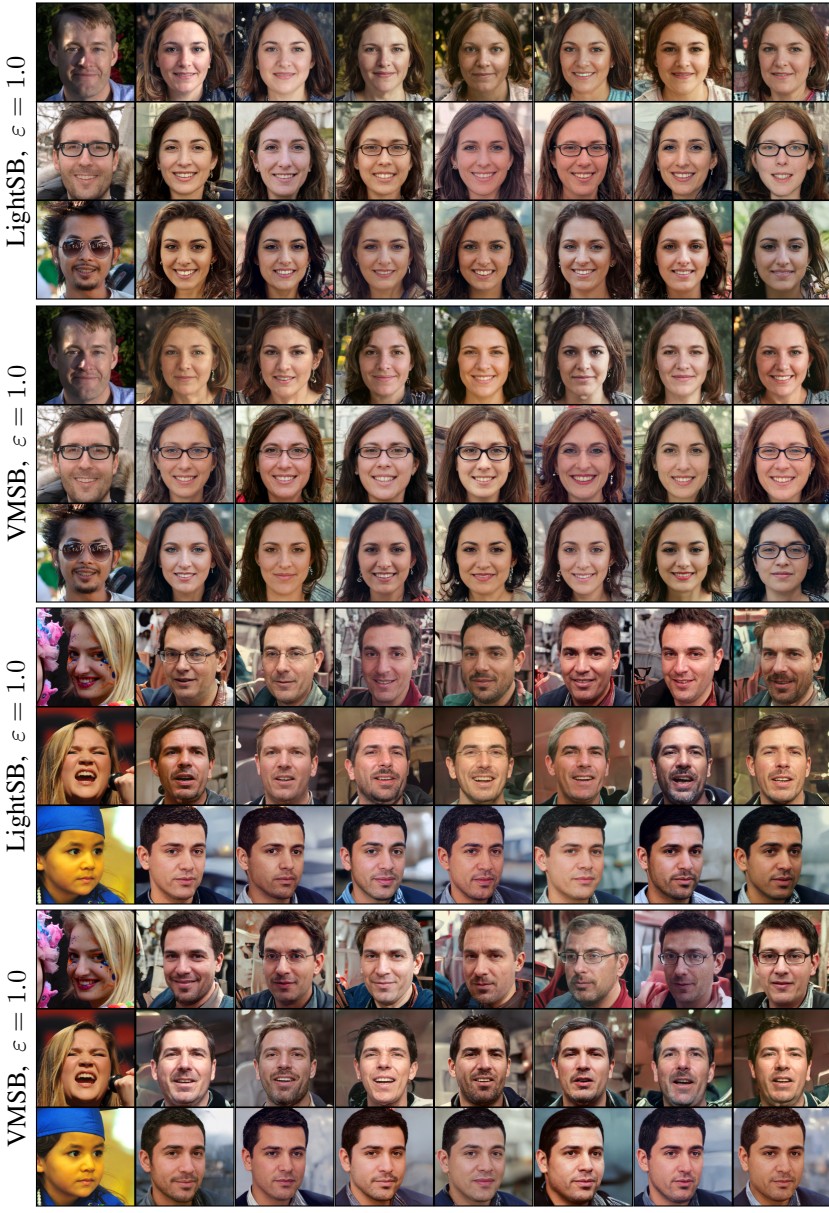

Figure 13: Qualitative comparison between LightSB and VMSB for relatively high volatility, $\varepsilon = 1.0$. Top (*Male → Female*): We find that VSBM has preserved more facial details, such as wearing glasses, than LightSB. Bottom (*Adult → Child*): VSBM was stable at retaining facial position even with high $\varepsilon$.

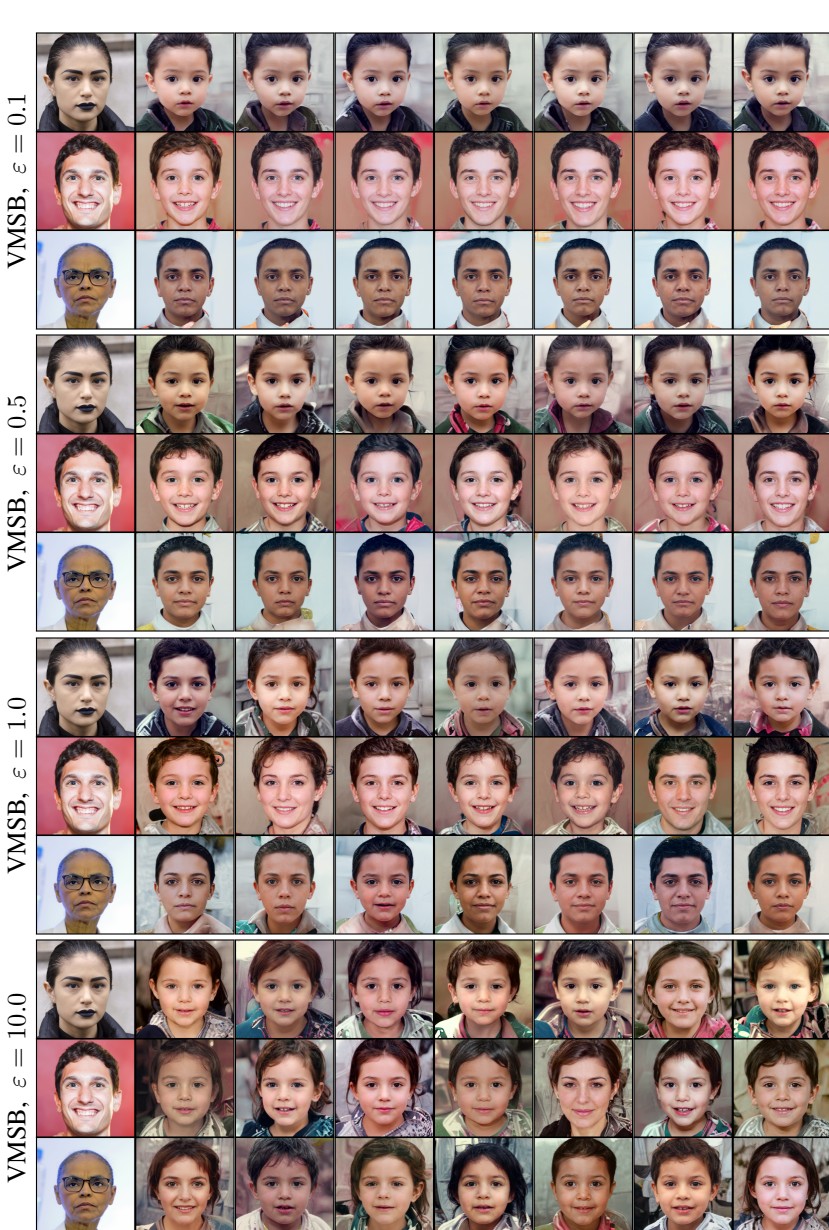

Figure 14: Generation results of VMSB (*Adult → Child*) with different volatility settings

