# OpenReview forum: "Variational Mirror Descent for Robust Learning in Schrödinger Bridge"
_ICLR.cc/2025/Conference — Submitted to ICLR 2025_

### Official Review · Reviewer_hbEJ · 2024-11-01

**Soundness:** 2
**Presentation:** 2
**Contribution:** 2
**Rating:** 5
**Confidence:** 4

**Summary:**

The paper proposes a new numerical solver for learning Schrodinger bridge, a new class of generative models that generalizes denoising diffusions. Specifically, the authors built upon the _known_ the connection between Sinkhorn/SB method and Mirror Descent, extending it to solving SB, while presenting strong theoretical results. An algorithm based on Gaussian mixture approximation was proposed and validated on 2D synthetic datasets, single-cell dataset, and latent-space image translation.

**Strengths:**

- Theoretical contributions are solid. I think many of their results in Sec 5 may as well be of interested for readers from other domains.

**Weaknesses:**

- On the theoretical side, most of the results in Sec 4 seem to be adopted from Karimi et al., 2024 [1]. Can the author clarify what's newly proposed in Sec 4?

- To me, the main contribution lies in Sec 5, where the author proposed an approximate algorithm for the Wasserstein flow using mixture of Gaussians. This line of "mixture-of-Gaussian approximation" was initially proposed as a (1) fast and (2) non-NN-based alternative to solving SB problems, at the cost of scalability to high-dimensional applications. However, neither advantages were highlighted in empirical evaluation. If the proposed method still requires NNs (for constructing $\pi_\phi$, I suppose, and surely in adv training), it's unclear why we should still consider mixture of Gaussian models.

- Following the previous point, I suspect this scalability of mixture Gaussian models is the reason why the empirical evaluation has been limited to benchmarks on 2D synthetic dataset and single-cell dataset, which, in my opinion, are no longer suitable for benchmarking SB methods (most methods yield visually indistinguishable learning results ...), and image translation yet on low-dim latent space or adversarial learning. Table 4 indicates the proposed method does not surpass DSBM-IMF. If a discriminator is required for training, it's unclear why we should still consider non-NN SB solver.

[1] Sinkhorn Flow as Mirror Flow: A Continuous-Time Framework for Generalizing the Sinkhorn Algorithm

**Questions:**

- What exactly is the difference between VMSB and VMSB-M algorithmically, the paragraph starting L445 is very unclear. Given their marginal performance differences, if not worse, I see no points of introducing VMSB-M except creating additional reading burden for reviewers.

- How exactly is $\phi$ being optimized? That one sentence in L394 seems overly simplified to me.

---

> ### Author Response · Authors · 2024-11-21
> **Official Response to Reviewer hbEJ (1/2)**
>
> We appreciate your helpful feedback and concerns. We address your comments below.
>
> **What's newly proposed in § 4?**
>
> Thank you for the question. Karimi et al. [1] introduced "a continuous-time analogue of the Sinkhorn algorithm" by leveraging the perspective of considering Sinkhorn as MD [2, 3]. They proposed γ-IPF formulation of MD, which particularly involves one of the marginals in SB (**Definition 1** of [1]). Overall, this can be understood as performing an algorithm akin to **Lemma 1,** but taking the limit $\eta\to0$.
>
> In contrast to [4], our work takes a distinct learning-theoretical approach. In **§ 4**, we constructed an online learning objective (**Lemma 2**) in a Wasserstein sense with a formal proof, allowing us to analyze convergence properties (**Theorem 1 & Proposition 1**) and a regret bound (**Theorem 2**). We would like to stress that conducting a precise analysis is important in learning theory. Moreover, we found that the variational principle can yield the OMD iterates in **Proposition 2**. To the best of our knowledge, the particular fact that the Wasserstein gradient flow of OMD in SB can be decomposed into a linear combination of KL functions is a novel result in the context of SB.
>
> We would like to emphasize that **§ 4** focuses on the essential mechanisms of SB learning and shows that many achievements in convex analysis can be translated into SB learning. Furthermore, **§ 4** establishes the fundamentals for VMSB, including the online cost function and learning via Wasserstein gradient flow. Consequently, **§ 4** is of critical importance in our paper and presents various novel ideas on the learning of SB models.
>
> **To me, the main contribution lies in § 5 ... If the proposed method still requires NNs, it's unclear why we should consider mixture of Gaussian models.**
>
> Thank you for your constructive feedback. As you mentioned, **§ 5** also makes significant contributions, particularly in implementing the Wasserstein gradient flow as described in **Proposition 2**. These contributions lie in enabling computation for performing OMD, thanks to the LightSB parameterization by Korotin et al. [5]. Our paper does not present strong arguments on the advantages or disadvantages of parametrization in the main text (we delineate these points in **Appendix D**), because the pros and cons of parametrization are extensively discussed in [5]. Our main arguments were to show consistent performance improvement from VMSB under the identical parameterization setting.
>
> In some of the experiments related to high-dimensional images, our algorithmic settings rely on discriminator and latent diffusion models for effective training. We would like to point out that the scalability limitations of GMMs in our cases are mostly problem-specific. Certain NN operations (e.g., convolution, U-Net) specialized for specific problems can help improve the SB solver’s performance. Once fully trained through interaction with NNs, we can efficiently sample from GMM-based SB models with competitive performance in generating MNIST data, achieving sampling speeds 1,854 times faster under the same hardware conditions.
>
> |  | GMM, K=64 | GMM, K=256 | GMM, K=1024 | GMM, K=4096 | NN (SDE)  |
> | --- | --- | --- | --- | --- | --- |
> | GPU | 721μs | 726μs | 739μs | 740μs | 1.372s |
>
> One of the distinctive advantages of the non-NN SB parameterization is that we can obtain SB samples without simulation inside an SDE, which reduces the time of sampling and avoids discretization errors from SDE solving. Therefore, we would like to emphasize that non-NN SB solvers can solve complex SB problems under certain conditions, and we believe GMM-based models still hold an irreplaceable role in theoretical and applied subfields of machine learning due to their simplicity and theoretical properties.

---

> > ### Author Response · Authors · 2024-11-21
> > **Official Response to Reviewer hbEJ (2/2)**
> >
> > **I suspect this scalability of mixture Gaussian models ... If a discriminator is required for training, it's unclear why we should still consider non-NN SB solver.**
> >
> > We appreciate your concern and partially agree with your point. However, we respectfully disagree to some extent, as we believe that EOT benchmarks and single-cell datasets are meaningful for numerically verifying our claims. Furthermore, we achieved comparable performance in unpaired image-to-image translation tasks, and we consider the diversity of our experiments to be a significant contribution.
> >
> > Although the LightSB parameterization we chose in the experiments might not be specialized for all cases, we would like to argue that studying non-NN SB solvers is important, as this parameterization often achieves great performance in a fraction of the time compared to SDE integration. Furthermore, GMM-based models still hold an irreplaceable role in numerous ML problems, such as Bayesian inference, latent diffusion, variational methods, and 3D rendering tasks, due to their distinctive properties. By successfully demonstrating two distinct ways of interacting with neural networks for solving unpaired image transfer, we also hope our theoretical and empirical findings will aid in architecture studies under the variational principle.
> >
> > **Q1. … I see no points of introducing VMSB-M.**
> >
> > Thank you for your detailed feedback. In response to your comment, we have enhanced the clarity of the **Baselines and VMSB Variants** section in the revised manuscript. We respectfully clarify that the algorithmic differences between VMSB and VMSB-M lie in **Line 4** of **Algorithm 1**; specifically, the target model of VMSB is trained using LightSB, whereas the target model of VMSB-M is trained using LightSB-M [6].
> >
> > We would like to emphasize that verifying consistent performance gains across various simulation-free SB solvers was crucial for validating our theoretical claims. Our variational OMD method operates via gradient dynamics of a divergence minimization problem for a target process without specifying the behavior of such a process, apart from a mild assumption (**Assumption 1**). Therefore, our theoretical arguments imply that the derived algorithm is agnostic to the choice of target models. The performance benefits from LightSB, LightSB-M, and LightSB-Adv equally strengthen the generality of our theoretical claims.
> >
> > **Q2 … How exactly is $\phi$ being optimized? ... L394 seems overly simplified.**
> >
> > As addressed in our previous comment, we acknowledge that we do not provide a strict argument on the exact optimization of $\phi$, as long as $\pi_\phi$ satisfies our assumptions. Consequently, we conducted experiments with various VMSB variants that differ in the training methods of \phi. In light of your insightful question, we have thoroughly revised **Line 394** and the surrounding paragraph to provide a more detailed explanation.
> >
> > Please feel free to reach out to us if you have further questions.
> >
> > Sincerely,
> >
> > Submission #5991 authors
> >
> > [1] Mohammad Reza Karimi, Ya-Ping Hsieh, and Andreas Krause. Sinkhorn flow as mirror flow: a continuous-time framework for generalizing the sinkhorn algorithm. In AISTATS, 2024.
> >
> > [2] Flavien Léger. A gradient descent perspective on Sinkhorn. Applied Mathematics & Optimization, 2021
> >
> > [3] Pierre-Cyril Aubin-Frankowski, Anna Korba, and Flavien Léger. Mirror descent with relative smoothness in measure spaces, with application to Sinkhorn and EM. Advances in NeurIPS, 2022.
> >
> > [4] Walid Krichene, Alexandre M. Bayen, and Peter L. Bartlett. Accelerated Mirror Descent in Continuous and Discrete Time. In NeurIPS, 2015.
> >
> > [5] Alexander Korotin, Nikita Gushchin, and Evgeny Burnaev. Light Schrödinger bridge. In ICLR, 2024.
> >
> > [6] Nikita Gushchin, Sergei Kholkin, Evgeny Burnaev, Alexander Korotin. Light and Optimal Schrödinger Bridge Matching, In ICML.

---

> > > ### Comment · Reviewer_hbEJ · 2024-11-25
> > >
> > > I thank the authors for their detailed reply. I acknowledge the clarity on the theoretical results and additional experiments. I still hold concerns and criticism on the use of discriminator in training, as it seems rather artificial and unrelated to what've been proposed theoretically. Adopting discriminator in training makes OT-based GAN a valid baseline and put doubts on the efficiency of the GMM models.
> > >
> > > Nevertheless, I update the score to 5 to reflect the current status, but note that I'd prefer a 4 if given such option.

---

> ### Author Response · Authors · 2024-11-26
> **Response to Reviewer hbEJ**
>
> Dear Reviewer `hbEJ`,
>
> We would like to sincerely thank you for your thoughtful feedback and earnest reassessment of our work. We are pleased to hear that our previous rebuttal has addressed your initial concerns regarding clarity and theory. We address your concern about the use of a discriminative network in training as follows.
>
> Overall, the experiments are designed to validate the generality of our claims across various OT-related ML problems. In particular, we observed that the current limitations of LightSB in image translation tasks at a certain level primarily can be learning-related issues rather than parameterization issues. Our VMSB demonstrated that the claimed theoretical properties of variational OMD can mitigate the mode collapsing issue considerably, which has also been considered one of the prominent learning-related issues in this field. We believe that proposing a practical adversarial algorithm that is “tolerant of unreliable objective estimation” is fundamentally connected to our central claim on learning theory, and is within our experimental scope.
>
> Thank you once again for your dedication and commitment. Please let us know if you have any further questions during the discussion phase.
>
> Best regards,
>
> Submission #5991 authors

---

### Official Review · Reviewer_vXhA · 2024-11-02

**Soundness:** 3
**Presentation:** 3
**Contribution:** 3
**Rating:** 8
**Confidence:** 3

**Summary:**

This paper proposed a novel variational online mirror descent algorithm for Schrodinger Bridge motivated by learning theory and justified by numerical experiments. Authors first provide a new perspective which formulates the MD as Wasserstein gradient flow using the SB cost. Based on this, authors further proposed a variational OMD for the SB problem.

**Strengths:**

This paper proposed a simulation-free variational OMD for solving SB problems based the Wasserstein-Fisher-Rao geometries of Gaussian Mixture. Rigorous theoretical results together with numerical results show this new method performs more robust with reduced uncertainties compared to previous methods.

**Weaknesses:**

I think the theoretical results, numerical results and presentations are all good.

**Questions:**

One advantage of this method is improved robustness and reduced uncertainty compare to other methods. From theoretical perspective, this is justified by the regret bound. Intuitively, this is due to the convexity of SB cost used for VOMD, compared to the non-convex nature of the other methods. I wonder if it's possible to show directly that the proposed method is better than other method under some situations?

---

> ### Author Response · Authors · 2024-11-21
> **Official Response to Reviewer vXhA**
>
> We sincerely thank you for your overall support of our submission. We address your question below.
>
> **Question. ... This is due to the convexity of SB cost used for VOMD, compared to the non-convex nature of the other methods. I wonder if it's possible to show directly that the proposed method is better than other methodsunder some situations?**
>
> As you mentioned, the distinct properties of the SB cost have significantly advanced our understanding of the MD-Sinkhorn relationship [1-3]. The properties are also highly important for our theoretical analysis of OMD improvement. In this paper, we claim that VOMD can be efficiently performed since infinitesimal variations of the Bregman divergence fundamentally resemble those of the KL divergences.
>
> While convex objectives are common in ML, non-convex learning is also prevalent in probabilistic generative models. In non-convex settings, general convergence guarantees are typically unavailable, and deriving regret bounds becomes considerably more challenging. Alternative notions, such as dynamic regret [4] and local regret [5], have been proposed to address these issues. However, adequate assumptions for regret analysis in the context of probabilistic generative models have not yet been rigorously established.
>
> To effectively show that our proposed method outperforms others under certain situations, we consider **Assumption 1** in our paper. This main assumption allows our algorithm to converge to an asymptotic dual mean even under nonstationary objectives. Although this approach involves an indirect comparison based on a theoretical postulate, the practicality and generality of this assumption are supported by the performance improvements observed in our experiments. Through a wide range of experiments, we demonstrated that the method can outperform existing methods and that the proposed assumptions hold for various situations.
>
> Thank you for your question. Please feel free to reach out if you have any further questions.
>
> Sincerely,
>
> Submission #5991 authors
>
> [1] Flavien Léger. A gradient descent perspective on Sinkhorn. Applied Mathematics & Optimization, 2021
>
> [2] Pierre-Cyril Aubin-Frankowski, Anna Korba, and Flavien Léger. Mirror descent with relative smoothness in measure spaces, with application to Sinkhorn and EM. Advances in NeurIPS, 2022.
>
> [3] Mohammad Reza Karimi, Ya-Ping Hsieh, and Andreas Krause. Sinkhorn flow as mirror flow: a continuous-time framework for generalizing the sinkhorn algorithm. In AISTATS, 2024.
>
> [4] Omar Besbes, Yonatan Gur, and Assaf Zeevi. Non-stationary Stochastic Optimization. Operations research, 2015
>
> [5] Elad Hazan, Karan Singh, and Cyril Zhang. Efficient Regret Minimization in Non-Convex Games. In ICML, 2017

---

### Official Review · Reviewer_7fDv · 2024-11-02

**Soundness:** 2
**Presentation:** 1
**Contribution:** 2
**Rating:** 5
**Confidence:** 3

**Summary:**

The authors present a practical implementation of the recent generalization [1] of Iterative Proportional Fitting (IPF) within the Mirror Descent (MD) framework for Schrödinger bridges (SB). This method utilizes a newly introduced Gaussian Mixture Model (GMM) parametrization for SB [2], enabling closed-form iterations in the MD process. The paper also includes a convergence analysis of the proposed algorithm.

**Post-rebuttal**. I keep my score as "borderline reject". The last authors' response to me answered my question 3, but I still have serious concerns regarding question 1. In fact, the newly provided results by the authors (nearest neightbors to generated samples in the train data), in my view, clearly indicate an overfit to data. Therefore, I really doubt the generative capabilities of the proposed GMM-based model. I feel a little bit like the authors try to sell something here which does not really work. I believe that this section in the paper may be misleading to the ML community and probably should be removed. If so, the paper should be judged by the rest results. The rest results are theoretically interesting, but it seems like in practice they lead to very minor improvements at the cost of increase of the computational time, so I think my score is fair.

**Strengths:**

The authors improve the convergence stability of the LightSB solver [2] by incorporating nonstationary estimates of conditional plans into the training objective, showcasing this enhancement through extensive experimental results. Additionally, they conduct some theoretical analysis of the proposed algorithm (provide a regret bound).

**Weaknesses:**

- *Clarity should be improved.* The paper’s overall structure is difficult to follow, and the notation is particularly challenging—especially compared to the clarity in the original works [1], [2], [5]. For instance, the authors use the sequence notation ${a_t}_{t=1}^\infty = {0, 1, \dots}$ to mean ${0, 1, 0, 1, \dots}$ instead of the standard sequence ${0, 1, 2, 3, \dots}$. Additionally, they reference equations that have not yet been introduced in the text. The manuscript also contains spelling and typographical errors, such as “OT problems with different volatility and .,” “impotence” instead of “idempotence,” and “bye the following arguments” instead of “by the following arguments.” Moreover, the definition of Maximum Mean Discrepancy (line 460) is incorrect: the formula introduces just the L2 distance between probability density functions. The part with adversarial training in the main text looks alien and it is not particularly clear how it fits to the proposed framework.

- *EMA baseline is omitted.* The authors appear to use smoothing via Mirror Descent (MD) in a way that resembles standard Exponential Moving Average (EMA) approach, which is a common trick in deep learning [6]. Given this, the paper lacks a comparison with EMA smoothing.

- *Theory is a combination of prior works.* One may argue that the contribution lacks substantial originality, as it primarily combines methods from three existing works [1], [2], [5]. Additionally, while the original work [1] suggests various adaptations for large-scale problems (e.g., image processing) in Appendix C, this paper opts for Gaussian Mixture Models, which, as highlighted in [2], face limitations when applied to large-scale scenarios. Furthermore, the connection between the theory of gradient flows and this natural smoothing concept is not clearly established.

- *Practical gain is not convincing.* While the authors claim applicability to high-dimensional problems, their demonstration is limited to images in a 512-dimensional latent space. Moreover, the improvement on the EOT benchmark [4] as well as in many other cases is minimal. At the same time, the convergence time as stated in Appendix D is about 10-30 minutes on GPU instead of few minutes on CPU as it was originally stated for LightSB [2]. Authors do not address exploring convergence time on different applications, which could be slower in real-life applications compared to carefully tuned optimizers like Adam [3].

**Questions:**

Please consider providing answers to the following questions:

- How could the authors justify the significance of improving the metrics in the considered experimental setups taking into account that this is achieved by a significantly increased computational time?
- Could you please provide a detailed experimental comparison with the standard EMA approach for smoothing the model training? Does the proposed clever smoothing really outperform the standard approach?
- Any models based on Gaussian mixtures are known to suffer from the mode collapse in the sense that only one component in the mixture is actually active. Does your method address this problem somehow?

I think clarifying these questions is of high importance because your method risks being overshadowed by baseline [2], which features an out-of-the-box minimization objective, operates with simplicity, and offers significantly faster computational speed.

**References**

[1] Karimi, Mohammad Reza, Ya-Ping Hsieh, and Andreas Krause. "Sinkhorn Flow as Mirror Flow: A Continuous-Time Framework for Generalizing the Sinkhorn Algorithm." In International Conference on Artificial Intelligence and Statistics, pp. 4186-4194. PMLR, 2024.

[2] Korotin, Alexander, Nikita Gushchin, and Evgeny Burnaev. "Light Schrödinger Bridge." In The Twelfth International Conference on Learning Representations.

[3] Kingma, Diederik P. "Adam: A method for stochastic optimization." arXiv preprint arXiv:1412.6980 (2014).

[4] Gushchin, Nikita, Alexander Kolesov, Petr Mokrov, Polina Karpikova, Andrei Spiridonov, Evgeny Burnaev, and Alexander Korotin. "Building the bridge of schrödinger: A continuous entropic optimal transport benchmark." Advances in Neural Information Processing Systems 36 (2023): 18932-18963.

[5] Lambert, Marc, Sinho Chewi, Francis Bach, Silvère Bonnabel, and Philippe Rigollet. "Variational inference via Wasserstein gradient flows." Advances in Neural Information Processing Systems 35 (2022): 14434-14447.

[6] Morales-Brotons, Daniel, Thijs Vogels, and Hadrien Hendrikx. "Exponential moving average of weights in deep learning: Dynamics and benefits." Transactions on Machine Learning Research (2024).

**Details Of Ethics Concerns:**

-

---

> ### Author Response · Authors · 2024-11-21
> **Official Reponse to Reviewer 7fDv (1/2)**
>
> We appreciate your constructive feedback and the time spent reviewing our work. Please check the revised manuscript reflecting your important suggestions as follows.
>
> - **(Notation & Clarity)** We agree that the enumeration style of $a_t$ is too brief. Following your suggestion, we enhanced the clarity of expressions in **L209**. Following fruitful suggestions, we also included additional theoretical details for clarity.
> - **(Maximum Mean Discrepancy)** We agree that the energy distance is a more appropriate term for our evaluation metric. We have unified the terminology to energy distance [1] and ED.
> - **(Typos)** Thank you for your detailed comments. We corrected spelling and typographical errors in the revised manuscript.
>
> We address your comments below. Please refresh the webpage if the latex expressions are not properly rendered.
>
> **EMA baselines.**
>
> Thank you for the insightful comment. We respectfully emphasize that MD works in more meaningful probabilistic spaces.
>
> - **The SB problem is convex for distributions.** We assumed that the expression $\lim_{t\to\infty} \mathbb{E}_t[\delta_C\Omega(\pi^\circ_t)]$ is stationary as a mapping of Gaussian mixtures (by consistently changing its mode) using Bregman potentials.
> - **The problem is non-convex to its parameters.** The exponential moving averaging (EMA) is the technique favors the Euclidean average of historical parameters $\{\phi_t\}^\infty_t$. It stabilizes the learning process, but analyzing its meaning in the probabilistic space could be difficult.
> -  Our OMD algorithm finds the best model that is closest to $\delta_D(\lim_{t\to\infty}\mathbb{E}_t[\delta_C\Omega(\pi^\circ_t)]$) where the Bregman potential is used to measure local distance. Therefore, we claimed that OMD brings theoretically pleasing outcomes regarding the statistical divergence induced by KL.
>
> To further validate our point, we added a row of LightSB-EMA in **Tables 6 & 7.** The table below presents the ε=1 results of cBW-UVP. This shows that VSMB consistently achieves better results than LightSB-EMA, and our theoretical argument rigorously supports these gains.
>
> |  | d=2 | d=16 | d=64 | d=128 |
> | --- | --- | --- | --- | --- |
> | LightSB | 0.014 | 0.026 | 0.060 | 0.140 |
> | LightSB-EMA | 0.012 | 0.022 | 0.051 | 0.127 |
> | VMSB (ours) | 0.010 | 0.018 | 0.044 | 0.114 |
>
> **The theory is a combination of prior works**
>
> We agree that our contributions are built upon the prior works on SB. We respectfully point out that our main contribution lies in the learning theory, and the significance our theoretical contribution can be to be justified by a fine level of convergence analysis. In **Theorems 1 & 2**, and also **Proposition 1**, we have provided novel convergence proofs of OMD in SB problems. We also established the fundamental connection between the variational principle and OMD update rules in the Wasserstein gradient flow, enabling our technical contribution to utilize the important theoretical framework from Lambert et al. [2].
>
> - Inspired by **Karimi et al. [3]**, we took a distinct theoretical learning approach. We constructed an online learning objective $F_t(\pi)$ and answered some important questions related to SB learning, such as conditions for ideal and general convergence (**Theorem 1 & Proposition 1**) and regret bounds (**Theorem 2**). Although the suggestions of Karimi et al. [3] for various possibilities of adaptations for large-scale problems in their appendix are promising, our work implemented the VMSB algorithm with real experiments, validating our theoretical analysis.
> - **Korotin et al. [4]** We would like to stress that we found a novel theoretical value of the LightSB parameterization by proposing its application in variational algorithms. Our analysis not only strengthened the mathematical rigor of the SB problem but also predicted a practical learning algorithm, which complements the contributions of [4].
> - **Lambert et al. [2].** We formally stated in **Proposition 2** that the OMD iterates can be attained by the variational principle. We discovered that the Wasserstein gradient flow of OMD can be decomposed into a linear combination of KL functions in SB problems. As a minor contribution, we applied a correction to the ODE system equation in § 6 of [2], allowing mixture weights ($\alpha_{k,\tau}$; $r^{(i)}_t$ in the notation of [2]) to correctly change their weights according to the geometry.
>
> **Practical gain is not convincing.**
>
> We respectfully remind you that our experiments were designed in a highly controlled setting with identical model configurations. Our primary goal was to verify our theory and method, which we achieved by surpassing LightSB [4] and LightSB-M [5] under identical parameterization conditions and numbers of parameters. With all other aspects controlled and outcomes determined solely by learning methods, the consistent performance gains of our work were an well-anticipated result from our theoretical analysis.

---

> ### Author Response · Authors · 2024-11-21
> **Official Response to Reviewer 7fDv (2/2)**
>
> **Adversarial training**
>
> Adversarial training has provided a powerful baseline in the MNIST-EMNIST translation tasks, which other LightSB algorithms cannot match. However, like GANs, the mode collapse phenomenon still presents a practical issue. From our theory, we considered that one cause of this problem is the unstable learning signals due to limited discriminative capability. In the adversarial training experiment, the proposed VMSB variant “VMSB-adv” showed a clear advantage since constrained OMD updates govern the progress of the adversarial training. This point is strongly related to our theoretical claims of convergence and is predicted by the assumptions.
>
> **Q1**
>
> Thank you for the question. We respectfully highlight that we have substantially reduced the amount of required computation for OMD by using the LightSB parameterization and variational inference, and we have verified its usefulness through experimental results.  Since the algorithm can be performed more efficiently than expected, the proposed VMSB obtained strong numerical results, especially in low- to mid-range generation problems with relatively low computational time.
>
> For high-dimensional problems, the computation of WGF may become more parallel, and we essentially favor a GPU for training. Once it is fully trained, we can deploy and utilize the model on CPUs and GPUs, which can be applied efficiently to various applications, considering its complexity. Here, we also report that collecting SB samples from 4096 Gaussian particles, which are equipped with competitive performance, can be done 1,854 times faster under the same hardware, and we can also generate samples with a CPU.
>
> |  | GMM, K=64 | GMM, K=256 | GMM, K=1024 | GMM, K=4096 | NN (SDE)  |
> | --- | --- | --- | --- | --- | --- |
> | GPU | 721μs | 726μs | 739μs | 740μs | 1.372s |
> | CPU | 60.140ms | 133.333ms | 428.433ms | 1.527s |  |
>
> Note that this generation process does not suffer from discretization errors of SDE.  Although the LightSB parameterization might suffer from very high-dimensional problems, we believe GMM-based models still hold an irreplaceable role in practice due to their simplicity and theoretical properties. As our primary intention was to validate our theoretical claims on learning, we hope that our experiments could help the training of non-NN models that has potential to be used in theoretical and applied ML problems
>
> **Q2**
>
> Following your suggestion, we reported experimental results with EMA above. We would like to summarize our method from both perspectives as below.
>
> - **Optimization for “high-level” SB probability distributions.**
>     - Current SB models can be seen as solving a convex optimization problem. We claim that this does not fully guarantee stability in SB learning.
>     - We propose VMSB, which performs online MD updates utilizing the variational approach for computationally efficient and theoretically explainable learning processes.
> - **Optimization for “low-level” parameters with EMA.**
>     - We do not specifically argue for or against training using optimizers and other parametric techniques.
>     - **VMSB.** VMSB optimizes SB along with the WFR gradients using an optimizer such as Adam, but the proximal local target of learning is the MD updates at the high-level probabilistic perspective, which makes the policy learning more robust.
>     - **EMA.** Even if the SB updates are smoothed, EMA might have practical issues in high-level distributional spaces, and this approach can be unreliable depending on the problems, as there are no theoretical guarantees in such spaces.
>
> **Q3**
>
> As mentioned earlier, the mode collapsing phenomenon was one of the considerations while contemplating Assumption 1. Our theory finds the best SB model $\pi_\theta$ that is close to $\delta_D \Omega^\ast(\lim_{t\to\infty}\mathbb{E}_t[\delta_C \Omega(\pi{\phi_t})])$, where the first variations of Bregman divergence are used for mappings between primal and dual spaces. In our theoretical framework, we claim that the OMD framework yields favorable outcomes by covering all modalities through the statistical divergence induced by $\Omega$.
>
> Please let us know if there are any remaining questions or comments.
>
> Sincerely,
>
> Submission #5991 authors
>
> [1] Maria L Rizzo and Gábor J. Székely. Energy distance. Wiley interdisciplinary reviews: Computational statistics, 2016
>
> [2] Marc Lambert, Sinho Chewi, Francis Bach, Silvère Bonnabel, and Philippe Rigollet. Variational inference via Wasserstein gradient flows. In NeurIPS, 2022.
>
> [3] Mohammad Reza Karimi, Ya-Ping Hsieh, and Andreas Krause. Sinkhorn flow as mirror flow: a continuous-time framework for generalizing the sinkhorn algorithm. In AISTATS, 2024.
>
> [4] Alexander Korotin, Nikita Gushchin, and Evgeny Burnaev. Light Schrödinger bridge. In ICLR, 2024.
>
> [5] Nikita Gushchin, Sergei Kholkin, Evgeny Burnaev, Alexander Korotin. Light and Optimal Schrödinger Bridge Matching, In ICML.

---

> > ### Author Response · Authors · 2024-11-27
> > **Kind reminder**
> >
> > Dear Reviewer ` 7fDv`,
> >
> > Thank you for your insightful and constructive feedback on our submission. Your comments have been invaluable in helping us improve our work. We hope that our careful response and revised manuscript address many of your concerns. If you require any additional clarification or have further questions, please do not hesitate to contact us. Thank you once again for your time and effort.
> >
> > Best regards,
> >
> > Submission #5991 authors

---

> > > ### Comment · Reviewer_7fDv · 2024-11-29
> > > **Response to the Authors**
> > >
> > > Dear Authors,
> > >
> > > Thank you for your thorough and detailed responses. I apologize for some delay in replying. While I am satisfied with your answer to Question 2, I still have concerns regarding Questions 1 and 3.
> > >
> > > **Question 1.** Analysing your answer about the time, I came up with additional questions, specificially about the adversarial training setup. Training GMM models with GAN looks a little bit like an overkill and I do not understand the actual usability of this in practice. It is questionable whether the GMM can learn some distributions supported on low dimensional manifolds without **overfitting** to data. Looking at the resulting samples generated by your method, it looks like it simply overfits to data and just puts mean Gaussian centers to the data points. For example, in Figure 7 VMSB-adv, I noticed identical images of the small letter “i” in row/column (4,2), (4, 3), capital letter “H” in (5, 8) and (8, 5) apeear to be the same; same apllies to letter "l" in (4,3) and (4,4), letter "j" in (4,10), (5,1), (6,2), letter "b" in (8,6) and (10,6), letter "l" in (3,3) and (10,9), "l" in (2,1) and (2,2). Given that, I think the result of the experiment is questionable and raise the concern whether the method truly generalizes or just overfits.
> > >
> > > Could the auhtors please provide a demonstration (e.g., via anonymous link) of the generated samples together with their nearest neighbors from the train dataset (say, 100 random samples)? If it overfits, then I do not see the point in the proposed method as Discrete OT can do the same but much faster.
> > >
> > > **Question 3.** Your response on this matter seems a little bit general and vague. Does your method avoid **mode collapse** or not? By "mode collapse" in this particular question I mean the GMM collapsing into a single Gaussian (but not the issue of producing the same samples/overfitting as above).
> > >
> > > Could you please provide a quick experimental test and report $K$ values $\{\mathbb{E}\_{x \sim \mu} [\alpha_k^x]\}_{k=1}^K$ (expectation is taken other the initial marginal) for the ALAE setup? This values show the average probability of sampling from each GMM component.

---

> ### Author Response · Authors · 2024-12-03
> **Response to Reviewer 7fDv (Discussion, 1/2)**
>
> Dear Reviewer `7fDv`,
>
> We appreciate for your thoughtful feedback and the positive remark on our response to **Question 2**. We further address **Questions 1 & 3** below. As requested, we provide an anonymous link containing a single-page PDF file that addresses one of your comments: [[anonymous link](https://anonymous.4open.science/r/iclr2025_submission5991_reubttal-810B/)].
>
> **Question 1**
>
> In addition to the exceptionally fast generation time of simulation-free samples, we respectfully emphasize the importance of VMSB-adv experiments due to the various practical implications in the image domain.
>
> - **Table 9** in the appendix presents the FID scores on both the training and testing datasets. The performance difference demonstrate that VMSB-adv suffer less from overfitting compared to its counterpart LightSB-adv, across all parameterization settings. This supports our claims regarding the stability of SB solution acquisition in pixel spaces.
> - We acknowledge the limitation on GMM architecture as noted in **L1664-L1672**.  Due to the architectural limitation, training GMM-based SB models tends to focus on instance-level associations within high-dimensional problems, rather than the sub-instance or feature-level associations intrinsic to deep neural network architectures. Consequently, the generated letters may appear synthetic, yet the proposed approach produces statistically sound pixel representations within the given constraints. We hope that the proposed OMD theory will inspire future studies across various domains, including novel neural architecture research.
> - We have included a demonstration of generated samples with their neighboring images from the training set based on MSD via the anonymous [[link](https://anonymous.4open.science/r/iclr2025_submission5991_reubttal-810B/)]. The figure shows that VMSB-adv demonstrates robust EOT images which are coherent to input and top-5 similar images. We would like to stress that the proposed method does not favor strict memorization of exact pixel positions and values of the training set, since such deterministic behavior does not meaningfully reduce adversarial loss throughout the time.
> - Additionally, we present quantitative results comparing MSD metrics with 500 samples, respectively.
>
> |  | Top-1 (nearest) | Top-5 | All (EMNIST) |
> | --- | --- | --- | --- |
> | LightSB-256 | $0.02925\pm.03519$ | $0.05840\pm.05860$ | $0.60312\pm.18710$ |
> | LightSB-1024 | $0.04136\pm.05516$ | $0.08247\pm.06807$ | $0.58987\pm.17217$ |
> | LightSB-4096 | $0.03696\pm.04025$ | $0.08470\pm.05691$ | $0.60489\pm.17568$ |
> | **VMSB-256** | $0.02053\pm.04380$ | $0.05964\pm.06833$ | $0.57130\pm.18103$ |
> | **VMSB-1024** | $0.01364\pm.02973$ | $0.07193\pm.06599$ | $0.58363\pm.17218$ |
> | **VMSB-4096** | $0.01845\pm.04044$ | $0.06614\pm.05592$  | **$0.58617\pm.17011$** |
>
> - We identified two key trends: (1) Samples generated by VMSB-adv consistently produce more representative images within local groups compared to LightSB-adv, as observed by improved Top-1 and Top-5 similarity metrics, and (2) VMSB-adv does not show signs of overfitting, and maintains a certain level of Top-1 similarity (VMSB-1024 -> VMSB-4096). Also, a substantial decrease in Top-5 similarity is observed when scaling from VMSB-1024 to VMSB-4096. These trends suggest that VMSB-adv enhances the stability of the learning process by aligning the modalities of distribution within local groups of multiple images, and the discriminator plays a central role in determining the convergence point. Consequently, our approach is less likely to reproduce identical instances from the data source, addressing the concern you raised.
>
> - **Learning vs. overfitting**. Similar to other ML models, overfitting might happens when K is very high due to (1) the discrete nature of the dataset in the pixel domain and (2) the models’ sufficient capacity to represent all the data, potentially leading to overly complex model structures. Since the learning of SB models is closely related to the instance-level discovery of optimal couplings, some degree of memorization occurs when data pairs have clear associations. Informally speaking, full memorization would require setting $K^\ast =(\textrm{Number of images in} \ \mu) \times (\textrm{Number of images in} \ \nu)$ to ensure allocation of such associations. Therefore, we strongly expect generalization in SB learning for the GMM-based models, as $K\in$ {256,1024,4096} is relatively small compared to $K^\ast$.

---

> > ### Author Response · Authors · 2024-12-03
> > **Response to Reviewer 7fDv (Discussion, 2/2)**
> >
> > **Question 3**
> >
> > **Mode collapse.** We would like to clarify that the proposed VMSB mitigates mode collapse by efficiently allocating its capacity.  We believe our argument can be better understood in the context of GAN studies, especially the *historical averaging* proposed by Salimans et al. [1]. Historical averaging is an advanced GAN technique that penalizes a model with a loss term when it deviates from its historical time average. Following your insightful comment, we would like to rearrange our argument with the following intuitive example:
> >
> > - Suppose that the ground truth SB $\pi^\ast$ is represented (by the universal approximation property [2]) as a mixture of $K^\ast$ Gaussians, i.e., $\vec{\pi}^\ast(y\vert x) = \frac{1}{K^\ast}\sum_{k=1}^{K^\ast}\mathcal{N}(y\vert m^x_k, \varepsilon\Sigma_k)$. Suppose that an estimation process $\pi^\circ_{t} \in \Pi$ does not fully converge because $K \ll K^\ast$, and the process specifically suffers from mode collapsing, which is stochastically selecting only a handful of modes at a time.
> > - We can find some $a>0$ such that $D_\Omega(\delta_D( \mathbb{E}{\scriptstyle t}[\delta_\mathcal{C}\Omega(\pi^\circ_t)]) \Vert \pi^\ast) < a$ for large $t$. In some optimal training regime, as $t\to\infty$, the limit $\pi^\circ_\mathcal{D} \coloneqq \delta_D( \lim_{t\to\infty}\mathbb{E}{\scriptstyle t}[\delta_\mathcal{C}\Omega(\pi^\circ_t)])$ can conver $\pi^\ast$ (i.e., $a\to0$) by **randomly changing modalities** of $\pi^\circ_{t}$ throughout the iteration, even though $K$ is fairly limited.
> > - Following our analysis, the SB distribution $\pi_t$ will converge to $\pi_\mathcal{D}$ by choosing an appropriate step size scheduling method for $\lbrace\eta_t\rbrace_{t=0}^\infty$.
> > - $\pi_\mathcal{D}$ is the best possible estimate derived from
> > $\lbrace\pi^\circ_t\rbrace_{t=0}^\infty$ over a long time interval, whose underlying argument is similar to those of historical averaging and EMA techniques.
> >
> > One of the important results in **§ 4** is that the regret is bounded by $O(\sqrt{T})$. Therefore, our claim is that the proposed update rules optimally allocate the limited capacity of SB models, embracing the discriminator's suboptimal learning signals by considering the entire time span. Furthermore, as we discussed in **Question 2**, OMD works more meaningfully than historical averaging in SB learning.
> >
> > - By the previous argument, $\pi^\circ_\mathcal{D}$ resembles $\pi^\ast$ and can achieve greater capacity than mode collapsed $\pi^\circ_t = \pi_{\phi_t}$.
> > - **Historical averaging on $\lbrace\phi_t\rbrace_{t=0}^\infty$.** This technique favors the average of historical SB parameters. It stabilizes the learning process like EMA, but analyzing its meaning in the probabilistic space is difficult.
> > - **OMD of $\pi_\theta$ for the target sequence $\lbrace\pi_{\phi_t}\rbrace_{t=0}^\infty$.** The OMD algorithm finds the best $\pi_\theta$ that is closest to $\pi^\circ_\mathcal{D}$, where the generalized Bregman divergence is used to analyze closeness. Therefore, we claim that our OMD theory yields favorable outcomes regarding the statistical divergence induced by the KL functional; hence, $\pi_{\theta}$ will cover important modalities under its architectural constraints.
> >
> > We respectfully emphasize that our learning-theoretic perspective is novel in the SB research and is supported by actual experimental results.
> >
> > **ALAE.** For each ALAE setup, number of modality in the GMM is fixed to $K=10$ following [2], which can be found in the official LightSB code. We report comparison of 5 significant values of $\lbrace\mathbb{E}_x [\alpha^x_k]\rbrace _{k=1}^K$, the average probability of sampling and also the average entropy $\mathrm{MeanEnt} \coloneqq \mathbb{E} _{x\sim\mu}[\sum _{k=1}^K -\alpha^x_k\log\alpha^x_k]$ (Child → Adult, ε=1.0) as the following table.
> >
> > |  | 1 | 2 | 3 | 4 | 5 | MeanEnt |
> > | --- | --- | --- | --- | --- | --- | --- |
> > | LigtSB | 0.218 | 0.216 | 0.200 | 0.188 | 0.177 | 0.11215 |
> > | **VMSB** | 0.241 | 0.199 | 0.195 | 0.182 | 0.182 | **0.11569** |
> >
> > The VMSB method exhibits a higher value for the most significant $\mathbb{E}_{x \sim \mu} [\alpha_k^x]$, while other values are more evenly distributed compared to LightSB. Additionally, VMSB demonstrates a higher overall average entropy, indicating a more balanced utilization of model capacity. This suggests that VMSB can be more efficient based on each GMM modality, and the significance of the mean entropy measure can be evaluated in the context of the methods' performance.
> >
> > We hope these answers adequately address your questions. Thank you for your question and your dedication to this discussion.
> >
> > Sincerely,
> >
> > Submission #5991 authors
> >
> > [1] Tim Salimans, Ian J. Goodfellow, Wojciech Zaremba, Vicki Cheung, Alec Radford, and Xi Chen. Improved techniques for training GANs. CoRR, abs/1606.03498, 2016. URL http://arxiv.org/abs/1606.03498.
> >
> > [2] Alexander Korotin, Nikita Gushchin, and Evgeny Burnaev. Light Schrödinger bridge. In ICLR, 2024.

---

### Official Review · Reviewer_iSPa · 2024-11-03

**Soundness:** 3
**Presentation:** 2
**Contribution:** 3
**Rating:** 8
**Confidence:** 3

**Summary:**

The paper introduces a novel approach to solving Schrödinger bridge (SB) problems in probabilistic generative modeling by proposing the Variational Mirrored Schrödinger Bridge (VMSB) algorithm. The experimental results show the VMSB algorithm consistently outperforms prior SB solvers in terms of stability and convergence across multiple generative modeling benchmarks.

**Strengths:**

1.It is a novel framework which reformulates SB problems using variational principles and online mirror descent (OMD). This results in a robust, simulation-free algorithm with theoretical guarantees.
2.The theoretical is solid. The paper establishes a convergence and regret bound for their OMD method, showing that it attains a regret bound, signifying efficient learning dynamics.
3.The experiments empirically validate VMSB’s performance in various tasks, including synthetic data generation and high-dimensional datasets, highlighting its robustness over existing methods.

**Weaknesses:**

1.The paper is heavily focused on theoretical descriptions, which can make it challenging to follow.
2.The WFR gradient decent algorithm is relatively simple; incorporating higher-order methods could enhance the efficiency of inner iterations.

**Questions:**

I am open to raising my rating if the authors can address the following question:
1. Could you conduct experiments to demonstrate the efficiency of this simulation-free model compared to traditional models? For instance, showing a comparison of time versus performance metrics would be helpful.
2. Could you give more detailed descriptions of the models used for each experiment?

---

> ### Author Response · Authors · 2024-11-21
> **Official Response to Reviewer iSPa**
>
> We greatly appreciate your insightful feedback and thank you for the encouraging comments. We address your concerns below.
>
> **Heavy theoretical descriptions**
>
> Thank you for your constructive feedback. In response, we have enhanced the clarity of the theoretical descriptions in the manuscript by providing additional information and detailed explanations. While our approach is theoretical, we believe that a rigorous method offers a unique contribution toward better understanding for the target problem and fundamentally closing the gap between theory and practice. Additionally, we respectfully highlight that we have made considerable efforts to provide helpful resources and intuitive figures, and we have conducted a wide range of experiments in the SB domain to strengthen our claims.
>
> **WFR gradient**
>
> We appreciate your insightful comment. We would like to emphasize our contributions to the computational aspects of OMD using first variations. We intentionally designed **Algorithm 1** to be a simple first-order method in parameterized SB models in order to verify our theoretical claims through experiments in a straightforward manner. Nevertheless, we have demonstrated that our algorithm can model complex distributions and solve various online learning problems. As you suggested, one can extend the efficiency of OMD with higher-order methods. However, technical limitations might arise since we did not assume high-order moments to be finite in $\mathcal{C}$ throughout the analysis. Therefore, a different theoretical approach would be required for a higher-order extension of OMD.
>
> **Q1 [efficiency]**
>
> Driven by parallel nature of Gaussian particles, we observed that the computation of **Proposition 2** favors vectorized instructions, and the expected speed enhancement from using GPUs is much more evident in the NN case. In the following table, we report the wall-clock time for a 100-dimensional single-cell data problem [1,2,3], where the performance is reported in **Table 3**.
>
> | Sinkhorn (IPF) | LightSB | VMSB |
> | --- | --- | --- |
> | 8m (GPU) | 66s (CPU) | 32s (GPU) / 22m (CPU) |
>
> Additionally, training time in the MNIST-EMNIST translation is reported in **Table 8** in the appendix**.** We believe this property also holds for sample generation, allowing us to deploy the model much faster on GPUs. Here, we also report that generating 100 MNIST samples from 4096 Gaussian particles, equipped with competitive performance, can be done 1,854 times faster under the same hardware.
>
> |  | GMM, K=64 | GMM, K=256 | GMM, K=1024 | GMM, K=4096 | NN (SDE)  |
> | --- | --- | --- | --- | --- | --- |
> | GPU | 721μs | 726μs | 739μs | 740μs | 1.372s |
>
> Note that the GMM generation process does not suffer from discretization errors of SDE. As we stated in **Appendix D**, we believe GMM-based models still hold an irreplaceable role in practice due to their simplicity and parallel nature.
>
> **Q2 [model details]**
>
> We used the LightSB parameterization [1] with diagonal GMMs. In response, we provide each modality $K$ for benchmarks in the following table.
>
> | K=10 | K=50 | K=100 |
> | --- | --- | --- |
> | **MCSI**, **ALAE** | **EOT** | **EOT** (d=128) |
>
> The rest of the step size scheduling and trading details are stated in **Appendix C** and can be found in setting files of the provided code.
>
> Please let us know if there are any further questions or comments.
>
> Sincerely,
>
> Submission #5991 authors
>
> [1] Francisco Vargas, Pierre Thodoroff, Austen Lamacraft, and Neil Lawrence. Solving Schr¨ odinger bridges via maximum likelihood. Entropy, 23(9):1134, 2021.
>
> [2] Alexander Korotin, Nikita Gushchin, and Evgeny Burnaev. Light Schrödinger bridge. In ICLR, 2024.
>
> [3] Nikita Gushchin, Sergei Kholkin, Evgeny Burnaev, Alexander Korotin. Light and Optimal Schrödinger Bridge Matching, In ICML.

---

### Official Review · Reviewer_FTE8 · 2024-11-04

**Soundness:** 3
**Presentation:** 3
**Contribution:** 3
**Rating:** 8
**Confidence:** 2

**Summary:**

This work builds on the recent connection between numerical algorithms for solving Schrodinger Bridge (SB) / EOT problems (Sinkhorn / IPF) and Mirrror Descent (MD).

The focus is on the online setting, where data becomes available as a stream, hence the extension to Online Mirror Descent.

Theoretical results regarding the proposal, under an idealized and more realistic settings, are provided in Section 4.2, while the computational and algorithmic implementation aspects are covered in Sections 4.3, and 5.1-5.2.

As noted, the confidence in my assignment is low (i.e. it is difficult for me to accurately judge the significance of the contributions), due to my limited familiarity with various prerequisites.

**Strengths:**

I enjoyed the focus on MD, i.e. equation (6), as a mean to derive existing and novel algorithms, which complements the explorations of Karimi et al, and further advances the field in this research direction.
The experimental results shows improvements across the tested scenarios, and are complemented by theoretical convergence results.
The implementation of the proposed algorithm rests on connections with Wasserstein gradient flow (specifically, in the setting of Gaussian measures), which are of interest in their own.
The writing style is formal and for the most part everything is clearly and precisely defined.

**Weaknesses:**

My main criticism of this work concerns the fact that it consists of a very dense presentation.
The paper covers a lot of ground, with topics ranging from the classical static and dynamics formulation of SB, to mirror descent (and related concepts for the infinite-dimensional version considered), to connections between SB and mirror descent, to Wasserstein flows and Otto's calculus.

I am well aware that the 10-pages limit makes it difficult to properly introduce all these concepts.
At the same time, unless the reader already has a (more than passing) familiarity with all the mentioned concepts, the paper and its contributions are quite difficult to parse, as it will become evident from my questions in the following section.
In particular, I found it difficult to identify the fundamental components making up the proposed methodology and its concrete implementation (i.e. what absolutely should be understood), and what main assumptions / approximations stand behind them.

My personal suggestion, would be to delegate (or shorten) some parts currently present in the manuscript to the appendices (on a side note: the appendices already contain quite a lot of interesting "introductory" material, a fact that is not so evident while reading the main text), that would not excessively impact the presentation, such as (these are just some possibilities):
- (1) and the part surrounding (1), which remains too abstract
- The results of 4.2 could maybe be stated as a single  informal (in the sense not detailing all the required conditions, with the current version moved to the appendix) Theorem
- The LightSB specific formulation (in practice it is a Gaussian mixture parametrization of the conditional static transport map)
- Proposition 2 as well (the specific form that enters (14) does not seem strictly necessary for understanding)

And use this space to better highlight key aspects of the proposal, and required concepts necessary to understand it.
I am not advocating turning the main text to an "Executive Summary", but I feel some improved balance can be found with the goal of expanding the intended target audience.

**Questions:**

To start with, I think I managed to follow, with some questions, the developments up to Lemma 2. Regarding this part:

1. I could not completely make sense of Figure 1. Is it meant to convey that the IPF iterations match the marginal distributions one at a time, while the proposed method lives somewhere else in the space 𝒞, while still converging to the EOT solution? (In this case, why is π_0 for IPF not on one of the red/blue curves)?

2. Relatedly: I was surprised that in Lemma2 the dynamics for π_t are expressed in terms of the conditional distribution π_t(•|x), should the LHS of the stated limit also be the conditional version? Does the stated result also result in an ODE flow limit as in (Karimi et al. ,eq (13))?

3. Equation (6) plays a central role, with different F_t resulting in different algorithms. In all cases, the proximity term remains fixed. Is there a fundamental intuitive reason for this? Can the authors envision further applications of (6) that would be relevant for SB problems?

4. (Minor): I got tricked for a moment into a completing the sequence α_t as {0, 1, 2, ...} around line 209.]

5. If I understood correctly, there are two challenges into applying Lemma2: how to implement the Wasserstein Gradient flow, and we do not have access to π*. Is this correct?

6. While I agree that "However, learning methods of SB remain somewhat atypical, each requiring a sophisticated approach to derive solutions.", some progress in that direction has been achieved in [1], [2]. While these works do not target the streaming setting specifically, they seem more suitable than iterative methods such as IPF, so they might be a reasonable baseline to show improvements upon?

Section 4.2 onward was more difficult to parse.

7. In Section 4.2, the idea is to substitute for π* with an (evolving) estimate π˚_t, and in this section assumptions are given regarding this approximating process such that, with some choice of step-sizes, convergence still holds....can the authors expand a bit on the high-level idea of the approach and on the result? What is the reference estimation fitted using a Monte Carlo method, is it π˚_t? To what extent can the assumptions be verified when applying the proposed method?

8. In Section 4.3, what is ρ_t? Is ε_t() given by f() in Lemma2 with π˚_t instead of π*? So the relevance of (8) is that, applied to Lemma2 with π˚_t  allows us to leverage Theorem 3's characterization, that then yields Algorithm 1 by leveraging the explicit form of (14) that is due to the use LightSB?

9. Is the approximation in (14) due exclusively to the use of the Mixture of Gaussian coupling parametrization from LightSB?

10. The method, like LightSB, centers around the static setting. Can the results of this work also yield some insights for methods employing neural networks to solve the dynamic version of SB, or is it possible to envision extensions of the present work in that direction?

11. What is the author's intuition on the superior performance of the proposed method in the non-streaming EOT benchmark? Is it because the proposed algorithm results in some kind of smoothing in the descent iterates? Would employing EWMA (exponentially weighted moving average scheme) for the parameters' updates in LightSB yield a similar effect?

[1] Schrödinger Bridge Flow for Unpaired Data Translation (https://www.arxiv.org/abs/2409.09347)

[2] BM2: Coupled Schrödinger Bridge Matching (https://www.arxiv.org/abs/2409.09376)

---

> ### Author Response · Authors · 2024-11-21
> **Official Response to Reviewer FTEB (1/3)**
>
> We greatly appreciate the time and effort you have invested in reviewing our submission. We are currently incorporating all of the invaluable feedback in the manuscript, and we are pleased to report that many of your suggestions are used in the current revision. We address your comments below.
>
> **Suggestions**
>
> Following your fruitful suggestions, we have carefully revised the manuscript. The clarity of the manuscript has improved significantly from the submission, thanks to your insightful feedback. We summarize the improvements below:
>
> - **(§ 1)** We have enhanced the clarity of sentences around Eq. (1), improving the focus on optimization and stability.
> - **(§ 4.2)** We overhauled **L288** and **L296-L302** to explain theoretical results in a less formal tone; some important formal parts of the current sentences have been moved moved to the end of **Appendix A.4**.
> - **(§ 5.1)** We added an opening sentence to describe the LightSB formulation. For more technical details for LightSB, we also added **Appendix C.1**, which provides theoretical and technical rationales for our choice of the GMM parameterization.
> - **(§ 5.2)** We enhanced the clarity of writing by including an explicit sentence regarding the algorithmic implication of the proposition.
>
> **Dense presentation style**
>
> We appreciate your insightful feedback. Starting from **Eq. (6)**, we devised a variational online mirror descent (OMD) algorithm (**Algorithm 1**) for the derived update rule **Eq. (10)**, using the tractable form of **Eq. (10)**.  We agree that the sheer volume to reach our algorithmic decision could be overwhelming to readers who are not familiar with the subjects. We would like to highlight our contributions and provide the context below.
>
> **Contributions & Key aspects of the proposal**
>
> The contributions of this paper stated in **L89-L94** can be explicitly delineated as follows.
>
> - **A new OMD method with strong theoretical guarantees.** This paper provides a fine level of convergence analysis of an online learning problem in SB with a rigorous regret bound.
> - **Simulation-free variational SB solver.** Compared to contemporary SB algorithms, the proposed VSMB is well-grounded and can work under various real-world circumstances.
> - **A computational SB method validated in a wide range of benchmarks.** Our work successfully verified the MD framework in SB and corresponding theoretical claims with experiments.
>
> In a less formal tone, we respectfully emphasize the following points.
>
> - **Solving SBP with learning theory.** Inspired by prior studies, our study casts a fresh geometric perspective to minimize online costs. We effectively show that each learning step of SB can be precisely taken analytically, with a gradient-based algorithm.
> - **New gradient-based SB algorithm.** We show a new learning method for SB, which is drawn from variational inference. Applying the proposed gradient dynamics to the LightSB parameterization, this technique effectively outperforms prior methods in our experiments.
> - As a minor contribution, we believe that readers can better understand the EOT/SB problem with a broader perspective from our study.
>
> **Fundamental & required concepts for methodology**
>
> The following points are a list of important concepts for drawing VMSB.
>
> - **Information geometry.** Bregman divergences and MD are closely related to information geometry [3,4]. Our paper deals with the geometric problem with infinite-dimensional analysis, and we use Wasserstein geometry of distributions as an infinite-dimensional analogue of the Riemannian geometry.
> - **Learning theory.** SB algorithms fundamentally struggle with imperfect learning signals (**Figs. 4 & 5**). Inspired by learning theory, we devised an online learning algorithm for a stable SB algorithm through the OMD theory.
> - **Measure theory.** One of our theoretical contribution lies in the application of log Sobolev inequalities (**Assumption 3**) into our analysis. Understanding of the measure theory is required to understand **Theorem 2** on its technical side, and the choice of GMM parameterization.

---

> ### Author Response · Authors · 2024-11-21
> **Official Response to Reviewer FTEB (2/3)**
>
> **Q1 [Fig. 1]**
>
> The space $\mathcal{C}$ represents the coupling space of bidirectional transportation given by $d\pi = e^{\varphi\oplus\psi - c_{\varepsilon}} d (\mu \otimes \nu)$, subject to the probabilistic constraints. This is also equipped with extra conditions, as stated in the definition of $\mathcal{C}$ (**L200**), to ensure a nice geometric structure. **Fig. 1** is a simplified representation of the solution space, where each SB solution $\pi_t$ is represented as a point.
>
> One could attain a better understanding by considering the 3D illustration in **Fig. 3**. The figure shows that the projection spaces $\Pi^\perp_\mu$ and $\Pi^\perp_\nu$ represent distinct surfaces. Thus $\mathcal{C}$ and $(\Pi^\perp_\mu,\Pi^\perp_\nu)$ meet along curves, and this concept is simplified as **Fig. 1**. In **Fig. 1**, both types of learning involve $\mathcal{C}$, yet IPF iterations have a slightly more sophisticated geometric interpretation of alternating Bregman projections. For ideal algorithms, both MD iterations will iteratively find solutions $\pi^\ast$. In this paper, we are dealing with problems where we can not expect such ideal conditions.
>
> **Q2 [Lemma 2]**
>
> There was a missing superscript mistake in **Lemma 2;** please check the revised version addressing this point. The original intention was to link KL functionals to the original formulation Wasserstein gradient operator $\nabla_{\mathbb{W}}$ by Otto [1]. Of course, both expressions $\nabla_\mathbb{W} F(\pi_t)$ and $\nabla_\mathbb{W} f(\vec{\pi}^x_t)$ are equivalent in SB under the disintegration theorem. The paper deliberately chose the latter form for its actual usage in our VMSB algorithm in **§ 5**.
>
> **Q3 [Eq. (6)]**
>
> We respectfully point out the influence of the divergence in **Eq (6)** indicates the proximity from the current point and is controlled by $\frac{1}{\eta_t}$. We find that gradually lowering $eta_t$ guarantees general convergence of OMD in SB. As you noticed, this is a fundamental equation. All of first-order algorithms discussed in this paper fall into **(6)** with specialization, where **Lemmas 1 & 2** are dedicated to this standpoint. Arbitrary $F_t(\pi)$ that makes convergence to $\pi^\ast$ can be used to train SB can be chosen, and our choice of online cost $F_t$ is drawn from the static case of Wasserstein gradient descent in **Lemma 2**.
>
> **Q4 [$\alpha_t$]**
>
> We agree that certain enumerating styles $a_t$ and $\alpha_t$ were too short or did not match the overall formal presentation style. Following your insightful comment, we enhanced the clarity of expressions in **L209** and **L211**, which became more formal and appropriate for both expressions.
>
> **Q5 [Lemma 2]**
>
> Yes, it is the apparent, yet vital premise that algorithms do not access $\pi^\ast$ ($\mu$ & $\nu$ might be accessed in assumptions of some OT papers).  Historically, this is also the reason why $\tilde{F}_t$ of Sinkhorn is defined as it is. Therefore, implementation of Wasserstein gradient flows and acquiring plausible cost $F_t$ without access of $\pi^\ast$ were our core algorithmic considerations.
>
> **Q6 [α-IMF & BM$^2$]**
>
> We appreciate your insightful comments and great papers [6] and [7]. We would like to clarify that we were unaware of these recent articles that were released this September. We generally agree that both achieved great progress in the generalization of SB learning, yet the directions of progress can also be understood from this paper’s perspective.
>
> - **α-IMF [6]**
>     - Similar to Sinkhorn, IMF deals with two distinct surfaces, named Markovian projection space and reciprocal projection space (**Fig. 1** of [6]). To the best of our knowledge, there is no definite answer to which choice is better.
>     - The paper proposes α-DSBM, where we can recover DSBM-IMF (Shi et al., 2023) by α=1. Considering $\alpha \to 0$ enables performing non-iterative, online versions of IMF, and α becomes a step size for approximation.
>     - We conducted several comparison studies with bridge matching by considering LightSB-M [8] and DSBM. See that **Theorem 3.3** of Gushchin et al. [8] matches **Eq. (9)** of [6]. Thus, we respectfully point out that the conducted experiments can cover the α-DSBM case by considering the performance of LightSB-M and DSBM.
> - **BM$^2$ [7]**
>     - BM$^2$, or coupled bridge matching, demonstrated an intriguing idea aiming for a simple loss. We think BM$^2$ offers streamlined procedures and balanced samples.
>     - In **Algorithm 1** of [7], the loss is computed by bridge matching, and the reference dynamics of the Brownian bridge are efficiently collected. We think this approach is promising, although it would be challenging to analyze what the theoretical benefits would be without some additional assumptions.
>     - By comparing with the experimental results of [7], we believe our VMSB method shows better performance in the EOT benchmark.

---

> ### Author Response · Authors · 2024-11-21
> **Official Response to Reviewer FTEB (3/3)**
>
> **Q7 [§ 4.2]**
>
> The high-level idea of **Assumption 1** is from the observation that a computational SB algorithm should be implemented through approximation. We devised a mild assumption in the sense that the dual points could be stationary only with their mean when $t\to\infty$. It is apparent that if the algorithm is optimal and finds probabilistic convergence to π*, VMSB also converges according to **Theorem 1**.
>
> In **Fig. 4**, we viewed the reference estimation fitted using LightSB as $\pi^\circ_t$. This setting can be changed to LightSB-M and adversarial learning, and we assumed that such an approximation of SB learning would create a suboptimal region that does not deviate drastically after a certain amount of time. The validity of assumption was considerably supported by the experiments.
>
> **Q8 [§ 4.3]**
>
> Following your comment, our revised manuscript has fixed the notation mismatch, which caused confusion about **Theorem 3**. In this paper, we mainly used $\rho_\tau \in \mathcal{C}$ for $\tau\in[0,\infty)$ in § 4.3 to denote the one-parameter semigroup for modeling OMD update from conditional distribution $\vec{\pi}$ₜ to $\vec{\pi}_{t+1}$. Therefore, $t$ is the time step for MD, which outputs static SB model $\pi$, and $\tau$ is the sub-time step. To avoid confusion, we have fixed the notation in **Theorem 3**.
>
> **Q9 [Eq. (14)]**
>
> Yes, **Eq. (14)** utilizes the parameterization from LightSB to perform variational OMD in the submanifold of Wasserstein-2 space, termed the Wasserstein Fisher Rao geometry.
>
> **Q10**
>
> In theory, if a push-forward measure from a non-trivial marginal distribution can be analytically drawn without critical errors, we can easily extend our results to dynamical models. For instance, one could apply multiple reparameterization tricks across the Wiener measure and draw a dynamic version of **Proposition 2**. Currently, flow-based NNs might be advancing in this direction. However, there can be multiple technical hurdles to overcome. Therefore, we mostly considered the theoretical side of learning and substituted the computational aspect with WFR geometry. As we believe GMMs are powerful nonlinear models for SB, we think the performance benefits under identical architecture constraints are sufficient within the scope of this paper to validate our theoretical claims.
>
> **Q11**
>
> Thank you for the insightful comment. We respectfully emphasize that MD works more meaningfully in the SB learning circumstance.
>
> - **The SB problem is convex for distributions.** We assumed that the expression $\lim_{t\to\infty} \mathbb{E}_t[\delta_C\Omega(\pi^\circ_t)]$ is stationary as a mapping of Gaussian mixtures (by consistently changing its mode) using Bregman potentials.
> - **The problem is non-convex to its parameters.**  The exponential moving averaging  (EMA) is the technique favors the "average" of historical parameters $\{\phi_t\}^\infty_t$. It stabilizes the learning process, but analyzing its meaning in the probabilistic space is difficult.
> - Our OMD algorithm finds the best model that is closest to $\delta_D(\lim_{t\to\infty}\mathbb{E}_t[\delta_C\Omega(\pi^\circ_t)]$) where the Bregman potential is used to measure distance. Therefore, we claim that MD brings pleasing outcomes regarding statistical divergence induced by KL.
>
> To further validate our point, we added a row of LightSB-EMA in **Tables 6 & 7.** The table below is the ε=1 result of cBW-UVP. This shows that VSMB consistently shows better results than LightSB-EMA, and our theoretical argument rigorously supports these gains.
>
> |  | d=2 | d=16 | d=64 | d=128 |
> | --- | --- | --- | --- | --- |
> | LightSB | 0.014 | 0.026 | 0.060 | 0.140 |
> | LightSB-EMA | 0.012 | 0.022 | 0.051 | 0.127 |
> | VMSB (ours) | 0.010 | 0.018 | 0.044 | 0.114 |
>
> Please feel free to reach out to us if you have further questions.
>
> Sincerely,
>
> Submission #5991 authors
>
> [1] Felix Otto. The geometry of dissipative evolution equations: the porous medium equation. Communications in Partial Differential Equations, 2001.
>
> [2] Mohammad Reza Karimi, Ya-Ping Hsieh, and Andreas Krause. Sinkhorn flow as mirror flow: a continuous-time framework for generalizing the sinkhorn algorithm. In AISTATS, 2024.
>
> [3] Shun-ichi Amari. Information geometry and its applications. Springer, 2016.
>
> [4] Frank Nielsen. An elementary introduction to information geometry. Entropy, 2020
>
> [5] Christian Léonard. A survey of Schrödinger problem  and some of its connections with optimal transport. arXiv preprint arXiv:1308.0215, 2013.
>
> [6] Valentin De Bortoli, Iryna Korshunova, Andriy Mnih, and Arnaud Doucet. Schrödinger bridge flow for unpaired data translation. In NeurIPS, 2024.
>
> [7] Stefano Peluchetti. BM$^2$: coupled Schrödinger bridge matching. arXiv preprint arXiv:2409.09376, 2024.
>
> [8] Nikita Gushchin, Sergei Kholkin, Evgeny Burnaev, Alexander Korotin. Light and Optimal Schrödinger Bridge Matching, In ICML, 2024.

---

### Meta-Review · Area_Chair_u3ev · 2024-12-19

**Metareview:**

This work is on the topic of optimal transport and Schrödinger Bridge. The authors postulate an online learning problem with the aim to solve the Schrödinger Bridge problem with robustness. A major weakness pointed out by the reviewers is that the practical gain of the proposed method is not clear. The experimental results to illustrate the advantages of the proposed algorithm over baselines, particularly on robustness, are not convincing. In addition, some important baselines are missing. Moreover, the GMM parameterization could lead to poor scalability and mode collapsing. Finally, the theory part is a combination of prior works. The presentation also needs work. The authors are encouraged to incorporate these comments from the reviewers to improve the manuscript.

**Additional Comments On Reviewer Discussion:**

The reviewers raises some questions on the theoretical and experimental results, as well as the presentation of the paper. The authors reply by modifying the paper, adding experiments in the paper, and adding clarifications in the response. Several reviewers are not convinced and believe the paper is not ready for publication.

---

### Decision · Program_Chairs · 2025-01-22

Reject